# Kinesin-4 KIF21B is a potent microtubule pausing factor

**Wilhelmina E van Riel[1†], Ankit Rai[1†], Sarah Bianchi[2†], Eugene A Katrukha[1], Qingyang Liu[1], Albert JR Heck[3], Casper C Hoogenraad[1], Michel O Steinmetz[2], Lukas C Kapitein[1], Anna Akhmanova[1]\***

[1]Cell Biology, Department of Biology, Faculty of Science, Utrecht University, Utrecht, Netherlands; [2]Laboratory of Biomolecular Research, Department of Biology and Chemistry, Paul Scherrer Institut, Villigen PSI, Switzerland; [3]Biomolecular Mass Spectrometry and Proteomics, Bijvoet Center for Biomolecular Research, Utrecht Institute for Pharmaceutical Sciences and The Netherlands Proteomics Centre, Utrecht University, Utrecht, Netherlands

**Abstract** Microtubules are dynamic polymers that in cells can grow, shrink or pause, but the factors that promote pausing are poorly understood. Here, we show that the mammalian kinesin-4 KIF21B is a processive motor that can accumulate at microtubule plus ends and induce pausing. A few KIF21B molecules are sufficient to induce strong growth inhibition of a microtubule plus end in vitro. This property depends on non-motor microtubule-binding domains located in the stalk region and the C-terminal WD40 domain. The WD40-containing KIF21B tail displays preference for a GTP-type over a GDP-type microtubule lattice and contributes to the interaction of KIF21B with microtubule plus ends. KIF21B also contains a motor-inhibiting domain that does not fully block the interaction of the protein with microtubules, but rather enhances its pause-inducing activity by preventing KIF21B detachment from microtubule tips. Thus, KIF21B combines microtubule-binding and regulatory activities that together constitute an autonomous microtubule pausing factor.

**\*For correspondence:**
a.akhmanova@uu.nl

[†]These authors contributed equally to this work

**Competing interests:** The authors declare that no competing interests exist.

## Introduction

The organization and function of microtubule (MT) networks critically depend on the dynamic instability of MTs – their ability to spontaneously switch between phases of growth and shrinkage (*Desai and Mitchison, 1997*). This MT behavior can be reconstituted in vitro using purified tubulin. In cells, numerous MT-associated proteins (MAPs) modulate the dynamic instability of MTs by controlling specific phases of MT dynamics. MAPs can accelerate MT polymerization, decorate and stabilize MTs, promote switching between growth and shortening (catastrophes), or induce reverse transitions (rescues). Many of these activities have been reconstituted in vitro in systems with purified components (*Akhmanova and Steinmetz, 2015*; *Gardner et al., 2013*). Importantly, the plus ends of MTs growing from purified tubulin in vitro typically undergo sharp transitions between growth and shortening, while in cells MT plus ends often exist in a paused state. This difference is due to the presence of cellular factors that can dampen or even block MT dynamics, but the nature of these factors and the molecular mechanisms underlying their activity are still poorly understood.

Proteins controlling MT dynamics can be broadly divided into molecular motors and MAPs that lack motor activity. The two types of MT-dependent motors, kinesins and dyneins, can both interact with MT ends to affect their dynamics (*Hu et al., 2015*; *Laan et al., 2012*; *Su et al., 2012*; *Walczak et al., 2013*). Amongst the kinesins, very different modes of regulation of MT polymerization have been reported. For example, the kinesin-13 family members have a centrally located motor domain, are immotile and use the energy of ATP hydrolysis to modify the structure of MT ends,

**eLife digest** Microtubules are tiny tubes that cells use as rails to move various cell compartments and structures to different locations within the cell. They are made of building blocks called tubulin and form extensive networks across the cell. Depending on the cell's needs, microtubule networks can be rapidly assembled and disassembled by adding or removing tubulin subunits at the ends of individual microtubules. While a lot is known about how cells regulate the growth and shrinkage of microtubules, much less is known about the factors that can pause these processes and thus stabilize a microtubule.

Proteins belonging to the kinesin family are molecular motors that can walk along microtubules and control how microtubules grow and shrink. A kinesin known as KIF21B is found in several types of cells including neurons and immune cells and genetic alterations in this protein have been linked with several neurodegenerative diseases. KIF21B is made up of three regions: a motor domain, a stalk and a tail domain that binds to microtubules. Recent studies have suggested that this kinesin affects the ability of one end of microtubules (known as the plus end) to grow.

Here, van Riel, Rai, Bianchi et al. used a biochemical approach to investigate the activity of KIF21B. The experiments show that KIF21B can walk to the plus end of microtubules and efficiently pause growth. Small numbers of KIF21B molecules are enough to inhibit microtubule growth and this activity depends on the motor domain and the tail domain of KIF21B working together. These experiments were performed a cell-free system and so the next challenge is to investigate how KIF21B works in living cells, including neurons and immune cells.

induce catastrophes and enhance depolymerization (*Moores and Milligan, 2006*; *Walczak et al., 2013*). Kinesin-8 family members have an N-terminal motor domain and can move processively to the plus ends where they induce MT disassembly or suppress MT dynamics (*Gardner et al., 2008*; *Stumpff et al., 2011*; *Su et al., 2012*).

Another family of MT-regulating kinesins is kinesin-4. The best-studied family member, KIF4/Xklp1, reduces the MT growth rate and suppresses catastrophes (*Bieling et al., 2010*; *Bringmann et al., 2004*). During mitosis, KIF4/Xklp1 binds to PRC1, a potent anti-parallel MT bundler involved in the formation of the central spindle (*Kurasawa et al., 2004*; *Zhu and Jiang, 2005*). The complex of KIF4/Xklp1 and PRC1 accumulates at MT ends and strongly inhibits MT elongation (*Bieling et al., 2010*; *Subramanian et al., 2013*). Another kinesin-4 family member, KIF7, is immotile; it participates in organizing the tips of ciliary MTs by reducing the MT growth rate and promoting catastrophes (*He et al., 2014*).

Other members of the kinesin-4 family are the two large motors KIF21A and KIF21B. KIF21A has been studied quite extensively, because point mutations in this protein cause a dominant eye movement syndrome, Congenital Fibrosis of the Extraocular Muscles type 1 (CFEOM1) (*Heidary et al., 2008*; *Yamada et al., 2003*). KIF21A is ubiquitously expressed, but the pathology in patients is associated with a specific defect in the development of the oculomotor nerve, likely due to a perturbation of axon guidance (*Cheng et al., 2014*; *Heidary et al., 2008*). In vitro, the KIF21A motor domain behaves similarly to that of Xklp1 – it reduces the MT growth rate and suppresses catastrophes (*van der Vaart et al., 2013*). There are also indications that in addition to controlling MT dynamics, KIF21A plays a role in membrane transport (*Lee et al., 2012*). All CFEOM1-associated mutations in KIF21A localize either to the motor domain or to a predicted short coiled-coil domain in the stalk region of the molecule, and each of them prevents the autoinhibitory interaction between these two elements (*Bianchi et al., 2016*; *Cheng et al., 2014*; *van der Vaart et al., 2013*). The dominant character of the CFEOM1 syndrome is thus connected to the increased activity of the mutant KIF21A kinesin caused by the loss of autoinhibition (*Cheng et al., 2014*; *van der Vaart et al., 2013*).

KIF21A and KIF21B are highly similar in sequence: they both contain an N-terminal motor domain followed by a stalk with several predicted coiled coils and a C-terminal WD40 domain (*Marszalek et al., 1999*). KIF21B has been reported to be expressed in brain, eye and spleen and to be enriched in dendrites of neurons (*Marszalek et al., 1999*). Polymorphisms in the *KIF21B* gene have been associated with multiple sclerosis and other inflammatory disorders (*Anderson et al.,*

*2009*; *Barrett et al., 2008*; *Goris et al., 2010*; *Yang et al., 2015*). An increase in expression of KIF21B was connected to accelerated progression of neurodegenerative diseases (*Kreft et al., 2014*), and microduplications of the locus bearing the *KIF21B* gene were linked to neurodevelopmental abnormalities (*Olson et al., 2012*). Furthermore, it has been demonstrated that KIF21B binds to the ubiquitin E3 ligase TRIM3, which can modulate the function of KIF21B (*Labonté et al., 2013*). The motor was also implicated in the surface delivery of GABA$_A$ receptors in neurons, but the interaction is likely indirect (*Labonté et al., 2014*).

While this manuscript was in preparation, a paper describing a mouse knockout of KIF21B has been published (*Muhia et al., 2016*). This work showed that mice lacking KIF21B are viable, but display defects in learning and memory, which are likely to be due to several dendritic phenotypes, such as reduced complexity of the dendritic arbor and diminished density of dendritic spines that correlate with defects in synaptic transmission. An even more recent paper showed that KIF21B contributes to activity-dependent regulation of some aspects of retrograde trafficking of brain-derived neurotrophic factor-TrkB complexes in cultured neurons (*Ghiretti et al., 2016*). Both papers showed that KIF21B can affect MT plus-end dynamics, although the results were complex: while both studies reported an increase in MT growth processivity upon KIF21B loss, MT grew slower in *Kif21b* knockout neurons, but faster in neurons depleted of KIF21B by RNA interference (*Ghiretti et al., 2016*; *Muhia et al., 2016*). In vitro reconstitution work suggested that KIF21B increases MT growth rate and catastrophe frequency, although, surprisingly, the purified protein mostly associated with depolymerizing MT plus ends in these experiments (*Ghiretti et al., 2016*).

Here, we have used in vitro single molecule assays to systematically explore how the biochemical activity of KIF21B depends on its domain architecture. We found that KIF21B is a processive kinesin that walks to and accumulates at MT plus ends. The dimeric KIF21B motor domain was sufficient to reduce MT growth rate, while the full-length molecule could 'hold on' to the growing MT tip and induce its pausing. Strikingly, a few KIF21B molecules were sufficient to trigger and sustain a pause. In cases when KIF21B persisted at the MT tip but did not induce pausing, MT growth perturbation and catastrophes were observed. The potent effect of KIF21B on MT plus-end polymerization is due to the presence of two MT-binding regions in its tail, which help to prevent kinesin dissociation from the tip of the growing MT. We also found that the region responsible for autoinhibition in KIF21A (*Bianchi et al., 2016*) is conserved in KIF21B. However, instead of blocking the motor, this element reduced motor detachment from growing MT plus ends and thus contributed to MT pause induction. Taken together, our data show how the interplay between the motor domain and MT-binding and regulatory regions makes KIF21B a highly potent regulator of MT plus-end dynamics.

## Results

### KIF21B can block MT elongation in cells

To get insight into the ability of KIF21B to regulate MT dynamics, we have expressed the full-length protein with a C-terminal GFP tag in COS-7 cells, which do not express endogenous KIF21B. Unlike its paralogue KIF21A, which is largely diffuse when expressed in similar conditions (*van der Vaart et al., 2013*), KIF21B bound to MTs and accumulated at their ends at the cell periphery (*Figure 1A*). Live cell imaging showed that KIF21B processively moves along MTs with an average speed of 0.63 ± 0.22 µm/s (mean±SD) (*Figure 1B*); this velocity is three times faster than that recently described for HaloTag-labeled KIF21B in neurons (*Ghiretti et al., 2016*). In internal cell regions, where no clear accumulation of the motor at growing MT ends was observed, the expression of KIF21B led to a ~1.5 fold reduction in the MT growth rate measured with the MT plus-end marker EB3-TagRFP-T (*Stepanova et al., 2003*; *van der Vaart et al., 2013*) (*Figure 1C*). At the cell periphery, strong accumulation of KIF21B-GFP and stalling of MT growth were observed; however, the exact quantification of MT dynamics at the periphery of KIF21B-overexpressing cells was severely complicated by the frequent sliding of MT tips against each other. Interestingly, in cells with high expression levels of KIF21B, the MT network often strongly retracted, leaving significant portions of the cytoplasm largely devoid of MTs (*Figure 1D*). The remaining MT network in such cells was still dense and appeared to be 'corralled' by KIF21B accumulations. Time lapse imaging showed that the retraction of the MT network in KIF21B-expressing cells was a gradual process that could be detected during 1–2 hr of observation (*Figure 1—figure supplement 1A*). In addition, expression of

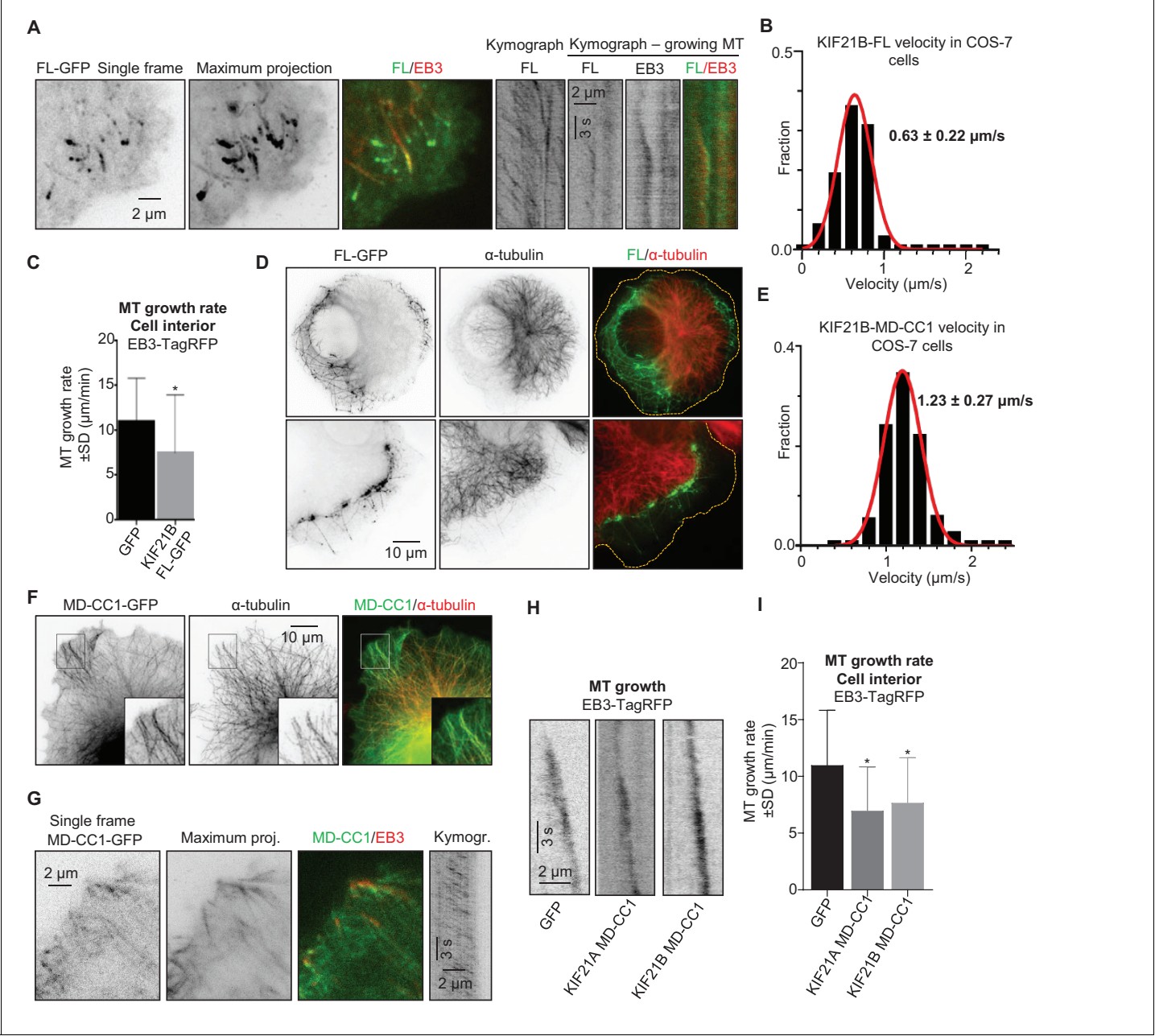

**Figure 1.** KIF21B inhibits MT growth in cells. (**A**) COS-7 cells were transiently transfected with KIF21B-FL-GFP and EB3-TagRFP-T and imaged using TIRF microscopy. Represented are a single-frame, maximum intensity projection of 500 frames for the GFP channel and an overlay of a single GFP frame in green and TagRFP-T in red. Kymographs illustrate the motility of KIF21B along the MT and its significant accumulation at a stationary but not a growing MT plus end. (**B**) Histogram of KIF21B-FL-GFP kinesin velocities in COS-7 cells is shown with black bars. Red line shows fitting with a normal distribution. n = 378 in 10 cells in two independent experiments. (**C**) Quantification of MT growth rate, measured by tracking EB3 labeled comets in cell interior. Three to ten MTs per cell were analyzed; n = 183 in 21 cells for GFP control, n = 214 in 12 cells for KIF21B-FL-GFP expressing cells, two independent experiments, p<0.0001, Mann-Whitney U test (indicated by an asterisk). (**D**) COS-7 cells were transiently transfected with KIF21B-FL-GFP, fixed the next day and stained for α-tubulin. Cell edges are indicated with yellow dashed lines in the overlay. (**E**) Histogram of KIF21B-MD-CC1-GFP velocities in COS-7 cells is shown with black bars. Red line shows fitting with a normal distribution. n = 431 in 14 cells in two independent experiments. (**F**) COS-7 cells transiently transfected with KIF21B-MD-CC1-GFP were fixed and stained for α-tubulin. (**G**) COS-7 cells were transiently transfected with KIF21B-MD-CC1-GFP and EB3-TagRFP-T and imaged using TIRF microscopy. Represented are a single-frame, maximum intensity projection of 500 frames for the GFP channel, an overlay of a single GFP frame in green and TagRFP-T in red and a kymograph along one of the EB3-labeled MTs showing the motility of the kinesin along the MT. (**H**) Kymographs showing EB3-TagRFP-T comet displacement in control COS-7 cells or cells expressing the MD-CC1 fragments of KIF21A or KIF21B. (**I**) Quantification of MT growth rate illustrated in H. n = 183 in 21 cells for GFP control, n = 136

*Figure 1 continued on next page*

*Figure 1 continued*

in 15 cells for KIF21A-MD-CC1-GFP, n = 179 in 22 cells for KIF21B-MD-CC1-GFP, two independent experiments, p<0.0001, Mann-Whitney U test (indicated by asterisks).

The following source data and figure supplement are available for figure 1:

**Source data 1.** An excel sheet with numerical data on the quantification of kinesin velocities and MT growth rate in COS-7 cells represented as plots in *Figure 1B,C,E,I*.

**Figure supplement 1.** Effects of KIF21B expression on MT organization and regrowth in cells

KIF21B prevented full extension of MTs in experiments where the MT network recovered from treatment with the MT-depolymerizing drug nocodazole (*Figure 1—figure supplement 1B*). We conclude that at high expression levels, KIF21B can accumulate at MT plus ends, block their polymerization and cause their very slow shortening (*Figure 1—figure supplement 1A*).

## The dimeric motor domain of KIF21B slows down MT polymerization

To investigate whether blocking of MT growth could be caused by the motor domain of KIF21B alone, we used a C-terminally tagged truncated version encompassing the motor and a part of the dimeric coiled coil, but missing the tail region of the protein (KIF21B-MD-CC1; see Figure 6A for the scheme of all constructs used in this study). KIF21B-MD-CC1 also bound to and walked along MTs with a velocity of 1.23 ± 0.27 μm/s (mean±SD) (*Figure 1E*), but did not specifically accumulate at MT plus ends (*Figure 1F,G*). The observed velocity was again ~3 times faster than that observed in neurons (*Ghiretti et al., 2016*), which might be due to the fact that in neuronal cells kinesins are slowed down by specific MAPs. Expression of KIF21B-MD-CC1 reduced MT growth rate by ~1.6 fold (*Figure 1H,I*), which is similar to what we previously observed with a comparable deletion mutant of KIF21A (*van der Vaart et al., 2013*).

To investigate whether the observed effect of KIF21B-MD-CC1 is direct, we next purified GFP alone and KIF21B-MD-CC1, which was C-terminally tagged with GFP, from HEK293T cells (*Figure 2—figure supplement 1*). Using mass spectrometry, we confirmed that this purification method did not result in co-isolation of known MT regulators (*Supplementary file 1*). Analysis of fluorescence intensity of single KIF21B-MD-CC1-GFP molecules in comparison to monomeric GFP and dimeric EB3-GFP indicated that they were dimers, as expected (*Figure 2A*, *Supplementary file 2*). This conclusion was confirmed by two-step photobleaching profiles (*Figure 2B*) and was in agreement with the published data obtained in HeLa cell lysates with a similar construct (*Ghiretti et al., 2016*).

We then examined the effect of KIF21B-MD-CC1 on dynamic MTs in vitro by using a Total Internal Reflection Fluorescence (TIRF) microscopy-based MT polymerization assay (*Bieling et al., 2007*; *van der Vaart et al., 2013*). In this assay, MTs are grown from GMPCPP-stabilized MT seeds attached to glass coverslips in the presence of fluorescently labeled or unlabeled tubulin and proteins of interest. We performed such assays in the presence of fluorescently labeled tubulin alone or with the addition of mCherry-EB3 (*Montenegro Gouveia et al., 2010*), as this fluorescent protein greatly facilitates the detection of small changes in the position of the growing MT plus end, and our previous work showed that it did not alter the effect of KIF21A on the MT plus-end dynamics (*van der Vaart et al., 2013*). Moreover, since EB proteins are highly abundant and ubiquitous MT plus-end binding proteins, EB-bound MT plus ends can be expected to represent 'natural' substrates for other MT regulators.

KIF21B-MD-CC1 displayed short plus end-directed runs on MTs and could reach MT plus ends but detached from them upon arrival and thus did not accumulate at the MT tips (*Figure 2C*). The intensity of individual KIF21B-MD-CC1 molecules moving on MTs was on average ~1.8 times higher than the intensity of individual monomeric GFP molecules immobilized in a separate chamber on the same coverslip (*Figure 2D*, *Supplementary file 2*). While a ratio of two might be expected for a dimer, we need to take into account that the motors are further away from the coverslip and that the evanescent field used for excitation decays exponentially. Given a penetration depth *d* of 80–200 nm, being 25 nm (MT diameter) away from the coverslip will yield a 12–27% reduction in

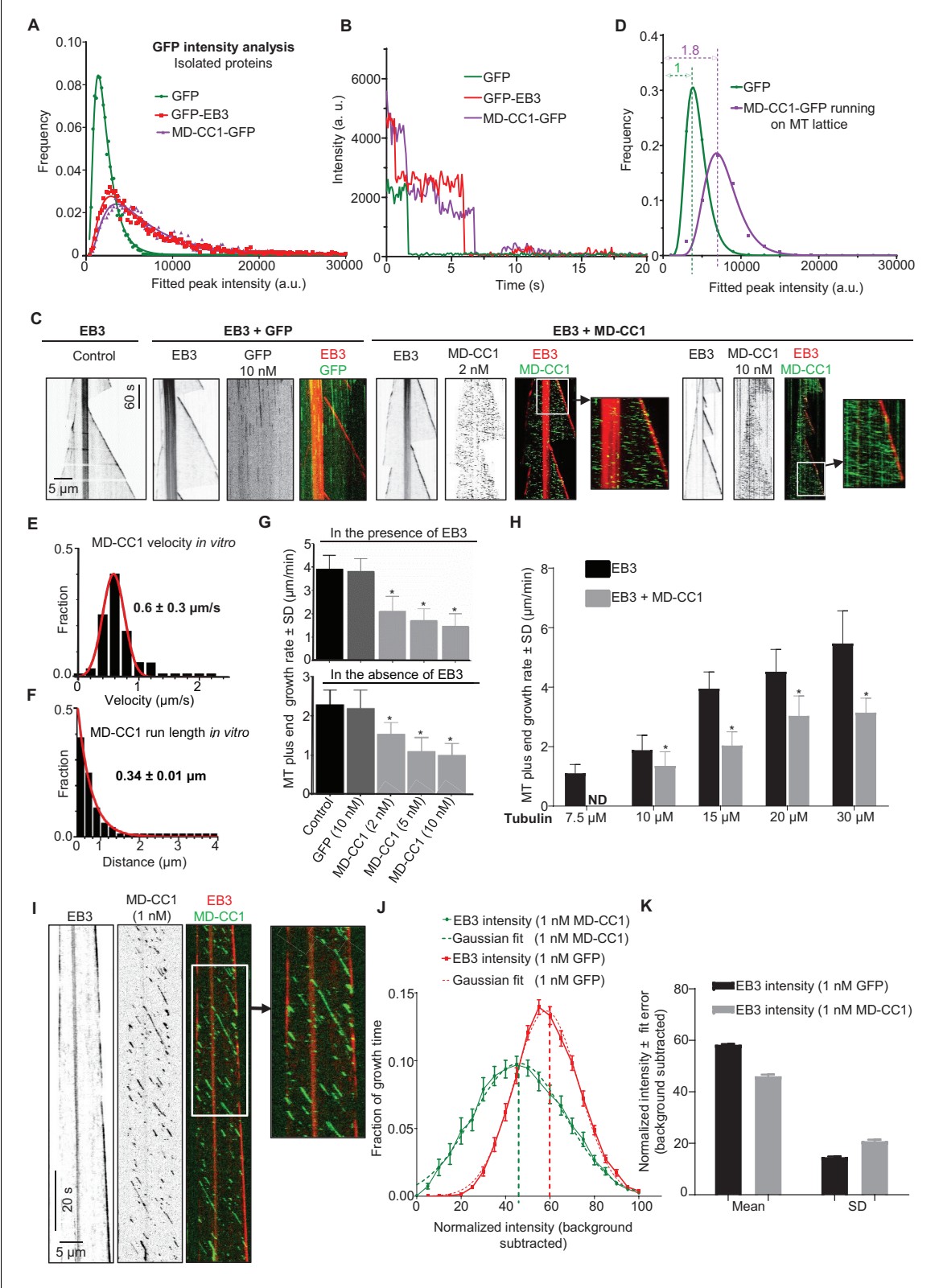

**Figure 2.** Dimeric motor domain of KIF21B slows down MT polymerization in vitro. (**A**) Histograms of fluorescence intensities at the initial moment of observation of single molecules of the indicated proteins immobilized on coverslips (symbols) and the corresponding fits with lognormal distributions (lines). n = 3107, 5802 and 4674 molecules and fluorophore density was 0.15, 0.28 and 0.23 $\mu m^{-2}$ for GFP, GFP-EB3 and KIF21B-MD-CC1-GFP proteins. (**B**) Representative photobleaching time traces of GFP, GFP-EB3 and KIF21B-MD-CC1-GFP individual molecules (background subtracted). (**C**)

*Figure 2 continued on next page*

*Figure 2 continued*

Kymographs illustrating the dynamics of MTs grown in vitro in the presence of 20 nM mCherry-EB3 alone, with 10 nM purified GFP or with 2 and 10 nM KIF21B-MD-CC1-GFP. Zooms of the boxed areas are shown on the right. Kymographs were generated from movies acquired using a Photometrics Evolve 512 EMCCD camera (Roper Scientific) (stream acquisition, exposure time 500 ms). (D) Histograms of fluorescence intensities of single GFP molecules immobilized on coverslips and KIF21B-MD-CC1-GFP moving on MTs in a separate chamber on the same coverslip (symbols) and the corresponding fits with lognormal distributions (lines). n = 4815 and 1381 molecules; fluorophore density was 0.16 and 0.09 µm$^{-2}$ for GFP and KIF21B-MD-CC1-GFP proteins (for the latter, MT-containing regions were manually selected for analysis). Dashed lines show corresponding relative median values. (E) Histogram of KIF21B-MD-CC1-GFP velocities in vitro is shown with black bars. Red line shows fitting with a normal distribution. n = 675 in two independent experiments. (F) Histogram of KIF21B-MD-CC1-GFP run lengths in vitro is shown with black bars. Red line shows fitting with an exponential distribution. n = 675 in two independent experiments. (G) Upper panel - quantification of the MT growth rate illustrated in C. n = 71 for control, n = 65 for purified GFP, n = 71, 67 and 54 for 2, 5 and 10 nM KIF21B-MD-CC1-GFP, respectively. Lower panel shows quantification of the MT growth rate with 15 µM tubulin alone or with 10 nM purified GFP or with 2, 5 and 10 nM KIF21B-MD-CC1-GFP as illustrated in *Figure 2—figure supplement 2*. n = 67 for control, n = 57 for purified GFP, n = 71, 66 and 80 for 2, 5 and 10 nM KIF21B-MD-CC1-GFP, respectively, two independent experiments, p<0.0001, Mann-Whitney U test (indicated by asterisks). (H) Quantification of the MT growth rate with different concentrations of tubulin along with 20 nM EB3 in the absence and presence of 2 nM KIF21B-MD-CC1-GFP as illustrated in *Figure 2—figure supplement 3*. ND; not determined, n = 71 for all conditions. two independent experiments, p<0.0001, Mann-Whitney U test (indicated by asterisks). (I) Kymographs illustrating the dynamics of MTs grown in vitro in the presence of 20 nM EB3 and 1 nM KIF21B-MD-CC1-GFP. Zoom of the boxed area is shown on the right. Kymographs were generated from a movie acquired using Photometrics Evolve 512 EMCCD camera (Roper Scientific) (stream acquisition, exposure time 100 ms). (J) Distribution of EB3 fluorescence intensity fluctuations over time (normalized to its maximum value during a course of a growth event) at MT tip in the presence of 1 nM GFP or 1 nM KIF21B-MD-CC1-GFP (solid line) with Gaussian fit (dotted line). n = 25 in both cases. Thick dotted lines show the peak of the Gaussian fitting. MT dynamics assay was performed in the presence of 15 µM tubulin, 20 nM EB3 and 1 nM GFP or 1 nM KIF21B-MD-CC1-GFP in the separate chambers of the same coverslip. (I) Plot of the mean and SD values of Gaussian fits shown in *Figure 2J*.

The following source data and figure supplements are available for figure 2:

**Source data 1.** An excel sheet with numerical data on the quantification of KIF21B-MD-CC1-GFP dimer analysis, photobleaching-step analysis, velocities, run length, effects on MT growth rate and distribution of EB3 fluorescence intensity represented as plots in *Figures 2A,B,D,E–H,J*.

**Figure supplement 1.** Coomassie blue stained gels with purified GFP, KIF21B-FL-GFP and its deletion mutants.

**Figure supplement 2.** Kymographs illustrating in vitro dynamics of MTs grown in the presence of 15 µM tubulin in the absence and presence of 10 nM purified GFP or 2, 5 and 10 nM KIF21B-MD-CC1-GFP.

**Figure supplement 3.** Effects of the dimeric motor domain of KIF21B on MT polymerization in vitro

**Figure Supplement 3—Source Data 1.** An excel sheet with numerical data on the quantification of the MT minus end growth rates represented as plot in *Figure 2—figure supplement 3B*.

intensity (i.e. $e^{-25/d}$) (*Grigoriev and Akhmanova, 2010*). We further note that the intensity distribution of KIF21B-MD-CC1 molecules walking on MTs lacked the tail in the high-intensity range that was observed for molecules immobilized on glass (compare *Figure 2A and D*), suggesting that larger KIF21B-MD-CC1 oligomers present in our preparations are unable to move on MTs.

Single KIF21B-MD-CC1 molecules displayed an average velocity of 0.6 ± 0.3 µm/s (mean and SD) and an average run length of 0.34 ± 0.01 µm (exponential fit to histogram and error of fit) (*Figure 2E,F*). KIF21B-MD-CC1 caused a concentration-dependent reduction of the MT plus-end growth rate both in the absence and in the presence of mCherry-EB3, while GFP alone had no effect (*Figure 2C,G*, *Figure 2—figure supplement 2*). This effect was similar to that observed previously with the kinesin-4 Xklp1 (*Bieling et al., 2010*; *Bringmann et al., 2004*) and with the dimeric motor domain of KIF21A (*van der Vaart et al., 2013*). A decrease in MT growth rate was observed at tubulin concentrations ranging from 10 to 30 µM, while at 7.5 µM tubulin, 2 nM KIF21B-MD-CC1 was sufficient to almost completely prevent MT outgrowth (*Figure 2H*, *Figure 2—figure supplement 3A*). In contrast, minus end growth was not affected in the presence of KIF21B-MD-CC1 (*Figure 2—figure supplement 3B*), indicating that the effect of this kinesin on MT dynamics is plus end-specific.

How can KIF21B-MD-CC1 inhibit MT growth without accumulating at MT tips? It is possible that transient association of the dimeric motor with MT ends might be sufficient to briefly affect the structure of the tip and thus reduce its growth rate. If this were the case, even infrequent events of KIF21B-MD-CC1arrival to the MT tip could cause some perturbation of MT growth, and we reasoned

such perturbations might be reflected in the brightness of the EB3 signal. To test this idea, we performed the assay in the presence of 1 nM KIF21B-MD-CC1 to create a situation when individual KIF21B-MD-CC1 would be occasionally hitting the MT tip, and used faster image acquisition conditions (100 ms/frame, *Figure 2I*), so that we could observe such events more clearly. Indeed, in these conditions, we observed irregularities of EB3 signal at the growing MT plus ends. To quantify this effect, we analyzed fluctuations of EB3 intensity in the presence of 1 nM GFP or 1 nM KIF21B-MD-CC1-GFP in separate chambers on the same coverslip. We excluded from our analysis MTs that were within ~40 s before catastrophe, since it is known that at this point the comet intensity is reduced (*Maurer et al., 2012*; *Mohan et al., 2013*). The distribution of EB3 intensities normalized to the maximum value was significantly broader (with a lower mean and a higher standard deviation) in the presence of KIF21B-MD-CC1-GFP than with GFP alone (*Figure 2J,K*), indicating that the EB3 signal was indeed more irregular. We note that this analysis is not dependent on the absolute MT growth rate, which can affect the absolute EB3 signal, because the analyzed intensities were normalized to the maximum value. We conclude that the motor domain of KIF21B in a dimeric configuration is motile and can reduce MT plus-end polymerization rate, possibly by perturbing the structure of the growing MT tip.

## Full-length KIF21B can induce MT pausing in vitro

Next, we purified the full-length KIF21B-GFP from HEK293T cells (*Figure 2—figure supplement 1*) and confirmed by single molecule analysis that it is a dimer (*Figure 3—figure supplement 1A,B*, *Supplementary file 2*). Mass spectrometry analysis of this purified protein revealed no known MT regulators (*Supplementary file 1*). Next, we assayed the activity of KIF21B-GFP on MTs in vitro (*Figure 3A*, *Figure 3—figure supplement 1C*). Strikingly, the full-length protein showed a strong preference for GMPCPP-stabilized MT seeds, on which it landed and moved in the direction of the plus-end, while hardly any motor landing events were observed on the dynamic (presumably GDP) MT lattice (*Figure 3A,C*). KIF21B-GFP motors accumulated at the tips of the seeds, and these accumulations could prevent MT outgrowth (*Figure 3A*). Both the enrichment of KIF21B-GFP at the tip of seeds and the inhibition of MT outgrowth were more prominent for longer seeds (*Figure 3A,B*). This indicates that GMPCPP-seeds act as 'antennae' that accumulate the kinesin motor at their ends in a length-dependent manner, similar to what has been previously described for the yeast kinesins Kip3 and Kip2 (*Hibbel et al., 2015*; *Su et al., 2012*; *Varga et al., 2006*). Significant blocking of growth from MT seeds, especially the longer ones, was observed already with 3 nM KIF21B-GFP, while complete inhibition of MT outgrowth from all seeds was seen at higher KIF21B-GFP concentrations. At lower concentrations of KIF21B (0.5 nM) growth of some seeds was still blocked, but some MTs were growing, and the effect of KIF21B on MT plus-end dynamics could be analyzed.

Since KIF21B was present in the assay at low nanomolar concentrations, we could easily detect motility of individual molecules (*Figure 3A,C,D*, *Figure 3—figure supplement 1C*). In cases where MT seed extension was observed, a large proportion of KIF21B-GFP motors was unable to transfer from the stabilized seed to the freshly grown part of the MT (*Figure 3A,C,D*). However, the motors that did pass over to the freshly polymerized lattice exhibited an approximately two-fold faster motility on this lattice compared to the seed (see below). These motors displayed a high degree of processivity (runs with a length up to 8.5 µm were measured) and typically reached the MT plus end (*Figure 3A,C,D*).

A number of distinct outcomes could be detected when KIF21B-GFP molecules reached MT plus ends. The most frequent one (~40% of all events) was stalling of the kinesin at MT plus end accompanied by MT pausing or very slow growth, which could be distinguished by the loss of mCherry-EB3 signal from MT plus ends (*Figure 3C,E*). We note that we cannot be sure that MTs did not undergo short (a few hundred nanometers long) growth and shrinkage episodes in these conditions, which we could not detect due to the resolution limit of fluorescence microscopy. Seventy-one percent of all observed pauses were induced by the arrival of what appeared to be a single kinesin dimer or a single small oligomer (see below) and had an average duration of 21.0 ± 5.9 s (n = 36). Tracking of the position of kinesin and MT tip together with kinesin's fluorescence intensity over time further confirmed that during such pausing events no additional kinesins were recruited (*Figure 3C*). We have also observed pauses where additional KIF21B-GFP molecules did arrive and stall at the plus end (*Figure 3F*). Accumulation of multiple KIF21B-GFP motors resulted in prolonged inhibition of MT plus-end growth (the longest pause detected was 231 s). We note that the pausing

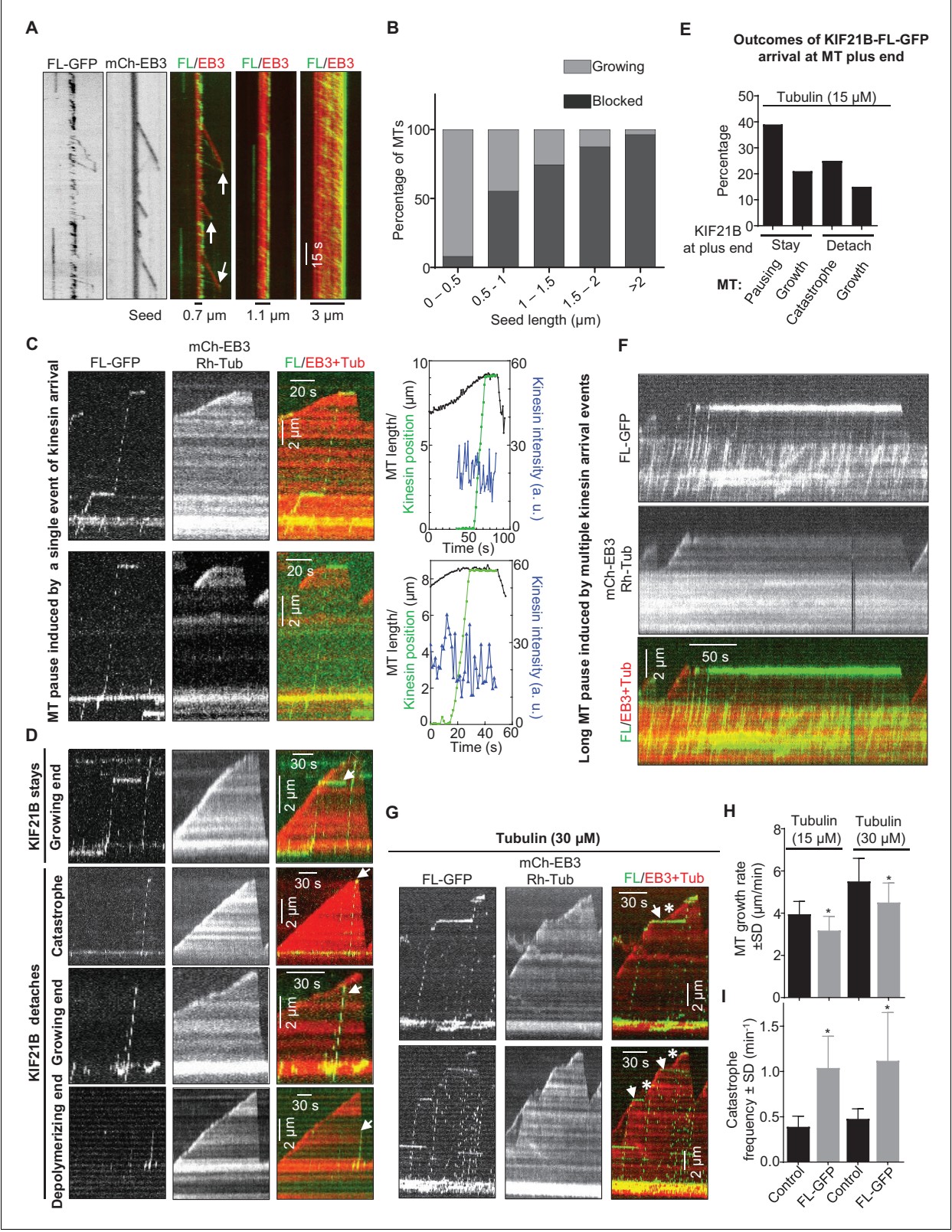

**Figure 3.** KIF21B can induce MT pausing or catastrophe in vitro. (A) Kymographs showing the behavior of KIF21B in in vitro reconstitution assays on dynamic MTs grown from Rhodamine-tubulin-labeled seeds in the presence of 15 μM tubulin, 100 nM mCherry-EB3 (red) and 3 nM KIF21B-FL-GFP (green). Kymographs were generated from movies acquired using a Photometrics Evolve 512 EMCCD camera (Roper Scientific) (stream acquisition, exposure time 500 ms). Pausing and catastrophe events are indicated by arrows. (B) Quantification of MT seed length-dependent blocking of MT

*Figure 3 continued on next page*

*Figure 3 continued*

growth by 0.5 nM KIF21B- FL-GFP in the presence of 20 nM mCherry-EB3. 188 MT seeds of different lengths were analyzed in four independent experiments. (C) Kymographs illustrating pausing events induced by KIF21B-FL-GFP (0.5 nM) on dynamic MTs in vitro in the presence of 15 µM tubulin, 20 nM mCherry-EB3, 3% Rhodamine-tubulin. MTs were grown from GMPCPP-stabilized seeds labeled with Rhodamine-tubulin. Kymographs were generated from the movies acquired using a CoolSNAP HQ2 CCD camera (Roper Scientific) with a 1.2-s interval between frames and an exposure time of 100 ms. The rightmost panels show tracked positions of the kinesins and MT tips together with the fluorescence intensity of the kinesins over time for the corresponding kymographs. (D) Kymographs illustrating various events induced by KIF21B-FL-GFP (0.5 nM) on dynamic MTs in vitro in the presence of 15 µM tubulin, 20 nM mCherry-EB3 and 3% Rhodamine-tubulin. MTs were grown from GMPCPP-stabilized seeds labeled with Rhodamine-tubulin. Different events are indicated by arrows. Kymographs were generated from movies acquired as described for *Figure 3C*. (E) Quantification of different events observed after KIF21B-FL-GFP (0.5 nM) reaches a growing MT plus end, as illustrated in C and D. n = 132 events, four independent experiments were analyzed. (F) Kymograph illustrating a long pause event induced by multiple KIF21B-FL-GFP molecules on dynamic MTs in vitro in the presence of 15 µM tubulin, 20 nM mCherry-EB3 and 3% Rhodamine-tubulin in solution. Kymographs are generated from a movie acquired as described for *Figure 3C*. (G) Kymographs illustrating the effects of KIF21B-FL-GFP (0.5 nM) on dynamic MTs in vitro in the presence of 30 µM tubulin with 3% Rhodamine-tubulin and 20 nM mCherry-EB3. MTs were grown from GMPCPP-stabilized seeds labeled with Rhodamine-tubulin. The arrows show the position of KIF21B at the site of MT pause and the asterisk indicates the growing MT tip beyond the position of KIF21B binding; note that the slope of the kymograph after KIF21B attachment is less steep than before, indicating that the growth rate is reduced. Kymographs are generated from movies acquired as described for *Figure 3C*. (H, I) Quantification of MT growth rate and catastrophe frequency in vitro in the presence of 15 or 30 µM tubulin with 20 nM mCherry-EB3 alone or together with 0.5 nM KIF21B-FL-GFP. MTs were grown in the presence of 3% Rhodamine-tubulin. For 15 µM tubulin, n = 71 for control and n = 100 for KIF21B-FL-GFP, three independent experiments. For 30 µM tubulin, n = 71 for control and n = 80 for KIF21B-FL-GFP, three independent experiments, p<0.0001 Mann-Whitney U test (indicated by asterisks).

The following source data and figure supplements are available for figure 3:

**Source data 1.** An excel sheet with numerical data on the quantification of KIF21B-FL seed blocking activity, pause induction, effects on MT growth rate and catastrophe frequency and outcomes of KIF21B-FL-GFP arrival at MT plus ends represented as plots in *Figure 3B,C,E,H,I*.

**Figure supplement 1.** Characterization of full-length KIF21B in vitro

**Figure supplements 1—source data 1.** An excel sheet with numerical data on the quantification of the KIF21B-FL dimer and photobleaching step analysis represented as plots in *Figure 3—figure supplement 1A,B*.

**Figure supplement 2.** KIF21B-FL-GFP induces pausing of a depolymerizing MT.

**Figure Supplement 2—Source Data 1.** An excel sheet with numerical data on the quantification of tracked positions of the kinesins and the MT tip together with the fluorescence intensities of the kinesins over time represented as plot in *Figure 3—figure supplement 2*.

induced by KIF21B in this assay did not dependent on the presence of EB3, because it was also observed in the presence of tubulin alone (*Figure 3—figure supplement 1C*).

Other possible outcomes of KIF21B-GFP arrival to the growing MT plus end, which all occurred at similar frequencies, were stalling of the kinesin on the MT without blocking MT elongation, catastrophe induction, which always led to kinesin dissociation from the MT tip, or immediate detachment of the kinesin from the MT plus end without perturbing MT growth (*Figure 3D,E*). KIF21B-GFP molecules that reached the plus ends of shrinking MTs usually detached without affecting MT depolymerization (*Figure 3D*, arrow in last kymograph). We did observe one example, where an event of KIF21B arrival to MT tip led to stalled MT depolymerization and a long pause with subsequent arrival of additional kinesins (*Figure 3—figure supplement 2*, Supplemental *Video 1*). However, we did not observe any events of persistent KIF21B tracking of depolymerizing MT ends.

We also examined the behavior of the full-length KIF21B at a higher tubulin concentration, 30 µM, and found that in these conditions 5 nM KIF21B was needed to induce strong inhibition of seed elongation (data not shown). At 0.5 nM KIF21B most MT seeds, including longer ones, could still grow. Since the seeds with a higher accumulation of KIF21B could still elongate, it was easier to observe multiple kinesins passing over to dynamic MTs (*Figure 3G*). The overall effects of KIF21B on MT dynamics were similar at both tubulin concentrations: KIF21B could induce pausing and catastrophes (*Figure 3G*). Measurement of MT dynamics showed that at both tubulin concentrations, KIF21B reduced MT growth rate and increased catastrophe frequency (*Figure 3H,I*).

How can relatively infrequent arrivals of KIF21B to growing MT tips significantly affect MT elongation rate? We noticed that, even when the polymerization of a MT plus end was not fully suppressed by the incoming KIF21B molecule, in cases when the kinesin did not immediately detach from the MT, MT elongation was typically strongly perturbed (*Figure 4*, *Figure 4—figure supplement 1A*). We observed many events where a MT tip was undergoing short repeated growth and shortening excursions from the point of KIF21B stalling (*Figure 4A,B*, Supplemental *Video 2*). Such MT behavior indicates that KIF21B immobilized at a MT tip prevented both its normal elongation and also its depolymerization. At 30 μM tubulin, the KIF21B stalling events typically led to very irregular growth, which ended in catastrophe (*Figure 4—figure supplement 1B*). Importantly, after the point where KIF21B was stalled, the path of the growing MT often became curved (*Figure 4*, *Figure 4—figure supplement 1*, Supplemental *Video 2*), while control MTs always grew straight in our assays. Taken together, these data suggest that KIF21B attached to the plus end might be blocking growth of a few protofilaments, leading to the extension of an incomplete and thus more flexible tube, which is more prone to undergo a catastrophe.

## A few KIF21B molecules can induce a MT pause

As indicated above, at 15 μM tubulin, we frequently observed MT pause induction by what appeared to be single KIF21B molecules or small oligomers. The ratio of the intensity of the motile KIF21B molecules responsible for the pausing events to the intensity of KIF21B dimers immobilized on the same coverslip was ~0.75–1.9 (*Figure 5A*). This is consistent with one or two KIF21B molecules, if one takes into account the decay of the evanescent field. To further examine this, we compared KIF21B intensities with those of an N-terminal fragment of kinesin-1, KIF5B residues 1–560 (denoted KIF5B-560), which is a well-studied motile dimeric kinesin (*Vale et al., 1996*). The intensities of single GFP-tagged KIF5B-560 molecules moving on MTs in vitro were ~1.8 higher than the intensities of single GFP proteins immobilized in a separate chamber on the same coverslip (*Figure 5B,C*, *Supplementary file 2*), confirming that KIF5B-560 is a dimer in our preparation, similar to what we published previously (*Doodhi et al., 2014*). We then compared the intensities of full length KIF21B molecules moving on seeds to the intensities of KIF5B-560 molecules moving on MTs in a separate chamber on the same coverslip, and found that they were also very similar (*Figure 5D*, *Supplementary file 2*). Importantly, the intensities of the few KIF21B molecules that transferred from seeds to the dynamic MT lattice were very similar to those of KIF21B molecules moving on seeds (*Figure 5E*, *Figure 5—figure supplement 1*, *Supplementary file 2*). Together, these data indicate that motile KIF21B kinesins are mostly dimers, and that these kinesins do not need to form larger oligomers in order to 'escape' from the seed to the dynamic lattice.

We have also considered that we might be underestimating the actual size of KIF21B clusters that induce pausing because of their photobleaching. For our regular imaging experiments (*Figure 3C,D,G*, *Figure 4*), we acquired the data

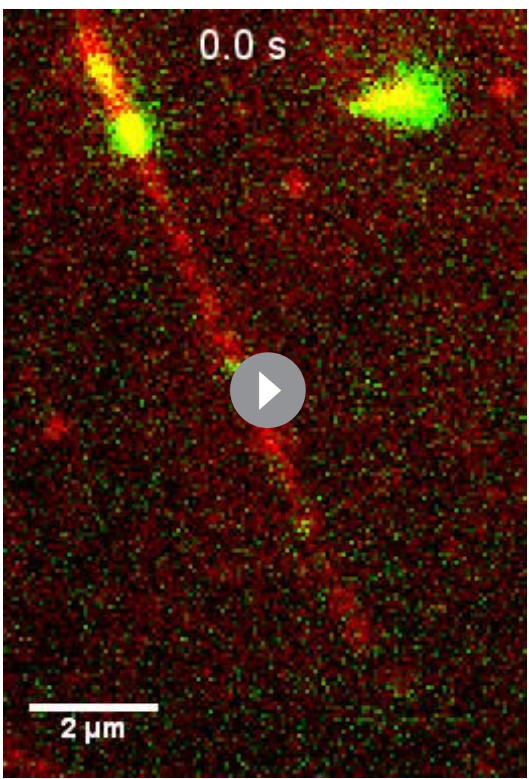

**Video 1.** KIF21B induces pausing of a depolymerizing MT. The movie shows the arrival of KIF21B-FL-GFP at the end of a depolymerizing MT and a subsequent pausing event. The arrival of additional KIF21B-FL-GFP molecules results in a long pause. The experiment was performed in the presence of 15 μM tubulin, Rhodamine-tubulin (0.5 μM), mCherry-EB3 (20 nM) and KIF21B-FL-GFP (0.5 nM). The movie consists of 200 frames acquired with a 1.2-s interval between frames and an exposure time of 100 ms. Scale bar, 2 μm.

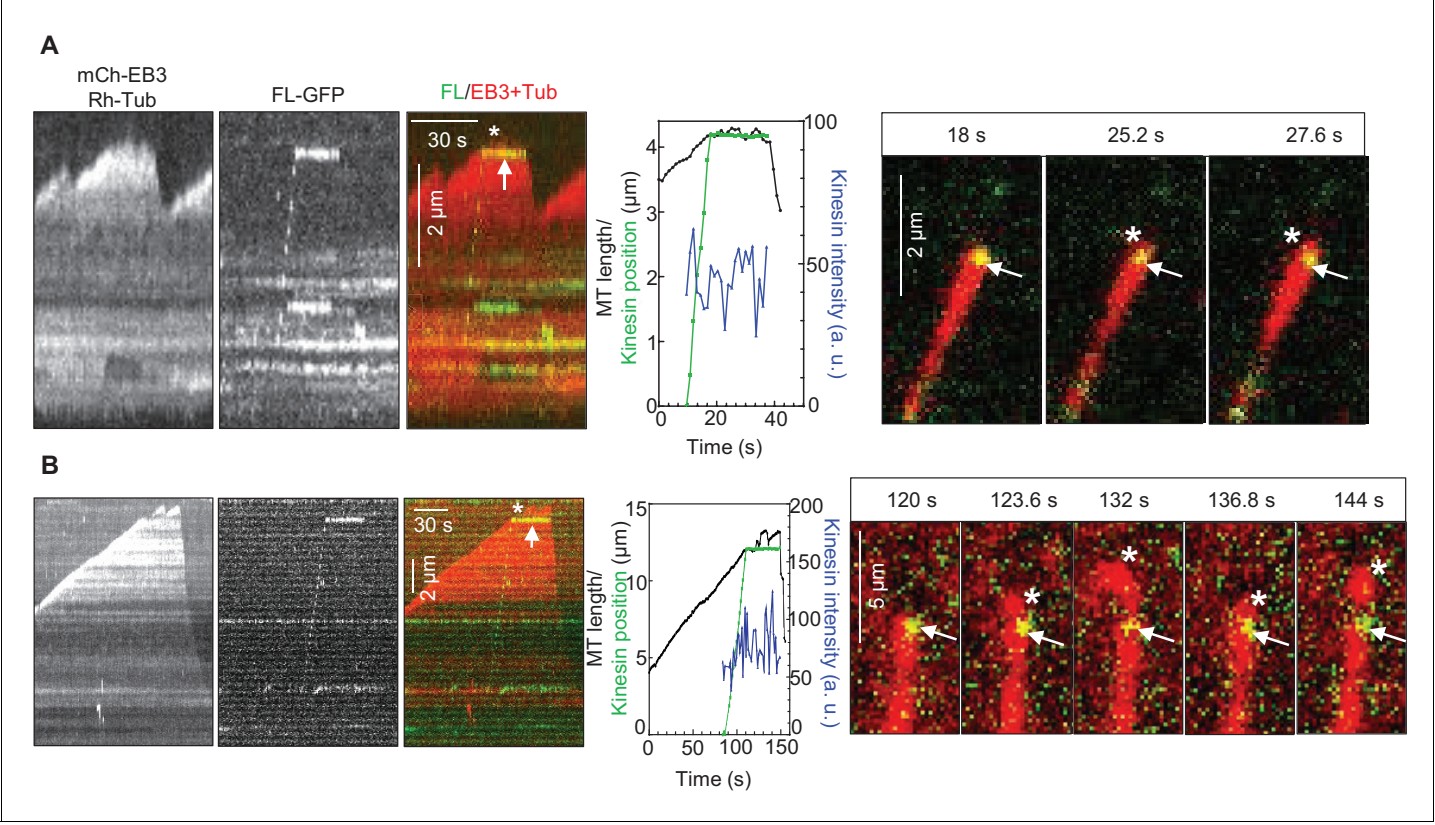

**Figure 4.** KIF21B molecules persisting on a MT tip can perturb MT growth. (A, B) Kymographs illustrating perturbation of MT growth in vitro by 0.5 nM KIF21B-FL-GFP in the presence of 15 µM tubulin with 3% Rhodamine-tubulin and 20 nM mCherry-EB3. Kymographs were generated from movies acquired as described for *Figure 3C*. Positions of the kinesins and MT tips together with the fluorescence intensity of the kinesins over time for the corresponding kymographs are also illustrated. Time lapse images on the right of the kymographs illustrate short excursions of MT plus tip (A) or curling of MT plus tip (B) after the binding of KIF21B-FL-GFP to the MT plus end. The position of the kinesin on the MT is indicated by arrows. Asterisks show the position of growing MT tips extending beyond the point of KIF21B attachment. See also Supplemental *Video 2*.

The following source data and figure supplement are available for figure 4:

**Source data 1.** An excel sheet with numerical data on the quantification of tracking of kinesins and MT tips over time represented as plots in *Figure 4A,B*.

**Figure supplement 1.** Perturbation of MT growth in vitro by full-length KIF21B

at 1.2 s/frame with an exposure time of 100 ms. For comparison, we collected data with the same laser power at a 12 times higher frame rate (100 ms/frame, stream acquisition), again in the presence of a 'reference chamber' with KIF5B-560 on the same coverslip. We observed several events of MT pause induction, in which the intensity of the KIF21B molecule inducing a pause was similar to that of single KIF5B-560 kinesins (*Figure 5F–H*, *Supplementary file 2*). For example, in the event shown in *Figure 5H*, the initial intensity of the analyzed KIF21B molecule when it starts its movement on the seed is in the range of the median of KIF5B-560 molecules (*Figure 5I*). While this molecule moves on the freshly polymerized MT lattice, its intensity is reduced by half, which we attribute to the bleaching of one of the GFP molecules. After arriving to the MT tip and inducing a pause, the molecule bleaches or desorbs as the MT switches to catastrophe. These data show that the laser power used for illumination was gentle enough for 20 s of imaging of a single KIF21B molecule at 10 frames per second. Measurements of the average photobleaching time showed that it was ~16 s at 100 ms/stream acquisition and ~197 s when the images were acquired with a 100 ms exposure with the interval of 1.2 s (*Figure 5—figure supplement 2*). This means that in our regular imaging experiments, the average bleaching time is close to 200 s and is thus significantly longer than the duration

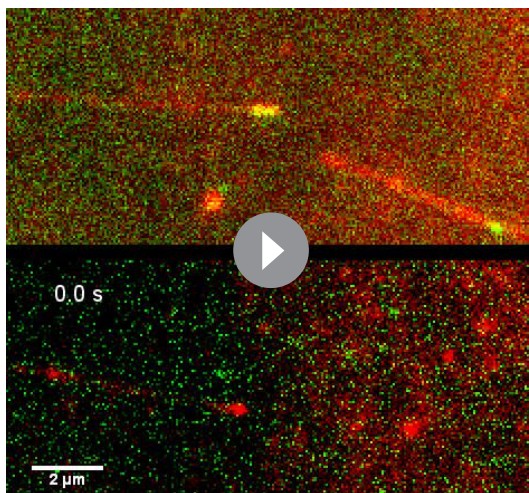

**Video 2.** KIF21B perturbs MT growth and induces MT bending. The combined movie shows the two different events illustrated in *Figure 4—figure supplement 1A* and *Figure 4B*. The movie shows bending of an MT growing beyond the point where KIF21B-FL-GFP was stalled (top panel, upper MT) and repeated short excursions from the point of KIF21B-FL-GFP stalling (bottom panel). The experiment was performed in the presence of 15 μM tubulin, Rhodamine-tubulin (0.5 μM), mCherry-EB3 (20 nM) and KIF21B-FL-GFP (0.5 nM). The movie consists of 128 frames acquired with a 1.2-s interval between frames and an exposure time of 100 ms. Scale bar, 2 μm.

of kinesin runs, which is typically tens of seconds. Photobleaching is thus unlikely to lead to a strong underestimate of the number of kinesins sufficient to trigger MT pausing.

## Characterization of MT-binding and autoinhibitory domains in the tail region of KIF21B

Our data presented so far indicate that a few KIF21B motors can prevent both growth and shortening by 'holding on' to a MT plus end. Since the dimeric motor domain of KIF21B by itself does not show such an activity, this result suggests that additional MT-binding sites that can associate with the MT plus ends must be present in the KIF21B tail. In line with this conclusion, we observed that when MTs were allowed to grow long in vitro in the presence of a low (0.5 nM) KIF21B concentration, KIF21B motors could pull a MT along another MT (*Figure 6—figure supplement 1*). This observation suggests that KIF21B can bind to one MT and walk along another MT at the same time.

We then set out to identify additional MT binding site(s) in the KIF21B tail by deletion mapping (*Figure 6A*). Different KIF21B fragments were expressed in COS-7 cells, and their colocalization with MTs was observed by fluorescence microscopy (*Figure 6B*, *Figure 6—figure supplement 2A*). We found that the tail of KIF21B alone strongly localized to MTs (*Figure 6A,B*). Subsequent mapping showed that two separate parts of the KIF21B tail could bind to MTs: the centrally located predicted coiled-coil part with adjacent sequences (CC2), as well as the C-terminal WD40 domain together with the N-terminal linker region enriched in proline, serine and arginine residues (termed L-WD40; *Figure 6A,B*, *Figure 6—figure supplements 2A* and *3*). Neither the WD40 domain alone, nor the linker alone showed robust MT binding, suggesting that the MT-binding affinity of this region depends on the combination of the two elements (*Figure 6A,B*, *Figure 6—figure supplement 2A*). Together, these data indicate that KIF21B can interact with MTs through three non-overlapping regions, the motor domain, the stalk region and the WD40 domain, and that the full length KIF21B molecule is likely to be folded when attached to MTs.

To test whether KIF21B assumes a compact conformation when bound to MTs, we used MT pelleting assays and electron microscopy with the isolated full-length KIF21B together with taxol-stabilized MTs (*Figure 6—figure supplement 2B*). As expected, after centrifugation, the full-length KIF21B was present in the pelleted fraction together with MTs. The pelleted fractions were further analyzed by negative stain electron microscopy. Electron micrographs of the full-length KIF21B bound to MTs indeed suggest that the motor has a highly folded, globular appearance, consistent with the presence of several MT interaction sites (*Figure 6C*, *Figure 6—figure supplement 2C–E*). Similar results were obtained with GMPCPP-stabilized MTs (*Figure 6—figure supplement 2F*).

Interestingly, the deletion of the linker region located N-terminally of the WD40 strongly reduced the MT binding activity of the resulting KIF21B mutant, while the deletion of the C-terminal WD40 domain rendered the KIF21B protein completely diffuse (KIF21B-MD-CC construct, *Figure 6A,D*). This was surprising, as the other two MT-binding sites, the motor domain and the CC2, were still present in these deletion mutants. Since a shorter KIF21B fragment, KIF21B-MD-CC1, bound to and moved along MTs, this result suggests that the CC2 region harbors not only a MT-binding, but also an autoregulatory activity.

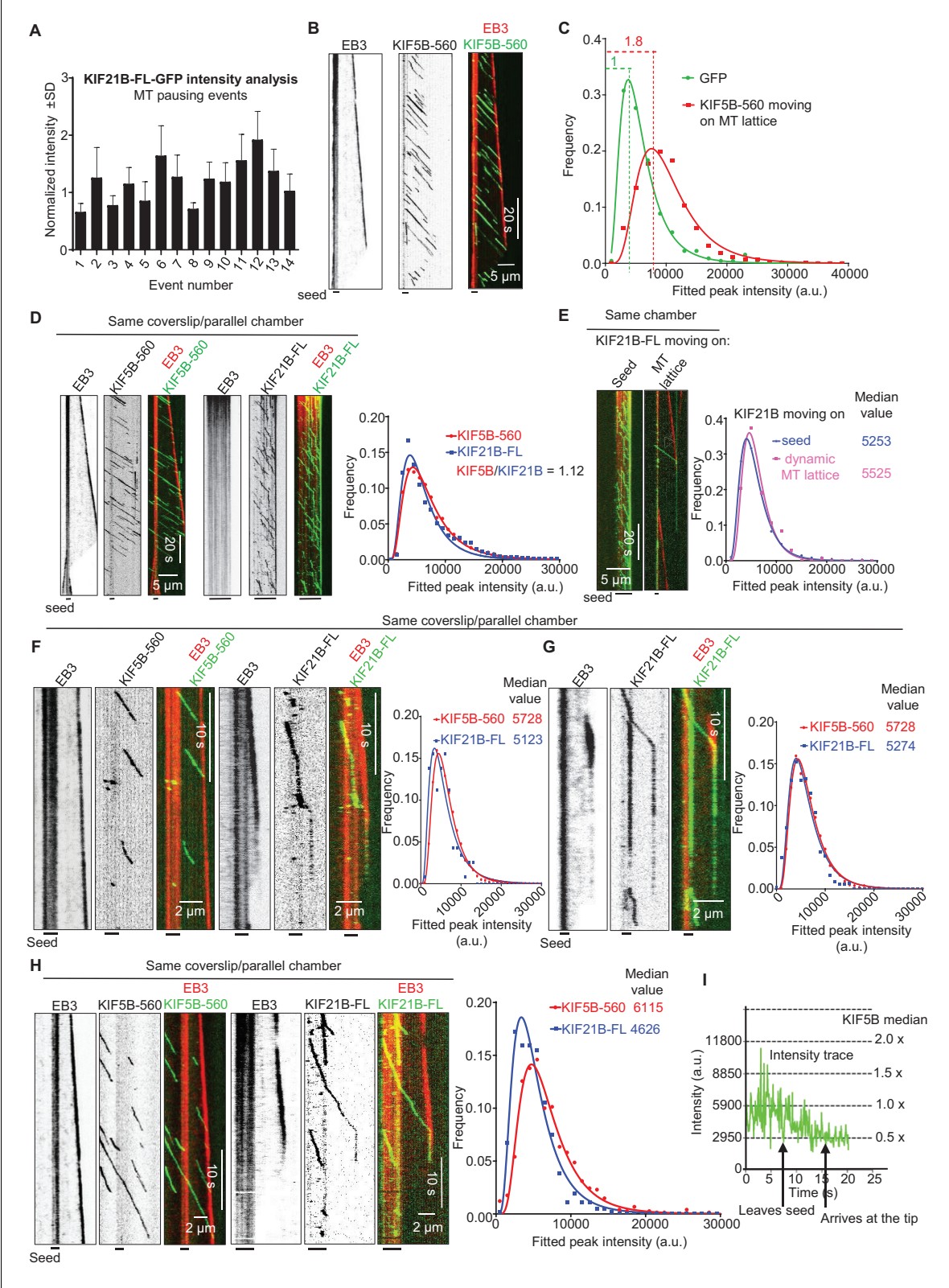

**Figure 5.** Single events of kinesin arrival to the MT plus end can induce MT pausing. (**A**) GFP intensity analysis of kinesins during MT pausing events. Values are normalized to the GFP intensity of proteins immobilized on the same coverslip in areas devoid of MTs. Data are from two independent experiments. (**B**) Kymographs showing the behavior of KIF5B-560-GFP in an in vitro reconstitution assay on dynamic MTs grown from Rhodamine-tubulin-labeled seeds in the presence of 15 μM tubulin, 20 nM mCherry-EB3 (red) and 5 nM KIF5B-560-GFP (green). (**C**) Histograms of fluorescence

*Figure 5 continued on next page*

*Figure 5 continued*

intensities of single GFP molecules immobilized on coverslips (initial moment of observation of single molecules) or KIF5B-560-GFP moving on MTs in a separate chamber on the same coverslip (symbols) and the corresponding fits with lognormal distributions (lines). n = 452 and 2040 molecules; fluorophore density was 0.05 and 0.09 $\mu m^{-2}$ for GFP and KIF5B-560-GFP proteins (for the latter, MT-containing regions were manually selected for analysis). Dashed lines show the corresponding relative median values. (D) Kymographs showing the behavior of 5 nM KIF5B-560-GFP (moving on dynamic MTs, green) and 0.5 nM KIF21B-FL-GFP (moving on seeds, green) in an in vitro reconstitution assay with MTs grown from Rhodamine-tubulin-labeled seeds in the presence of 15 $\mu$M tubulin and 20 nM mCherry-EB3 (red). Histograms illustrate fluorescence intensities of KIF5B-560-GFP moving on MTs and KIF21B-FL-GFP moving on seeds in separate chambers on the same coverslip (symbols) and the corresponding fits with lognormal distributions (lines). n = 5123 and 8728 molecules; fluorophore density was 0.15 and 0.18 $\mu m^{-2}$ for KIF5B-560-GFP and KIF21B-FL-GFP proteins (MT-containing regions were manually selected for analysis). Ratio of the corresponding median values is also indicated. (E) Kymographs illustrating KIF21B-FL-GFP (0.5 nM) movement on seeds and dynamic MTs, histograms of the corresponding fluorescence intensities measured within the same sample (symbols) and their fits with lognormal distributions (lines). Median values are also indicated. (F–H) Kymographs illustrating motility of KIF5B-560-GFP (5 nM) on dynamic MTs and KIF21B-FL-GFP (0.5 nM) moving from an MT seed to the dynamic MT lattice and inducing a MT pause in an in vitro reconstitution assay with MTs grown from Rhodamine-tubulin-labeled seeds in the presence of 15 $\mu$M tubulin and 20 nM mCherry-EB3 (red). Histograms show fluorescence intensities of motile KIF5B-560-GFP molecules and KIF21B-FL-GFP inducing a MT pause in separate chambers on the same coverslip (symbols) and the corresponding fits with lognormal distributions (lines). Median values are also indicated. In all the conditions, kymographs were generated from movies of 1500 frames (stream acquisition, exposure time 100 ms) using Photometrics Evolve 512 EMCCD camera (Roper Scientific). Positions of seeds in each kymograph are indicated. (I) Fitted peak intensity time trace for the trajectory of a moving KIF21B-FL-GFP molecule from the event shown in *Figure 5H*. Dashed lines correspond to the scaled values of median fluorescence fitted peak intensity of KIF5B-560-GFP molecules moving on dynamic MT in a parallel chamber on the same coverslip.

The following source data and figure supplements are available for figure 5:

**Source data 1.** An excel sheet with numerical data on the quantification of KIF21B-FL intensity during MT pausing events, KIF5B-560 dimer analysis and comparison of fluorescence intensities of KIF5B-560 with KIF21B-FL represented as plots in *Figure 5A,C,D–I*.

**Figure Supplement 2—Source data 1.** An excel sheet with numerical data on the quantification of photobleaching traces of KIF21B-FL-GFP represented as plots in *Figure 5—figure supplement 2*.

**Figure supplement 1.** Kymographs illustrating KIF21B-FL-GFP (0.5 nM) moving on seeds and dynamic MTs in in vitro reconstitution assays, histograms of the corresponding fluorescence intensities (symbols) and the corresponding fits with lognormal distributions (lines).

**Figure supplements 1—Source data 1.** An excel sheet with numerical data on the quantification of KIF21B-FL fluorescence intensities represented as plots in *Figure 5—figure supplement 1*.

**Figure supplement 2.** Characteristic photobleaching traces of KIF21B-FL-GFP under two different imaging conditions.

Previous work showed that the part of KIF21A that corresponds to the CC2 region of KIF21B contains an autoinhibitory element, which interacts with the motor domain, and that mutations in this region cause loss of autoinhibition and lead to CFEOM1 (*Cheng et al., 2014*; *van der Vaart et al., 2013*). Recently, we have characterized this interaction in detail and showed that it is mediated by a regulatory region that forms an intramolecular antiparallel coiled coil (*Bianchi et al., 2016*). This sequence region, including the CFEOM1-associated residues, is well conserved in KIF21B (rCC, amino acids 931–1010) (*Figure 6—figure supplement 3*), suggesting that it might have a similar autoinhibitory function. To test this possibility, we first assessed the secondary structure content and the stability of KIF21B rCC by performing circular dichroism (CD) spectroscopy experiments. The far-ultraviolet CD spectra recorded for the polypeptide chain fragments revealed a significant amount of $\alpha$-helical structure with distinct minima centered around 208 and 222 nm (*Figure 6—figure supplement 4A*, inset). The stability of KIF21B rCC was subsequently assessed by a thermal unfolding profile monitored by CD at 222 nm, which yielded a melting temperature of 43.6°C *Figure 6—figure supplement 4A*). To assess the oligomerization state of KIF21B rCC, we performed sedimentation velocity experiments (*Figure 6—figure supplement 4B*), which revealed a molecular weight of 11 kDa for the polypeptide chain fragment (calculated molecular weight of KIF21B rCC: 9.5 kDa). These biophysical results are consistent with KIF21B rCC forming an intramolecular antiparallel coiled coil in solution, very similar to the one we described for KIF21A (*Bianchi et al., 2016*).

In agreement with the expected autoinhibitory function of rCC, its deletion in the KIF21B-MD-CC restored MT binding activity and motility of this KIF21B fragment (*Figure 6A,D*, *Figure 6—figure*

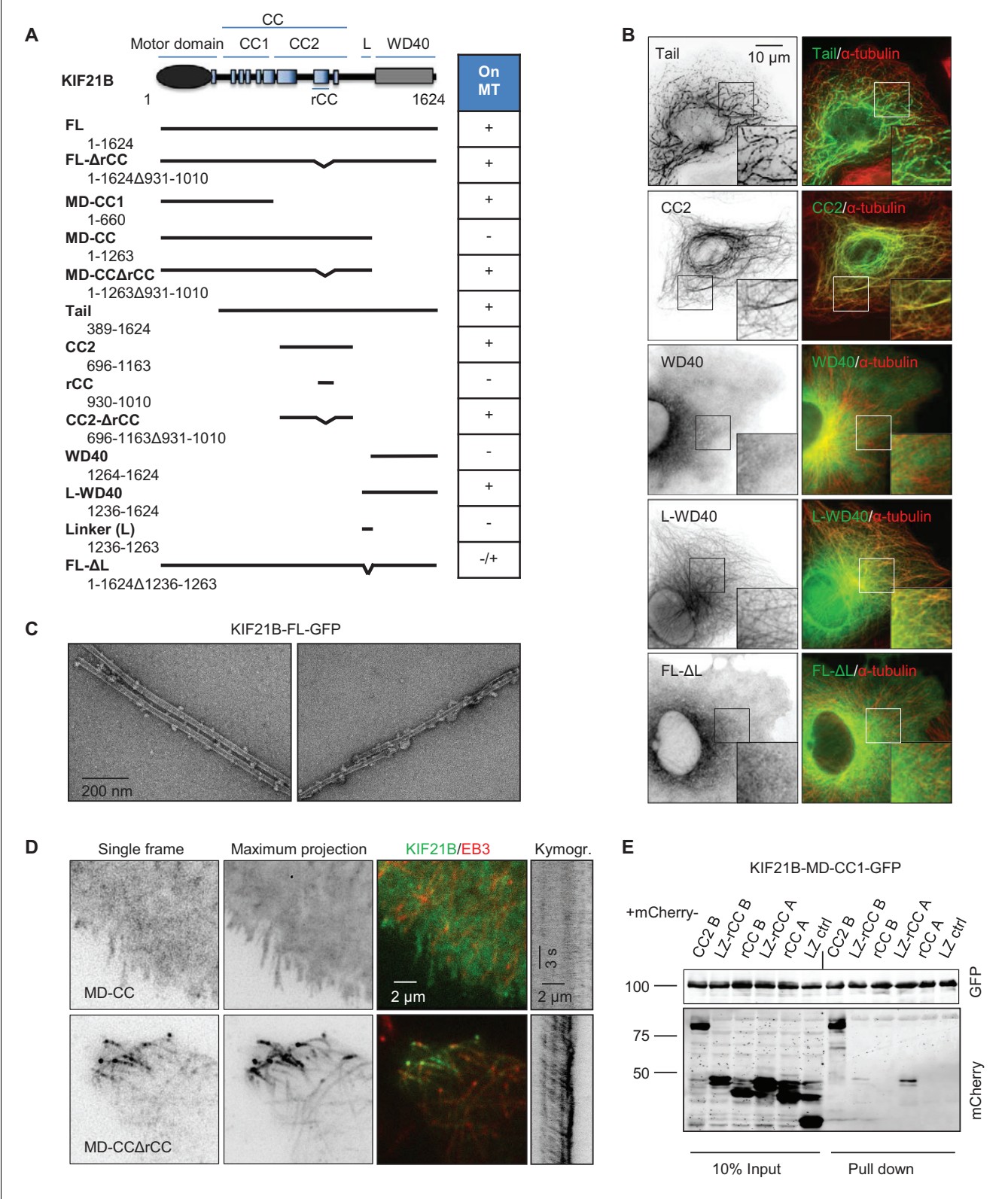

**Figure 6.** Mapping of the MT-binding domains in the tail of KIF21B. (**A**) Overview of deletion mutants used in this study. Colocalization of the GFP-tagged KIF21B deletion mutants with MTs in transiently transfected COS-7 cells is indicated. +, localization to MTs, -, diffuse distribution, -/+, diffuse in most cells, with occasional MT localization observed in some cells. (**B**) COS-7 cells were fixed one day after transient transfection with the indicated constructs and stained for α-tubulin. (**C**) Electron micrographs of negatively stained taxol-stabilized MTs in complex with KIF21B-FL-GFP. (**D**) Live

*Figure 6 continued on next page*

*Figure 6 continued*

imaging of COS-7 cells transiently transfected with KIF21B-MD-CC-GFP or MD-CCΔrCC-GFP and EB3-TagRFP-T. Represented are a single-frame, maximum intensity projection of 500 frames for the GFP channel, an overlay of single GFP frame in green and TagRFP-T in red and a kymograph along one of the EB3-labeled MTs showing kinesin motility. (E) Streptavidin pull down assay with the extracts of HEK293T cell expressing BirA, KIF21B-MD-CC1-GFP-TEV-Bio and the indicated mCherry-labeled proteins. A and B stand for KIF21A and KIF21B; LZ, leucine zipper from GCN4 used for dimerization. The other abbreviations are explained in panel **A**. The results were analyzed by Western blotting with the antibodies against the GFP- and mCherry.

The following source data and figure supplements are available for figure 6:

**Figure supplement 4—Source data 1.** An excel sheet with numerical data on the quantification of far-UV CD spectra (inset) and thermal unfolding profile of recombinant KIF21B rCC1 represented as plots in *Figure 6—figure supplement 4A*.
**Figure supplement 1.** In vitro reconstitution of MT growth in the presence of 20 nM mCherry-EB3, 3% Rhodamine-tubulin and 0.5 nM KIF21B-FL-GFP.
**Figure supplement 2.**
**Figure supplement 3.** Alignment of human KIF21A and KIF21B sequences.
**Figure supplement 4.**

*supplement 2A*). By itself, the rCC did not bind MTs in cells, and its deletion had no effect on the MT binding properties of the CC2 fragment (*Figure 6A*, *Figure 6—figure supplement 2A*). Using immunoprecipitation assays, we detected an interaction between the CC2 and the MD-CC1 region of KIF21B (*Figure 6E*). A weak binding was also observed with an rCC variant that was fused to the dimeric leucine zipper (LZ) of GCN4, although not with the monomeric version of rCC (*Figure 6E*). A stronger binding of the KIF21B motor domain was observed to the rCC region of KIF21A (*Figure 6E*), suggesting that the autoinhibitory interaction within KIF21B is attenuated compared to KIF21A. In agreement with this view, overexpressed KIF21A is largely diffuse in cells and presumably only becomes active when bound to appropriate partners (*van der Vaart et al., 2013*), while KIF21B shows constitutive MT association.

## Regulation of the MT pausing activity of KIF21B by its tail region

If the rCC does not fully inhibit the full-length KIF21B motor, what is the function of this region? To address this question, we have purified the KIF21B protein lacking the rCC (KIF21B-FL-ΔrCC), and a shorter version of this protein, which also lacked the WD40 domain (KIF21B-MD-CCΔrCC) (*Figure 2—figure supplement 1*). Mass spectrometry analysis demonstrated that the contaminants present in these two KIF21B preparations were essentially the same as in the isolated full-length KIF21B (*Supplementary file 1*). Analysis of fluorescence intensity and photobleaching confirmed that both deletion mutants are dimers, similar to the full-length molecule (*Figure 7A*, *Supplementary file 2*). In vitro assays showed that unlike the full-length protein, both kinesins lacking the rCC could land not only on GMPCPP-seeds but also on newly polymerized MT lattices (*Figure 7B*). Both full-length KIF21B and KIF21B-FL-ΔrCC exhibited slower motility on seeds compared to fresh GDP-MT lattices (*Figure 7C*). In contrast, the KIF21B-MD-CCΔrCC protein showed no reduced velocity on the MT lattice (*Figure 7B,C*), suggesting that the C-terminal WD40-containing region creates friction on GMPCPP-seeds. Consistent with this notion, we found that the L-WD40-GFP fragment displayed high preference for GMPCPP-seeds in vitro, both when present in cell extracts and in purified form, while the CC2 fragment showed no such preference (*Figure 7D*, *Figure 7—figure supplement 1A–C*). The preference for the GMPCPP seeds was not due to their attachment to glass or inclusion of biotinylated tubulin, because L-WD40-GFP showed no preference for taxol-stabilized MT seeds prepared in the same way as the GMPCPP seeds (*Figure 7D*, *Figure 7—figure supplement 1B,C*). However, we did not observe any accumulation of L-WD40-GFP at the growing MT plus ends, indicating that the preference for a specific nucleotide state is not sufficient to induce plus-end tracking of this protein fragment.

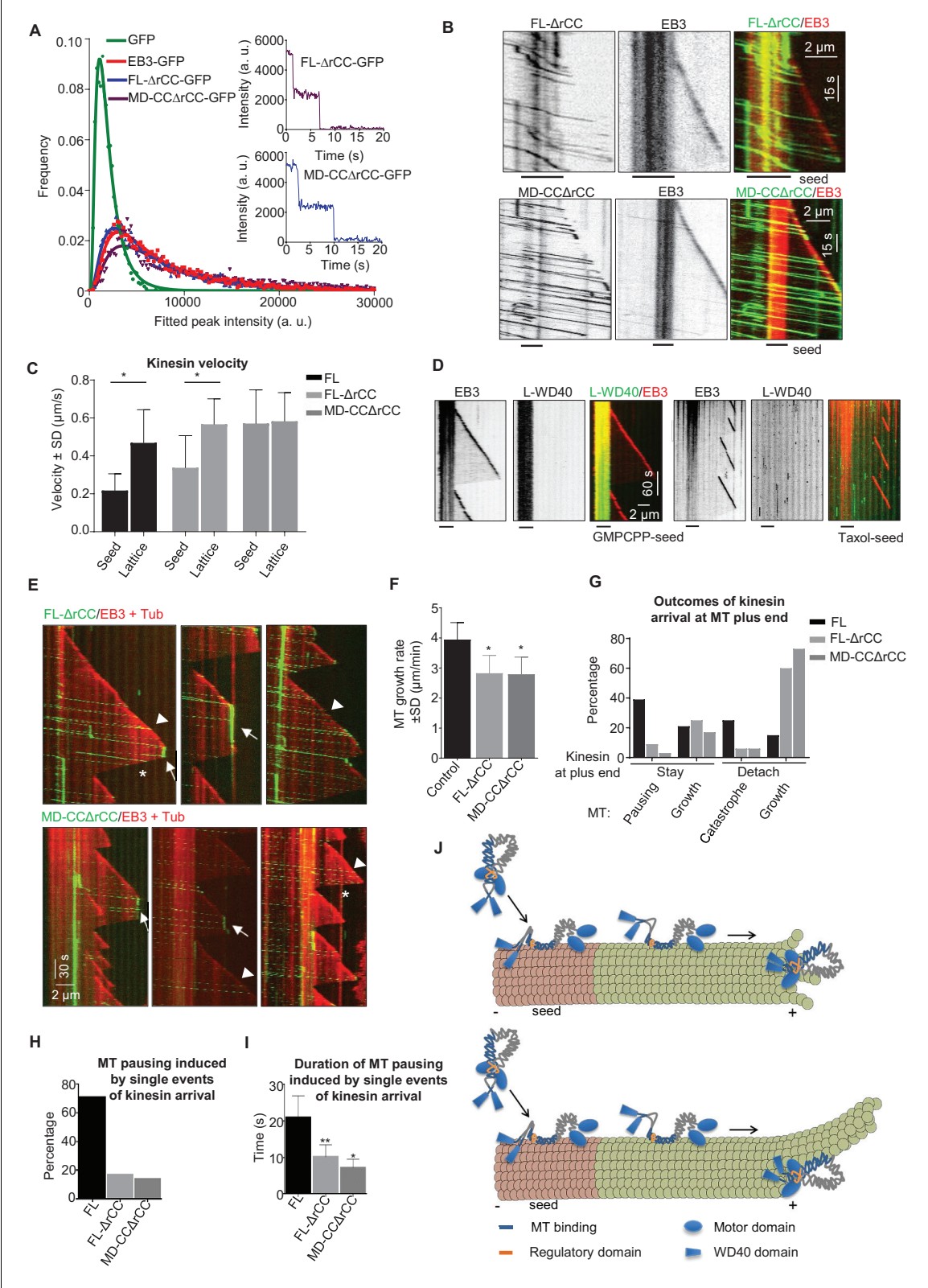

**Figure 7.** The WD40 domain and the autoinhibitory coiled coil region contribute to the pause-promoting activity of KIF21B. (**A**) Histograms of fluorescence intensities at the initial moment of observation of single molecules of the indicated proteins immobilized on coverslips (symbols) and the corresponding fits with lognormal distributions (lines). n = 3907, 5002, 6725 and 6943 molecules; fluorophore density was 0.19, 0.24, 0.30 and 0.33 $\mu m^{-2}$ for GFP, GFP-EB3, KIF21B-FL-ΔrCC-GFP and KIF21B-MD-CCΔrCC-GFP proteins. Insets show representative photobleaching traces of individual

*Figure 7 continued on next page*

*Figure 7 continued*

molecules (background subtracted). (**B**) Kymographs illustrating the behavior of the indicated deletion mutants of KIF21B at 3 nM concentration on dynamic MTs in the presence of 100 nM mCherry-EB3. GMPCPP-stabilized MT seeds were labeled with Rhodamine-tubulin (lines below kymographs). Kymographs were generated from the movies acquired using Photometrics Evolve 512 EMCCD (Roper Scientific) camera (stream acquisition with an exposure time of 500 ms). (**C**) Quantification of the velocity of KIF21B-FL-GFP and the deletion mutants on seeds and freshly polymerized MT lattices, shown in *Figures 3A* and *7B*. Seed: n = 295 for KIF21B-FL-GFP, n = 195 for KIF21B-FL-ΔrCC-GFP, n = 434 for KIF21B-MD-CCΔrCC-GFP; lattice: n = 131 for KIF21B-FL, n = 133 for KIF21B-FL-ΔrCC-GFP, n = 434 for KIF21B-MD-CCΔrCC-GFP. Data are from two or three independent experiments. Values significantly different from each other are indicated by asterisks, p<0.0001, Mann-Whitney U test. (**D**) Kymographs illustrating the interaction of purified GFP-L-WD40 (100 nM) with dynamic MTs grown from Rhodamine-tubulin labeled GMPCPP- or taxol-stabilized seeds (as indicated) in the presence of 20 nM mCherry-EB3. Kymographs were generated from the movies acquired in stream acquisition mode with an exposure time of 500 ms using Photometrics Evolve 512 EMCCD camera (Roper Scientific). (**E**) Kymographs illustrating the behavior of KIF21B deletion mutants on dynamic MTs in the presence of 20 nM mCherry-EB3 and 3% Rhodamine-tubulin. Pauses and KIF21B detachment from a depolymerizing MT end are indicated by arrows and asterisks, respectively. Arrowheads indicate kinesin detachment from the growing MT tip. Kymographs were generated from the movies acquired using CoolSNAP HQ2 CCD camera (Roper Scientific) with a 1.2-s interval between frames and an exposure time of 100 ms. (**F**) Quantification of MT growth rate in vitro in the presence of 15 µM tubulin with 20 nM mCherry-EB3 alone (n = 71) or together with 3 nM KIF21B-FL-ΔrCC-GFP (n = 79) or KIF21B-MD-CCΔrCC-GFP (n = 79). MTs were grown in the presence of 3% Rhodamine-tubulin. two independent experiments. (**G**) Quantification of different events observed after KIF21B-FL or its mutants reach a growing MT plus end. Data shown in *Figure 3E* are included here for comparison. n = 501 for KIF21B-FL-ΔrCC-GFP, n = 647 for KIF21B-MD-CCΔrCC-GFP. Data are from at least two independent experiments. (**H**) Percentage of pausing events induced by a single event of kinesin arrival from all detected pauses. Total number of pausing events: n = 51 for KIF21B-FL-GFP, n = 46 for KIF21B-FL-ΔrCC-GFP, n = 22 for KIF21B-MD-CCΔrCC-GFP. Data are from at least two independent experiments. (**I**) Quantification of the duration of MT pausing induced by a single kinesin arrival event at the growing MT plus end. n = 36 for KIF21B-FL-GFP, n = 8 for KIF21B-FL-ΔrCC-GFP, n = 3 for KIF21B-MD-CCΔrCC-GFP. Data are from at least two independent experiments.**p<0.0001, *p<0.0004 Mann-Whitney U test. (**J**) Model for the regulation of KIF21B motility and pause induction by the tail domain. In solution, KIF21B motor domains are inhibited by the regulatory region, while the WD40 domains are available for the interaction with MTs; WD40 domains show preference for the GMPCPP-stabilized seeds (red). After binding to seeds, KIF21B becomes activated and can walk to the plus end; it is likely that both the WD40 and the CC2 region contribute to MT binding. The kinesin can transfer from the seed to the freshly polymerized MT lattice; the interaction of the CC2 but not of the WD40 with the lattice promotes motor processivity. At the tip, the conversion to the autoinhibited conformation and the WD40 domain can prevent KIF21B from stepping off the MT plus end. This allows the motor to prevent both elongation and shortening of a small number of protofilaments with which it interacts. The remaining protofilaments might undergo short excursions of growth and shrinkage (upper panel); alternatively, they might elongate for some time and such an incomplete MT will be prone to bending and catastrophe (lower panel).

The following source data and figure supplements are available for figure 7:

**Source data 1.** An excel sheet with numerical data on the quantification of KIF21B mutants dimer analysis, photobleaching step analysis, velocities on seeds and MT lattices, MT growth rate in vitro and outcomes of the arrival of KIF21B mutants at MT plus ends, represented as plots in *Figure 7A,C,F–I*.
**Figure supplement 1.** Characterization of KIF21B tail fragments in vitro
**Figure supplements 1—source data 1.** An excel sheet with numerical data on the quantification of the intensity of KIF21B-L-WD40 on seeds and dynamic MTs represented as plot in *Figure 7—figure supplement 1B*.

We next examined the ability of the two KIF21B mutants to affect MT growth. Similar to the full-length molecule, both proteins showed a high degree of processivity, with run lengths of up to ~8–8.5 µm (*Figure 7E*). The two mutants had no effect on MT depolymerization, as they detached from shrinking MTs (asterisks in *Figure 7E*). Both mutants reduced MT growth rate, similar to the full-length KIF21B and the KIF21B-MD-CC1 mutant (*Figure 7F*). Furthermore, upon arrival to the growing MT plus end, the two mutants could cause MT pausing and induce catastrophes, again similar to what was observed with the full-length molecule (*Figure 7E,G*). However, both mutated motors were much less potent than full-length KIF21B, because in the majority of the cases (~60%), the mutated motors detached from the MT tip without pausing MT growth (*Figure 7G*). The pauses induced by the two mutant kinesins were typically due to the presence of multiple independently arriving motors (*Figure 7E,G,H*), and the duration of pauses induced by a single arrival event of either KIF21B-FL-ΔrCC or KIF21B-MD-CCΔrCC were significantly shorter than those triggered by the full-length KIF21B protein (*Figure 7I*). Taken together, these results suggest that both the regulatory rCC region and the C-terminal WD40-containing domain can contribute to the ability of KIF21B to stay attached to the growing MT plus end and to induce pausing (*Figure 7J*).

# Discussion

In this study, we showed that KIF21B is a highly processive MT plus-end directed motor, which can potently induce pausing of MT plus ends. This activity depends on several regions of this large motor protein. The N-terminally located motor domain is motile, thus ensuring protein accumulation at the MT plus ends, and similar to the motor domains of other kinesin-4 family members, it slows down MT polymerization (*Bieling et al., 2010*; *Bringmann et al., 2004*; *van der Vaart et al., 2013*). The dimeric version of the motor is sufficient to reduce MT growth, even though it does not accumulate at the MT tips. It is possible that KIF21B motors arriving at the MT tip somehow affect the conformation of terminal tubulin dimers, and in this way transiently perturb the structure of the polymerizing MT end. Our analysis of EB3 intensity in the presence of a low concentration of KIF21B-MD-CC1 supports this idea.

Importantly, in contrast to the full-length kinesin, the dimeric version of the motor domain alone does not show high processivity or the ability to stay attached to growing MT plus ends, thereby inducing their pausing. These properties are conferred by the stalk and the tail regions of the protein, which constitute two separate MT-binding sites. The presence of additional MT-binding domains is quite common in kinesins and has been established for kinesin-1, kinesin-5, kinesin-8 family members and CENP-E (*Gudimchuk et al., 2013*; *Navone et al., 1992*; *Stumpff et al., 2011*; *Su et al., 2011*; *van den Wildenberg et al., 2008*; *Weaver et al., 2011*). These domains are often basic polypeptide regions that can interact with the negatively charged surface of MTs. The distinguishing feature of KIF21B is the presence of a C-terminal WD40 domain involved in MT binding together with the adjacent positively charged linker region. The combined MT-binding activity of a folded domain augmented by a basic polypeptide region is reminiscent of that found in other MAPs such as EBs, CLIPs and the Ndc80 complex, in which globular calponin homology or CAP-Gly domains cooperate with positively charged linkers for MT binding (*Alushin et al., 2012*; *Hoogenraad et al., 2000*; *Komarova et al., 2009*). An interesting feature of the KIF21B C-terminus is its ability to distinguish between different types of MT lattices, as it binds much better to GMPCPP than to taxol-stabilized GDP-MTs. This property has an impact on the full-length protein, as the WD40 domain promotes the binding of KIF21B to GMPCPP seeds and slows down its motility on the seeds. Since GMPCPP-MTs are believed to mimic certain features of the GTP-MT lattice (*Alushin et al., 2014*), which is enriched at growing MT plus ends, it is tempting to speculate that in the context of the full-length protein this property helps to prevent kinesin detachment from the polymerizing plus ends. We should note, however, that the mechanistic basis of the preference of the L-WD40 fragment of KIF21B for the GMPCPP lattice is unclear, and it is not sufficient to confer MT plus-end tracking behavior. It is possible that at growing MT plus ends the number of binding sites for which the C-terminus of KIF21B would have preference or its affinity for these sites would be affected by protofilament curvature (*Brouhard and Rice, 2014*). Still, it could act in concert with other MT-binding domains of KIF21B to increase the bias for end-binding.

Another feature that helps to prevent the detachment of KIF21B from the MT plus end once it arrives at the tip is the presence of the autoregulatory region. Autoinhibition mechanisms that prevent motility of the cargo-unbound motors are common in different kinesins, including kinesin-1, kinesin-2 and the close KIF21B homologue KIF21A (*van der Vaart et al., 2013*; *Verhey and Hammond, 2009*). However, in contrast to other autoinhibited kinesins, which require an activating partner, KIF21B proteins can still interact with MTs through the binding of the WD40 domain-containing tail with MTs. In our in vitro assays, the WD40 region induces a strong preference of the motor for GMPCPP-MTs, suggesting that the motor domains are autoinhibited, while the WD40 domains are available for MT binding (*Figure 7J*). KIF21B with a deleted WD40 domain is fully autoinhibited, which indicates that the interaction between the motor and the rCC blocks the ability of both the motor and the MT-binding stalk (CC2 region) to interact with MTs.

Once the motor is loaded on an MT, the autoinhibition is relieved and the motor starts to walk. It is possible that both the WD40 and the MT-binding CC2 regions contribute to the processivity of KIF21B (*Figure 7J*). We observe frequent detachment of full-length motors at the border between the GMPCPP-seed and the GDP-MT lattice, suggesting that KIF21B might be switching back to an autoinhibited state. Such switching might be stimulated by the presence of MT lattice defects at the border between the seed and the freshly grown lattice. In contrast, motors lacking the rCC region land more easily on dynamic MT lattices and pass more frequently to such lattices from GMPCPP-

seeds. The motors that do move along the GDP-MT lattice are highly processive, most likely due to the MT-binding CC2 domain in the stalk region, because the KIF21B-FL-ΔrCC and KIF21B-MD-CCΔrCC behave very similarly. In contrast, the KIF21B-MD-CC1, which lacks the CC2 region, displays only short runs. The WD40 domain and the interaction between the motor domains and the autoregulatory rCC region become important once the kinesin reaches the growing MT plus end. KIF21B mutants lacking these regions often detach from the MT tip, while the full-length motor frequently persists and promotes pausing. It is possible that this behavior depends on the switching of the kinesin from the stepping mode to a conformation in which it attaches to the MT through its MT-binding tail domains (*Figure 7J*). As discussed above, in these conditions, the WD40 domain-containing C-terminus might help to recognize some structural feature of the plus end that is related to the presence of the GTP-cap.

It is striking that a few kinesins can induce a stable pause with an average duration of ~20 s, suggesting that different MT-binding domains within one molecule might interact with several protofilaments and prevent both their growth and depolymerization, thus inducing a pausing state. Noteworthy in this respect is our observation that when a MT continued growing after KIF21B was stalled, its growth was often strongly perturbed: we observed switching between growth and shortening episodes, as well as strong MT bending after the point of KIF21B attachment (*Figure 7J*). These data are reminiscent of our work on the effect of binding of a protofilament-blocking drug, Eribulin (*Doodhi et al., 2016*). We recently showed that attachment of a single Eribulin molecule, which, based on structural data can inhibit growth of only one MT protofilament, was sufficient to either cause a catastrophe or induce MT growth perturbation, suggesting elongation of an incomplete MT (*Doodhi et al., 2016*). It is tempting to speculate that similar to Eribulin, a KIF21B molecule stalled at the MT tip would be sufficient to block a small number of protofilaments, and this would result in inefficient elongation of the remaining protofilaments. Importantly, in contrast to Eribulin, KIF21B can also stabilize the protofilaments which it blocks, and therefore it is able to prevent, at least for some time, MT depolymerization. The resulting event often appears as a pause at the level of fluorescence microscopy, although the protofilaments that are not occluded by KIF21B are still likely to be dynamic (*Figure 7J*). The presence of multiple KIF21B molecules would result in blocking and stabilization of more protofilaments and thus more effective pausing, as we have observed. KIF21B-induced pausing events were typically followed by a catastrophe. This is in line with the slow retraction of the whole MT network observed in KIF21B-overexpressing cells (*Figure 1—figure supplement 1A*), which can be explained by the gradual loss of tubulin subunits from KIF21B-stabilized MT plus ends.

The effects induced by purified KIF21B in vitro are partly consistent with the recent analyses of MT plus-end dynamics in neuronal cells (*Ghiretti et al., 2016*; *Muhia et al., 2016*), in which a reduction of MT growth processivity was observed upon *Kif21b* knockout or depletion. This observation is in line with the idea that the presence of KIF21B induces either catastrophes or pausing, since both types of events would cause disappearance of an EB-positive comet. The effects on MT growth rate were opposite in the two studies: a decrease in MT growth rate was observed in knockout cells, while an increase was seen after RNA interference-mediated knockdown of KIF21B (*Ghiretti et al., 2016*; *Muhia et al., 2016*). These complexities might be due to some indirect effects on the tubulin pool or other MAPs, and are therefore difficult to compare to our in vitro analyses.

*Ghiretti et al. (2016)* also carried out in vitro experiments with purified KIF21B and its fragments. Similar to our study, they have identified a MT-binding domain in the stalk of the kinesin, but since MT binding was only investigated by co-pelleting assays with stabilized MTs, the WD40-containing C-terminal MT-binding region was not detected in these experiments, consistent with our observation that this domain does not bind to taxol-stabilized MTs. We note that the results of the analyses of the effect of KIF21B on MT dynamics were very different from ours, as the full length KIF21B, although motile in cells, did not seem to display a motile behavior in vitro even at 300 nM concentration and mostly associated with depolymerizing MT ends. However, it could still increase the polymerization rate and promote catastrophes of growing MT plus ends (*Ghiretti et al., 2016*). In contrast, in our experiments, 3–5 nM KIF21B was sufficient to block MT outgrowth from seeds at different tubulin concentrations. Furthermore, we did not observe KIF21B accumulation on depolymerizing MT ends, and the impact of KIF21B on MT growth (pausing, growth perturbation or catastrophe induction) was strongly associated with motile motors 'catching up' with growing MT plus ends. Finally, while in our experiments the KIF21B motor alone could slow down MT growth at

2–10 nM, Ghiretti et al. observed no effect of a similar construct even at 300 nM. We attribute these discrepancies to the differences in kinesin preparations and assay conditions (e.g. different ionic strength of the assay buffer, which seemed to be significantly lower in the experiments by Ghiretti et al. than in our study).

Our results suggest that in cells, KIF21B might use some additional factors for its loading onto MTs, and it is of course also possible that these or other factors would contribute to the association of KIF21B with MT plus ends. For example, the L-WD40 fragment of KIF21B fully decorates dynamic MTs in cells while it fails to do so in our in vitro assays, suggesting involvement of additional MAPs or post-translational modifications of tubulin. Further, an interesting implication of the observation that KIF21B is highly processive is that it is expected to display a stronger accumulation and thus a stronger pausing effect on the plus ends of longer MTs. Such a length-dependent effect would be similar to that described for other processive kinesins regulating MT plus-end dynamics (*Hibbel et al., 2015*; *Su et al., 2012*; *Varga et al., 2006*). It is possible that MT length-dependent regulation of pausing or catastrophe might help to achieve more uniform MT lengths in long neurites of neuronal cells, where MT growth is not bounded by the cell margin. In addition, since KIF21B can bind to one MT and step on another one, it might also play a role in organizing MT arrays by sliding MTs against each other. KIF21B is thus an interesting player in the cell's versatile toolbox responsible for MT-based transport and shaping of MT arrays. Changes in these arrays caused by alterations in KIF21B activity combined with its potential transport-related functions might explain the involvement of KIF21B in human diseases.

## Materials and methods

### DNA constructs, cell culture and transfection

We used previously described COS-7 cells (*van Bergeijk et al., 2015*) and HEK293T cells (*Bouchet et al., 2016*), which were cultured in DMEM/F10 (1/1 ratio, Lonza, Basel, Switzerland) supplemented with 10% fetal calf serum and penicillin and streptomycin. The cell lines were routinely checked for mycoplasma contamination using LT07-518 Mycoalert assay (Lonza). KIF21B expression constructs were made using human cDNA clone KIAA0449 (Kazusa DNA Research Institute, Japan) in pEGFP-N3, pEGFP-C1, mCherry-C1 or TagRFP-N3 vectors by PCR-based strategies. Additional TEV-protease recognition (ENLYFQG) and Biotinylation tag sequences (MASGLNDIFEAQKIE WHEGGG) were introduced in the EGFP vectors for protein purification purposes (as described previously, (*van der Vaart et al., 2013*)). Biotin ligase BirA expression construct (*Driegen et al., 2005*) was a gift from D. Meijer (University of Edinburgh, UK), EB3-TagRFP-T was described previously (*van der Vaart et al., 2013*) and TagRFP-α-tubulin was from Evrogen. Plasmids were transfected with polyethylenimine (PEI) or FuGene6 (Roche, Basel, Switzerland).

### Antibodies and cell fixation

Rabbit-anti-GFP (ab290, Abcam, Cambridge, UK), mouse-anti-mCherry (632543, Clontech, CA), rat-anti-α-tubulin (YL1/2) (MA1-80017, Pierce Antibodies, MA) were used on fixed cells and Western blotting. We used the following secondary antibodies: IRDye 800CW Goat anti-rabbit and anti-mouse (Li-Cor Biosciences, Lincoln, NE), Alexa-488 and Alexa-568 conjugated goat antibodies against rat IgG (Molecular Probes, Eugene, OR).

For tubulin staining, COS-7 cells were fixed with –20°C methanol for 10 min and subsequently fixed with 4% paraformaldehyde (PFA) in phosphate-buffered saline (PBS) for 15 min at RT. Cell membranes were permeabilized with 0.1% Triton X-100 in PBS and washed with 0.1% Tween-20 in PBS. Blocking and labeling were done in 0.1% Tween-20 in PBS supplemented with 1% bovine albumin serum. Slides were rinsed with 70% ethanol in the last wash step, air-dried and mounded in Vectashield mounting medium (Vector laboratories, Burlingame, CA).

For the nocodazole wash-out, cells were treated with 5 μM nocodazole for 2 hr, subsequently washed four times and re-incubated in normal culture medium at 37°C for indicated time points. Cells were fixed and stained as described above. All cell biological experiments were performed at least twice.

## Streptavidin pull down assays

Bio-GFP-tagged bait constructs and mCherry-tagged prey constructs were cotransfected in HEK293T cells. A construct encoding BirA was co-transfected to induce biotinylation of the Bio-tag. Cell lysates were prepared in 20 mM Tris pH7.5, 100 mM NaCl, 1% Triton-X100, 1x cOmplete protease inhibitor cocktail tablet (Roche) and incubated with M-280 Streptavidin Dynabeads (Invitrogen, CA) for 1 hr. Samples were washed three times in 20 mM Tris pH 7.5, 100 mM NaCl, 0.1% Triton-X100 and analyzed by SDS-PAGE and Western blotting.

## Protein purification from HEK293T cells

Constructs tagged with GFP-TEV-Bio were co-transfected with BirA in HEK293T cells as described for streptavidin pull down assays. Cell lysates were prepared in 50 mM Hepes pH 7.4, 300 mM NaCl, 1 mM $MgCl_2$, 0.5% Triton-X100, 1 mM DTT, 1x cOmplete protease inhibitor cocktail tablet (Roche) and incubated with M-280 Streptavidin Dynabeads (Invitrogen) for one hour. Samples were subsequently washed with 50 mM Hepes pH 7.4, 300 mM NaCl, 1 mM $MgCl_2$, 0.5% Triton-X100, 1 mM DTT three times and another three times with cleavage buffer (50 mM Hepes pH 7.4, 150 mM NaCl, 1 mM $MgCl_2$, 0.05% Triton-X100, 1 mM DTT, 1 mM EGTA), after which they were incubated in cleavage buffer supplemented with 40 ng/µl (770 nM) TEV protease (Sigma-Aldrich, St Louis, MO) for 2 hr at 4°C. Supernatant was collected and stored at −80°C prior to use. Purity of the samples was analyzed via SDS-PAGE and Coomassie staining. Concentrations of stock solutions varied between ~50–250 nM for full-length KIF21B-GFP, KIF21B-FLΔrCC-GFP and KIF21B-MD-CCΔrCC-GFP, and 0.6–1.2 µM for MD-CC1-GFP and GFP-L-WD40. Strep-tag-based KIF5B-560-GFP protein purification from HEK293T cells was done using Strep(II)-streptactin affinity purification method (*Sharma et al., 2016*).

## Recombinant protein production

cDNA encoding KIF21B rCC (residues 930–1010) were PCR amplified from a human cDNA library (*Frey et al., 2007*) and cloned into the pET-based bacterial expression vector PSTCm1 (*Olieric et al., 2010*). Subsequently, the protein was expressed in BL21 (DE3) at 37°C grown in LB media supplemented with a mixture of 50 µg/ml kanamycin and 30 µg/ml chloramphenicol to an $OD_{600}$ of 0.4–0.6. Expression was induced with 0.5 mM isopropyl 1-thio-$\beta$- galactopyranoside (IPTG; Sigma-Aldrich) and grown overnight at 20°C. Cell pellets were resuspended in lysis buffer (50 mM HEPES, pH 8, 500 mM NaCl, 10 mM Imidazole, 10% glycerol, 2 mM $\beta$-mercaptoethanol and 1 cOmplete EDTA-free protease inhibitor cocktail tablet (Roche) and lysed on ice by ultrasonication. Lysates were cleared by ultracentrifugation. Resulting supernatants were subsequently filtered (0.45 µm filter). The protein was affinity purified by IMAC on a 5 ml HisTrap FF Crude column (GE Healthcare, Chicago, Illinois) according to manufacturer's instructions. The 6xHis tag was cleaved using 2 units of human thrombin (Sigma-Aldrich) per milligram of recombinant protein and cleavage was performed over night at 4°C by dialysis in thrombin cleavage buffer (20 mM Tris-HCl pH 7.4, 150 mM NaCl, 2.5 mM $CaCl_2$ and 2 mM DTT). The 6xHis tag was separated from the target protein by re-application to the IMAC column. The processed protein was concentrated and further purified by size exclusion chromatography on a HiLoad Superdex 75 16/60 size-exclusion column (GE Healthcare) equilibrated in 20 mM Tris-HCl, pH 7.5, supplemented with 150 mM NaCl and 2 mM DTT.

## Biophysical characterization of the rCC fragment of KIF21B

CD spectra were recorded at 5°C and at a protein concentration of 0.166 mg/ml in PBS supplemented with 0.5 mM TCEP using a Chirascan spectropolarimeter (Applied Photophysics Ltd, Leatherhead, UK) and a cuvette of 0.1 cm path length. Thermal unfolding profiles between 5°C and 90°C were recorded by increasing the temperature at a ramping rate of 1°C/min monitoring the CD signal at 222 nm. Midpoints of thermal unfolding were calculated using the Glob3 program (Applied Photophysics).

Sedimentation velocity experiments were performed at 20°C and 42,000 rpm in Tris-HCl, pH 7.5, 150 mM NaCl, 2 mM DTT using a Beckman XLI analytical ultracentrifuge (Beckman Coulter Inc., CA). Sedimentation profiles were recorded by UV absorbance (280 nm) and interference scanning optics. The partial specific volume of the samples as well as the density and viscosity of the buffer were calculated with SEDNTERP (http://sednterp.unh.edu/). Data were fitted with SEDFIT (*Schuck, 2000*)

using the continuous distribution model. Graphical representations were processed with GUSSI (bio-physics.swmed.edu/MBR/software.html).

## In vitro analysis of MT dynamics

In vitro assays were performed as described previously (*van der Vaart et al., 2013*). MT seeds were grown using 20 µM tubulin mix containing 18% biotin-tubulin and 12% Rhodamine- or HiLyte Fluor 488-tubulin (Cytoskeleton, Inc., Denver, CO) and 1 mM GMPCPP by polymerization at 37°C for 30 min, pelleting by centrifugation in an Airfuge for 5 min and depolymerization on ice. After a subsequent round of polymerization and pelleting, seeds were stored in MRB80 buffer (80 mM K-PIPES, pH 6.8, 4 mM MgCl$_2$, 1 mM EGTA) with 10% glycerol. Flow chambers were made with microscopy slides and plasma-cleaned glass coverslips. Coating was done with 0.2 mg/ml PLL-PEG-biotin (Surface Solutions, Dübendorf, Switzerland) in MRB80 buffer and 0.8 mg/ml NeutrAvidin for 5 min each. The seeds were attached to the coverslips via biotin-NeutrAvidin links and blocked with 0.8 mg/ml κ-casein. Reaction mixtures consisting of MRB80 supplemented with different concentrations of tubulin (indicated in figure legends), containing 3% Rhodamine-tubulin when indicated, 50 mM KCl, 0.1% methylcellulose, 0.5 mg/ml κ-casein, 1 mM GTP, oxygen scavenging system (20 mM glucose, 200 µg/ml catalase, 400 µg/ml glucose-oxidase, 4 mM DTT), 2 mM ATP, 20 or 100 nM mCherry-EB3 when indicated and the specified concentration of purified GFP or KIF21B-GFP proteins (stored in 50 mM Hepes pH 7.4, 150 mM NaCl, 1 mM MgCl$_2$, 0.05% Triton-X100, 1 mM DTT, 1 mM EGTA, 770 nM TEV protease and diluted by at least 10 or more times in MRB80 buffer for in vitro assays) or KIF5B-560 (stored in 50 mM Hepes pH 7.4, 150 mM NaCl, 1 mM MgCl$_2$, 0.05% Triton-X100, 1 mM DTT, 1 mM EGTA, 2.5 mM d-Desthiobiotin and diluted 20 times in MRB80 buffer for in vitro assays) were added to the flow chambers. Movies were collected using TIRF microscopy. For mCherry-CC2 and mCherry-L-WD40, extracts of HEK293T cells expressing the proteins, prepared in MBR80 supplemented with 1x cOmplete protease inhibitor cocktail tablet (Roche) and 1% Triton-X100, were used in the reaction mixture in a ratio of 1:4. All samples were incubated at 30°C during imaging. The quantitative data reported for each experiment were collected in at least two or more independent assays.

## Image acquisition and processing

Fixed cells were imaged with a Nikon Eclipse 80i upright fluorescence microscope equipped with Plan Apo VC N.A. 1.40 oil 100x and 60x objectives, or Nikon Eclipse Ni-E upright fluorescence microscope equipped with Plan Apo Lambda 100x N.A. 1.45 oil and 60x N.A. 1.40 oil objectives microscopes, Chroma ET-BFP2, - GFP or -mCherry filters and Photometrics CoolSNAP HQ2 CCD (Roper Scientific, Trenton, NJ) camera. The microscopes were controlled by Nikon NIS Br software.

Live cell imaging and in vitro assays were performed on an inverted research microscope Nikon Eclipse Ti-E (Nikon) with the perfect focus system (PFS) (Nikon), equipped with Nikon CFI Apo TIRF 100 × 1.49 N.A. oil objective (Nikon, Tokyo, Japan), Photometrics Evolve 512 EMCCD (Roper Scientific) and Photometrics CoolSNAP HQ2 CCD (Roper Scientific) and controlled with MetaMorph 7.7 software (Molecular Devices, CA). The microscope was equipped with TIRF-E motorized TIRF illuminator modified by Roper Scientific France/PICT-IBiSA, Institut Curie, and an ET-GFP filter set (Chroma, Bellow Falls, VT) for imaging of GFP-tagged proteins. For simultaneous imaging of green and red fluorescence we used triple-band TIRF polychroic ZT405/488/561rpc (Chroma) and triple-band laser emission filter ZET405/488/561m (Chroma), mounted in the metal cube (Chroma, 91032) together with Optosplit III beamsplitter (Cairn Research Ltd, Faversham, UK) equipped with double emission filter cube configured with ET525/50m, ET630/75m and T585LPXR (Chroma).

Long-term imaging was performed on an inverted research microscope Nikon Ti equipped with Plan Fluor 40x/1.30 Oil DIC and Plan Apochromat 20 × 0.75 Phase Contrast objectives, ET-GFP (49002) and ET-mCherry (49008) filters (Chroma) and controlled with MicroManager.

Cells were kept at 37°C, and in vitro samples at 30°C in a Tokai Hit INUBG2E-ZILCS Stage Top Incubator. Images and movies were processed using ImageJ. All images were modified by adjustments of brightness and contrast; smooth and sharp masks were applied in some cases. Maximum intensity projections were made using z projection. MT growth rates and kinesin velocities were obtained from kymograph analysis, using ImageJ plugin KymoResliceWide v.0.4 https://github.com/ekatrukha/KymoResliceWide (*Katrukha, 2015*); copy archived at https://github.com/elifesciences-

publications/KymoResliceWide). Results were plotted in Graphpad Prism 6. Statistical analysis was performed using non-parametric Mann-Whitney U-test.

## Tracking and single molecule intensity analysis

To build single molecule fluorescence histograms (*Figure 2A*, *Figure 3—figure supplements 1A*), purified GFP or GFP-fusion proteins were diluted in phosphate buffered saline (PBS) and added to the different imaging flow chambers of the same plasma cleaned coverslips. Chambers were subsequently washed with PBS, leaving a fraction of the GFP-tagged proteins immobilized on the coverslip. After sealing with vacuum grease to prevent evaporation, samples were imaged at room temperature using TIRF microscopy. Protein dilution was optimized to provide images of 0.1–0.4 fluorophores per $\mu m^2$ for each condition. At least 20 images were acquired at different positions on the coverslip to avoid pre-bleaching. ImageJ plugin DoM_Utrecht v.0.9.1 https://github.com/ekatrukha/DoM_Utrecht (*Katrukha et al., 2016*); copy archived at https://github.com/elifesciences-publications/DoM_Utrecht) was used for detection and fitting of single molecule fluorescent spots as described previously (*Yau et al., 2014*).. In short, individual spots were fitted with 2D Gaussian and the amplitude of the fitted Gaussian function was used as a measure of the fluorescence intensity value of an individual spot. The same parameter was used to build histograms in *Figure 2A*, *Figure 3—figure supplements 1A* and 7A. The histograms were fitted to lognormal distributions using GraphPad Prism 6.

To estimate the number of GFP molecules per kinesin (*Figure 5B–H*), KIF5B-560 and KIF21B-FL moving along the lattice of MTs in the different imaging flow chambers of the same plasma cleaned coverslips were analyzed and compared. ImageJ plugin DoM_Utrecht v.0.9.1 was used for detection and fitting of single molecule fluorescent spots. The histograms were fitted to lognormal distributions using GraphPad Prism 6. For MT pausing events induced by kinesin (*Figure 5A*), particles on the MT lattice and tip were detected using ComDet v.0.3.5 https://github.com/ekatrukha/ComDet (*Katrukha, 2016*); copy archived at https://github.com/elifesciences-publications/ComDet) and DoM_Utrecht v.0.9.1 (*Katrukha et al., 2016*) ImageJ plugins. Their single molecule intensity values were normalized to the average intensity of GFP-kinesins non-specifically attached to the coverslip in areas devoid of MTs.

Kinesin intensity and position for *Figure 3C*, *Figure 3—figure supplement 2*, *Figure 4A,B* were calculated using the same plugins. The position of MT tip was estimated from fitting of each *x*-profile of corresponding kymograph with error function with offset using custom written Matlab script.

All mentioned ImageJ plugins have source code available and are licensed under open-source GNU GPL v3 license.

To monitor the decay of fluorescence caused by photobleaching, single particles of KIF21B-FL-GFP were immobilized on the surface of a coverslip. Movies were recorded at 100 ms/stream acquisition for 100s or at 100 ms/1 frame per 1.2 s for 720s with low laser power (same laser power used for imaging in *Figures 3–5* and *7*) to allow KIF21B-FL-GFP to be photobleached. GraphPad Prism 6 software was used for the data fitting.

## MT pelleting and electron microscopy

Taxol- or GMPCPP-stabilized MTs were polymerized to a final concentration of 10 µM as described (*Kevenaar et al., 2016*). Afterwards, 50 µl of 0.19 µM of HEK293T purified full length KIF21B was incubated together with 0.45 µM stabilized MTs for 10 min at room temperature. As a control, the same amount of MTs was incubated separately. A taxol-glycerol cushion containing 55% 2x BRB80 buffer (80 mM K-PIPES, pH 6.8, 1 mM EGTA, 1 mM $MgCl_2$), 44% glycerol and 6% 2 mM paclitaxel was added to the centrifugation tubes prior to sample addition. After centrifugation of the samples at 174,500 x g for 10 min, the supernatants were carefully removed and the pellets were resuspended in 50 µl BRB80 buffer. Twenty microliters of each supernatant and pellet were mixed with 5 µl 5x SDS loading dye and analyzed on Coomassie stained 7.5% SDS-PAGE.

For negative staining electron microscopy, 5 µl aliquots of pellet samples prepared in the presence of either 1 µM taxol or 1 µM GMPCPP were transferred to freshly UV activated homemade carbon-coated copper grids. After 20 s of incubation, excess liquid was removed by side-blotting and the grids were washed twice with BRB80 buffer supplemented with either 1 µM taxol or 1 µM GMPCPP and once with double distilled water. Subsequently, the grid was stained three times with

a freshly prepared uranyl acetate solution. Electron micrographs were taken at a nominal magnification of 40 k with a JEM2200FS (JEOL, Peabody, MA) electron microscope operated at 200 kV and equipped with a TVIPS F416 camera.

## Mass spectrometry

Samples of purified proteins were ran on SDS-PAGE gel (150 ng FL, 30 ng FL$\Delta$rCC, 45 ng MD-CC$\Delta$rCC, 255 ng L-WD40, 150 ng MD-CC1). After in-gel digestion, samples were resuspended in 10% formic acid (FA)/5% DMSO and were analyzed with an Agilent 1290 Infinity (Agilent Technologies, CA) LC, operating in reverse-phase (C18) mode, coupled to a TripleTOF 5600 (AB Sciex, Nieuwerkerk aan de IJssel, Netherlands). MS spectra (350–1250 m/z) were acquired in high-resolution mode (R > 30,000), whereas MS2 in high-sensitivity mode (R > 15 000).

Raw files were processed using Proteome Discoverer 1.4 (version 1.4.0.288, Thermo Scientific, Bremen, Germany). The database search was performed using Mascot (version 2.4.1, Matrix Science, UK) against a Swiss-Prot database (taxonomy human). Carbamidomethylation of cysteines was set as a fixed modification and oxidation of methionine was set as a variable modification. Trypsin was specified as enzyme and up to two miss cleavages were allowed. Data filtering was performed using percolator, resulting in 1% false discovery rate (FDR). Additional filters were; search engine rank 1 peptides and ion score >20.

## Acknowledgements

We thank Dr. D Meijer (University of Edinburgh, UK) for the gift of BirA construct. This work was supported by the Netherlands Organization for Scientific Research grants: an ALW NWO VICI grant 865.08.002 to AA, an ALW NWO VICI grant 865.10.010 to CCH, a CW ECHO grant 711.011.005 to AA and AJRH, and as part of the National Roadmap Large-scale Research Facilities of the Netherlands (project number 184.032.201). The work was also supported by an ERC Synergy grant 609822 to AA and a grant of the Swiss National Science Foundation (310030B_138659 and 31003A_166608) to MOS.

## Additional information

### Funding

| Funder | Grant reference number | Author |
| --- | --- | --- |
| Nederlandse Organisatie voor Wetenschappelijk Onderzoek | 865.08.002 | Anna Akhmanova |
| European Research Council | 609822 | Anna Akhmanova |
| Schweizerischer Nationalfonds zur Förderung der Wissenschaftlichen Forschung | 310030B_138659 | Michel O Steinmetz |
| Nederlandse Organisatie voor Wetenschappelijk Onderzoek | 865.10.010 | Casper C Hoogenraad |
| Nederlandse Organisatie voor Wetenschappelijk Onderzoek | 711.011.005 | Albert JR Heck Anna Akhmanova |
| Nederlandse Organisatie voor Wetenschappelijk Onderzoek | 184.032.201 | Albert JR Heck |
| Schweizerischer Nationalfonds zur Förderung der Wissenschaftlichen Forschung | 31003A_166608 | Michel O Steinmetz |

The funders had no role in study design, data collection and interpretation, or the decision to submit the work for publication.

### Author contributions

WEvR, Conceptualization, Data curation, Formal analysis, Validation, Investigation, Visualization, Methodology, Writing—original draft; AR, Conceptualization, Data curation, Formal analysis,

Validation, Investigation, Visualization, Methodology, Writing—review and editing; SB, Conceptualization, Data curation, Validation, Investigation, Visualization, Methodology, Writing—original draft, Writing—review and editing; EAK, Data curation, Formal analysis, Validation, Visualization, Methodology, Writing—original draft, Writing—review and editing; QL, Data curation, Formal analysis, Investigation, Writing—review and editing; AJRH, Formal analysis, Funding acquisition, Methodology, Writing—review and editing; CCH, Funding acquisition, Methodology, Writing—original draft, Project administration; MOS, Formal analysis, Funding acquisition, Methodology, Writing—original draft, Project administration, Writing—review and editing; LCK, Conceptualization, Formal analysis, Supervision, Methodology, Writing—original draft, Writing—review and editing; AA, Conceptualization, Supervision, Funding acquisition, Writing—original draft, Project administration, Writing—review and editing

### Author ORCIDs
Ankit Rai, http://orcid.org/0000-0002-1569-0919
Albert JR Heck, http://orcid.org/0000-0002-2405-4404
Lukas C Kapitein, http://orcid.org/0000-0001-9418-6739
Anna Akhmanova, http://orcid.org/0000-0002-9048-8614

## Additional files

**Supplementary files**

• Supplementary file 1. Analysis of purified KIF21B and its deletion mutants used in this study by mass spectrometry. Samples of purified KIF21B proteins were loaded on SDS-PAGE, isolated from the gel after in-gel digestion and subsequently analyzed by mass spectrometry to test for purity. All identified proteins are included in *Supplementary file 1*in alphabetical order. Indicated are the molecular weight and the number of unique peptides found for identified proteins in the different KIF21B samples. In total, 121 proteins were identified for KIF21B-FL-GFP, 183 for KIF21B-FL-ΔrCC-GFP, 107 for KIF21B-MD-CCΔrCC-GFP, 92 for GFP-L-WD40 and 63 for KIF21B-MD-CC1-GFP.

• Supplementary file 2. Lognormal (best fit) values for the fluorescence intensity measurements.

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
