## [Decision Letter]

Thank you for submitting your article "Kinesin-4 KIF21B is a potent microtubule pausing factor" for consideration by *eLife*. Your article has been reviewed by three peer reviewers, and the evaluation has been overseen by a Reviewing Editor and Vivek Malhotra as the Senior Editor. The reviewers have opted to remain anonymous.

The reviewers have discussed the reviews with one another and the Reviewing Editor has drafted this decision to help you prepare a revised submission.

Summary:

A sizeable body of literature suggests that a defining feature of kinesin-4 motor proteins is that they slow (or halt) microtubule growth. In this manuscript, van Riel and colleagues show that KIF21B, an uncharacterized kinesin-4 closely related to KIF21A, also induces microtubule pausing. This activity is important from the standpoint of kinesin biochemistry and human disease, as KIF21B is associated with several human neurological disorders. The authors also demonstrate that KIF21B is highly processive, and that two additional microtubule binding sites are present within the motor. One of these microtubule binding sites resides within the C-terminal WD40 domain, and is particularly interesting as it appears to bind GMPCPP microtubules specifically. The implication of this finding is that the WD40 domain may help to anchor the motor at growing microtubule plus-ends, thus enhancing its pause-inducing activity. The authors go on to suggest that single KIF21B molecules are able to pause microtubule growth.

Essential revisions:

All three reviewers raised major concerns that must be addressed in order to consider publication in *eLife*.

1) The authors' conclusions for Figure 1 are very puzzling. Although the quality of the kymographs in Figure 1 is not high, individual particles of MD-CC1 can be seen arriving at the microtubule tip occasionally and immediately detaching. The fraction of time that the tip is in contact with one such molecule is very small relative to the total time of microtubule growth, while during this time the microtubule end polymerizes smoothly and continuously at a rate slower than in the control. The authors' conclusion that the slowing of growth velocity is due to these infrequent encounters seems improbable because this would imply a long-distance effect (e.g. the motor walks on the wall and this causes the tip to polymerize slowly) or some kind of memory (e.g. the motor did something to the tip and even after the motor detaches the rate of assembly remains slower than normal). Other explanations are possible. For example, are there any differences in the buffer conditions? The authors did not provide the final concentrations for buffers in which these observations were made, but the methods section suggests that the experiments with MD-CC1 were done in the presence of some additional reagents (e.g. detergent) and TEV protease, while EB3 control experiments lacked these additions. The authors should rule out this trivial explanation by adding GFP protein purified analogously to MD-CC1 to their EB3 control. If it really appears that microtubule growth is perturbed even when no MD-CC1 molecules are seen at the tip, the authors should add a discussion about possible molecular mechanisms for this very striking observation. A similar conclusion was made for the full length motor, but again microtubule polymerization appears to have been slowed when only a few motor molecules were present at the tip.

Related to this concern, the microtubule growth rate (~4 μm/ min at 15 μm tubulin) shown in Figure 1 is quite fast compared to other reports in the literature. Please provide a complete plot of microtubule growth rate vs. tubulin concentration, with and without KIF21B as well as microtubule growth rate vs KIF21B concentration.

2) That single molecules of KIF21B may be sufficient to cause microtubules to cease growing and shortening is fascinating. The main evidence is Figure 2, where a kymograph shows an alleged single molecule hitting the end and causing it to pause. However, the authors do not pursue this phenomenon at a mechanistic level, or provide a reasonable discussion that explains the observation. Both should be addressed in a revised manuscript and the following points must be taken into account:

A) What is the positional resolution of that experiment? Has the microtubule really paused, or has growth simply dropped to a level that is below detection?

B) It is suggested that the affinity of the WD40 domain for GMPCPP tubulin might stabilize motor-tip interactions, but this hypothesis still cannot explain an enzyme activity that probably involves more than one protofilament. A discussion of motor dimensions may help. The WD40 domain, for example, is 4 nanometers in diameter. What is the predicted length of the stalk? Can a single KIF21B motor crosslink multiple neighboring protofilaments? A minor issue related to this point concerns the purity of the KIF21B preparation (Figure 1—figure supplement 2). What is the stoichiometric band that runs at ~65 kDa? Are the authors sure that this protein is irrelevant for the observed KIF21B effect on microtubule dynamics?

C) The FL motor kymographs in Figure 2 are difficult to see, but almost all of the visible events in panel D are consistent with "stay on MT lattice" behavior, while the authors claim that the majority of events (~45%) are pausing of MT growth. The pausing is very striking and compelling for high motor decoration, but is the same phenotype observed for the dim complexes, which might be single molecules? For example, in panel D, the first 2 events in the upper kymograph have dim signals and are clearly "lattice" events. The third event with a somewhat brighter complex does look like pausing, but strangely the pause appears to start before the motor arrives. In the second row of kymographs the first event is also clearly a wall-bound complex that does not induce pausing. In the leftmost kymograph in Figure 2—figure supplement 1, the microtubule tip appears to pause even in the absence of any complex. In the middle panel in the same figure one pause starts before the motors arrive.

D) The lines for moving motors are segmented, with only 3-6 dots per 2 micron (based on Figure 2—figure supplement 1 images). Thus, single GFP dimers near the tip would be easy to miss (see rightmost image in the supplemental figure where only the longest lines are clear). This raises the possibility that several molecules could cause pausing events. Please provide more images with improved time resolution and better contrast to determine if single molecules can indeed induce pausing. XY plots for motor vs. tip position should also be presented.

E) Plotting the duration of pause vs. tip brightness will address the concern that the pauses are brought about by multiple molecules.

F) The quantification of the number of GFP molecules is very important for this study. For example, the evidence in Figure 2 is not very convincing. The population of molecules appears to be very heterogeneous. The authors should show bleaching step analysis with pairwise intensity distributions, as is common in the single molecule field. The impact of the evanescence field depth should also be taken into account and discussed.

3) The claim that the full-length motor induces catastrophe is not well justified. Please provide quantitative data to confirm this.

4) The authors provide velocity measurements for KIF21B constructs (full length and MD-CC1) that are expressed in cells. Do these represent the speed distributions of single molecules or motors that travel in teams? Either should be substantiated by fluorescence intensity distributions of GFP spots that have been tracked.

5) Although GFP intensity measurements for full length KIF21B are provided (Figure 2) – which support the conclusion that KIF21B is a dimer – similar measurements are not provided for other constructs. In some cases, statements are made that are not supported by such data, for example "We conclude that the motor domain of KIF21B in a dimeric configuration is motile and can inhibit but not block microtubule plus-end depolymerization". This should be addressed.

6) There is some negative-stain EM that is presented as evidence that KIF21B is folded and globular. Please include more than 2 images as evidence, perhaps in a small gallery in the Supplement.

7) Why do so few KIF21B molecules can escape the seed area (see Figure 2). A higher affinity for the seed might recruit motors there, but why are they stuck?

8) The data on the length-dependence of these results (Figure 2) is underdeveloped relative to the data on other kinesin-4's and especially kinesin-8's.

9) The extent of protein overexpression in cells (Figure 1 and Figure 3) is not document or discussed.

10) It is interesting that the L-WD40 construct localizes strongly to the GMPCPP seed, but what is the explanation for why the construct does not localize to growing microtubule tips (Figure 4)? If the WD40 domain of KIF21B "senses" the nucleotide state of tubulin within the lattice, then one might expect for it to target tips (similar to TPX2, as recently shown by the Surrey laboratory). That it does not suggests that the interaction is blocked by a feature at microtubule plus ends, perhaps curvature. Related to this point, L-WD40 localizes uniformly along the microtubule lattice in cells (Figure 3), which is presumably representative of GDP-tubulin binding. This requires clarification.

11) The authors emphasize repeatedly that L-WD40 domain shows "strong preference" for the GMCPP lattice, speculating about the implications of this property for motor's behavior at the tip. This statement is based on couple images in Figure 4—figure supplement 1, in which GMPCPP seed and GDP microtubule walls are compared. However, these microtubule fragments differ in other respects, such as percent of labeled tubulin, biotin label and attachment to the coverslip (seed is attached while the GDP part is not). Moreover, this experiment was done with cell extracts, rather than with purified proteins, so many other MAPs were present. To be certain that L-WD40 can discriminate different microtubule lattices, microtubule binding affinity should be quantified using purified proteins and similarly prepared and coverslip-attached microtubules.

12) Please provide histograms of run lengths and velocities, not just their averages.

13) The authors did a great job dissecting the domain structure of KIF21B, and the microtubule binding results are represented clearly with Table in Figure 3. An analogous table for the in vitro data should be included.

14) Please describe all buffer conditions in more detail (see major point 1). Also, is ATP present in the buffers used for the microtubule co-pelleting experiments (Figure 3)?

15) A short discussion on how the results obtained in this study compares to the recent paper by Muhia et al., published in Cell Reports should be included. Muhia et al. find that microtubule growth rates are increased by KIF21B: in their hands, microtubules grow faster when the motor is overexpressed, and slower when the motor is depleted. Muhia et al. do show that KIF21B reduces the average time to catastrophe (although the effect is quite modest), which is consistent with results here.

[Editors’ note: what now follows is the decision letter after the authors submitted for further consideration.]

Thank you for submitting your work entitled "Kinesin-4 KIF21B is a potent microtubule pausing factor" for consideration by *eLife*. Your article has been reviewed by three peer reviewers, and the evaluation has been overseen by a Reviewing Editor and a Senior Editor. The reviewers have opted to remain anonymous.

Our decision has been reached after extensive consultation between the reviewers. Based on these discussions and the individual reviews below, we regret to inform you that your work will not be considered further for publication in *eLife*.

Three reviewers, whose evaluation was overseen by a Reviewing Editor, have now reviewed your revised manuscript. While all agreed that this was a stronger manuscript, unfortunately there were remaining major concerns that preclude acceptance for publication in *eLife*. We hope that these specific comments will help you revise your work for publication elsewhere.

1) The reviewers continued to be unconvinced that single molecules of KIF21B cause pausing, a statement that was toned down in places in the manuscript, but explicitly stated in others. Some comments included:

– The KIF21B preparation appears to be heterogeneous (as evidenced by the broad distribution of brightness in Figure 5). There are many noticeable differences in brightness throughout the kymographs that are shown. This raises the possibility that multiple motors are being analyzed in some cases.

– The brightness of single GFP molecules is compared to KIF21B. However, single GFP molecules will be present on the glass surface, while KIF21B will be located on the MT. This places GFP and KIF21B in a different region of the evanescent field, which decays exponentially. This could also lead to an underestimate of the number of KIF21B molecules.

– Photobleaching of KIF21B that could induce pausing was not taken into account. Prior photobleaching could also lead to an underestimate of the number of molecules present.

2) Perhaps most troubling, in two places in the revision the figures that are shown do not match the data described in the text.

– In Figure 1 at 2 nM MD-CC1 KIF21B moves towards the microtubule plus end, but at 10 nM MD-CC1 KIF21B appears to move in the opposite direction. Does this mean that it was difficult to distinguish plus and minus ends? How often where minus ends confused with plus ends? This is not discussed in the text and is quite troubling.

– The text claims that pausing is never observed in the presence of EB3 alone and that all observed pauses correlated with the presence of KIF21B. However, there are numerous examples where this is not the case: 1) Figure 1—figure supplement 4: there is a pause in the presence of 10 nM GFP. 2) Figure 2—figure supplement 1: in the first panel the second pause is not associated with GFP-labeled KIF21B. 3) Figure 3—figure supplement 3 it appears that the microtubule is already paused prior to KIF21B-GFP arrival.

The detailed comments of the reviewers are below.

Reviewer #1:

The revised manuscript by van Riel et al. contains a substantial amount of new data. Most new data relate to reviewers' comments. It is notable, however, that the authors now include new data that address the mechanism of Kif21B autoinhibition (Figure 4—figure supplement 3).

Overall, I found the revised paper to be significantly improved. I still remain mystified by how single Kif21B molecules can affect microtubule plus-end dynamics (even after the plus-end has extended beyond the point of Kif21B-microtubule association), and was not terribly satisfied by its comparison to the Eribulin. However, the work has been carefully conducted and contains the appropriate controls. I now support publication of this manuscript in *eLife*.

Reviewer #2:

The authors have addressed many critical points, leading to a stronger manuscript. However, the main criticism concerning the authors' claim that a single molecule of KIF21B is capable of causing specific modulation of the microtubule tip behavior has not been addressed fully. Although some statements to this effect have been softened in the main text, this claim is firmly stated in several places, e.g. see impact statement or labels in Figure 5.

I am concerned about this statement because there are multiple indications in this manuscript that the molecular composition of KIF21B preparation is heterogeneous: the distribution of brightnesses of these molecules is broad (e.g. see Figure 5 and point 2 below), there are visible differences in the brightness of molecules in virtually all kymogpraphs (e.g. see Figure 3) and movies (e.g. Video 2, Figure 3—figure supplement 3), there are visible differences in the size of globules attributed to KIF21B in EM images (some globules are almost 15-20 nm in diameter, exceeding the size of single molecule of Kif21B molecular weight), and the molecular effects at microtubule tip are also heterogeneous (e.g. some molecules induce pausing, while others induce catastrophe). My interpretation of these results is that small oligomers, not just the single molecules of KIF21B, are present in experimental chambers, which is often the case with purified proteins.

To support their claim that only single molecules can elicit certain effects the authors provide few selected examples in which one fluorescent complex appears to lead to some specific effect (e.g. pausing) but the rigorous proof that these specific dots correspond to 1 or 2 Kif21B molecules is lacking. Many or some of these dots, in my opinion, could correspond to complexes of 3-4 molecules, consistent with the heterogeneities listed above. I do not think the slightly larger size of the complexes changes the main message of the paper, but if the authors insist on their quantitative 1-2 molecules statement, I will have to ask for a more convincing evidence. This manuscript is considered for a publication in a very visible and respected journal, so keeping high standards for single molecule aspects in this paper is important.

Specific criticism of the brightness analyses in this paper:

1) The single most important quantification that is lacking in this paper is a quantitative comparison of the brightness of a single GFP molecule with the brightness of the complexes at the MT tip imaged under identical conditions and while taking into account the differences that could not be eliminated. Brightness of single GFP molecule is measured in this paper but it is reported in the manner that prevents any meaningful comparisons. Indeed, brightness of a single GFP molecule is roughly 2,000 au in Figure 1,Figure 5 au in Figure 2; ~1,200 au in Figure 2; in the range of 1,700- 2,500 au in different panels in Figure 5. At the same time, the authors claim that brightness 20 au corresponds to a single motor (2 GFPs) in Figure 2, ~ 40 au and > 60 au in Figure 3—figure supplement 1 A and B. This clearly illustrates a lack of effort to use these quantifications for obtaining a convincing estimate of the number of GFP molecules at the microtubule tip.

To address this omission, the authors should provide all peak values with errors for their fits and use consistent imaging conditions and data representation, so that all measurements for single GFP reported in this work are consistent and can be used to compare with the brightness of kinesin molecules that elicit certain response at the microtubule tip (e.g. pausing, see previous criticism 2E, which has not been addressed). For accurate comparison, the authors will have to take into account two additional factors that can greatly affect their conclusions:

First, brightness of single GFP molecules was measured in this work at the coverslip surface, while microtubule tips are located at some distance away. I do not believe that measuring this distance for every event can be done (or should be done) accurately, but since the evanescent field decays exponentially, this effect cannot be simply ignored, as in this paper. Approximate estimate of the average depth at which MT tips were imaged should be obtained and used to calculate the brightness at the coverslip surface. Without such an adjustment, the ratio of approximately 1 of Kif21B intensity at the microtubule tip to the GFP intensity on the coverslip (Figure 2) cannot be used to argue that these complexes have the same number of molecules.

Second, the authors should provide evidence that they are justified to ignore bleaching. The initial intensity of GFP molecules was measured, while KIF21B dots were visualized for extended time. Few examples of Kif21B intensity kinetics on microtubule shown in this paper are too noisy to make reliable conclusions about the extent of bleaching during the observation time, so this issue should be addressed by direct measurements. This concern is valid especially for KIF21B because, as reported in this paper, this motor first binds at the GMPCPP seeds and only few molecules escape later and run along the GDP lattice. The exact time these "escaped" motors spent on the seed is not known. It is possible that only the relatively large clusters escape and that at this time some of their GFPs have already bleached.

Importantly, both factors, the z distance and bleaching, can lead to underestimation of the size of Kif21B complexes, so the dots that seem to contain 1-2 molecules may in fact be larger. Moreover, It has not been ruled out by the authors that processivity of Kif21B molecules, or their ability to walk on GDP lattice, or the strength of the effects they cause at the MT tip increase with increasing number of clustered molecules. Any one of these plausible possibilities could help to "select" larger complexes from heterogeneous population even if they were present in minor quantities.

Other errors in the manuscript:

2) In the rebuttal letter and manuscript text the authors state that the lognormal distribution (Figure 1, Figure 2, Figure 5) of the intensities of individual GFP fluorophores "precisely accounts for the effect of evanescence field on otherwise expected normal distribution of intensities". No reference was provided. I am not familiar with this effect and believe that this statement is wrong. Methods section explains that these intensities were collected from molecules attached to the coverslip surface, so all molecules were positioned in the same plane. The decay in intensity at distances away from the coverslip is therefore irrelevant for interpreting the shape of these experimental distributions. Moreover, even if the molecules were located at variable distances from the coverslips, this would have increased the proportion of molecules with the lower than peak intensity. The lognormal shape of the distribution, characterized by the presence of complexes with higher than peak intensities, can be caused by other experimental factors. First, by a relatively high density of molecules on the coverslip, so that 2 or 3 single molecules could not be resolved. This explanation is indeed possible in the current work because the authors use 2-9 times higher density than recommended by others for this type of analysis (Jain et al., 2012). Second, the lognormal shape is expected for heterogeneous preparations, so it may simply reflect presence of a significant proportion of complexes that are larger than single molecules. This second possibility supports my concern that the actual size of Kif21B complexes acting at the microtubule tip is larger than stated by the authors.

3) New Figure 1 is remarkable because it shows that at 2 nM the MD-CC1 motor runs toward the tip of MT, but at 10 nM it switches polarity and runs toward the seed. The authors will have to explain this result, but I suspect that for 10 nM panel they show imaging of the minus MT end, not the plus. This is very disturbing because it implies that these experiments were done under conditions when discriminating between different MT ends was difficult, the directionality of motor's walking was not taken into account, and the data from different types of ends were combined to show retardation of MT polymerization rate. In Discussion section the authors write: "The dimeric version of the motor is sufficient to inhibit microtubule growth.. It is possible that KIF21B motors arriving at the microtubule tip somehow affect the conformation of terminal tubulin dimers, thereby reducing the tubulin on-rate." If this is the model, how do the authors explain that the velocity of growth of minus end was also reduced in Kif21B presence? Or is it possible that the reported results are affected by the relative sampling of the plus and minus end kinetics and the authors failed to discriminate these effects? This situation seems unacceptable to me because it raises serious concern about the quality of data collection and analysis.

4) The authors state that they "..have never observed any pausing in our in vitro assays in the presence of EB3 alone, and all observed pauses could be correlated with the presence of KIF21B at the microtubule tip. However, it is true that we did observe events when microtubule polymerization appeared to slow down before the arrival of KIF21B and subsequent pausing." The distinction between "pausing" and "slowing down" is unclear to me. Multiple examples in this paper show pausing/slowing in the absence of KIF21B. See Figure 1—figure supplement 4 (Tub 15 μm with GFP middle of the first track). In Figure 2—figure supplement 1 panel A only first pause is associated with GFP. In Figure 3—figure supplement 3 polymerization pauses at 10-25 s, ie before the arrival of GFP-labeled molecule. Figure 3 panel F lower example clearly shows that MT growth is stalled in the absence of GFP signal. It could either be caused by the bleached GFP-kinesins or by a low quality of tubulin preparation. These issues have not been addressed or even mentioned. Since pausing at the MT end is the primary focus of this paper, I am very concerned about the discrepancy between author's statement and actual images.

Reviewer #3:

van Riel et al. have resubmitted their paper on the kinesin-4 KIF21B. Overall, I feel they have done a fairly solid job of responding to our concerns. The data quality has improved in many of the figure panels. The mass-spec of their contaminants and the analysis of particle brightness are particularly welcome.

It is still very puzzling that a single transient encounter of KIF21B with the microtubule end can slow down the growth rate by half. Their rebuttal just says: "this is how it is"--and they make a point about Eribulin. It was indeed a striking point about their Curr Biol paper that a single eribulin could slow down growth (and produce a secondary EB comet!). But of course it was also the case that microtubule growth resumed after the Eribulin unbinds (e.g., see Figure 4, bottom half).

I was disappointed that they did not perform their experiments at multiple tubulin concentrations. They say that "further mechanistic insights" won't come out of such experiments, but that's clearly not true. Microtubule end structure is thought to change as a function of tubulin concentration, so the motor may struggle/succeed in suppressing growth differently at different tubulin concentrations. I'm on the fence about whether to insist that they perform these experiments. At the end of the day, this paper isn't the kind that deals with biophysical precision of that type, and while we could force them to perform the experiment, it's unclear that their data/interpretation would be definitive.

[Editors’ note: what now follows is the decision letter after the authors submitted for further consideration.]

Thank you for submitting your article "Kinesin-4 KIF21B is a potent microtubule pausing factor" for consideration by *eLife*. Your article has been reviewed by two peer reviewers, and the evaluation has been overseen by a Reviewing Editor and Vivek Malhotra as the Senior Editor. The reviewers have opted to remain anonymous.

The reviewers have discussed the reviews with one another and the Reviewing Editor has drafted this decision to help you prepare a revised submission.

As you will see from the two reviews below (both are two of the original reviewers of your paper), the reviewers are now quite positive about your work. I would ask that you only revise the manuscript to take into account the two points raised by reviewer #2: 1) the relationship between the dwell time of the motor on a protofilament and the ability of the motor to induce a catastrophe, and 2) be more explicit in your discussion of protofilament curvature in the Discussion or remove the sentence referred to by reviewer #2.

Reviewer #2:

In this second revision by van Riel et al., the authors address major criticisms related to whether single molecules of KIF21B are indeed capable of causing microtubule plus ends to pause. There is an abundance of new content, including more intensive data analysis and experiments using kinesin-1 as a dimer control. As a result, the paper is now quite dense (I recognize that this is the outcome of the review process), and I am more convinced that low numbers (likely single molecules) of KIF21B motors are capable of stunting microtubule assembly. The authors are now much more explicit regarding models regarding how KIF21B might act; I found these text additions quite useful. I would encourage the authors to also update their model figure to include the action of KIF21B on single protofilaments. Overall, my disposition is that this manuscript is appropriate for *eLife*, but there are two issues that should be addressed. The first concerns the relationship between the dwell time of the motor on a protofilament, and the ability of the motor to induce a catastrophe. The MD-CC1 truncation does a good job of slowing microtubule growth, without dwelling appreciably anywhere on the lattice. This is quite a remarkable activity, and an indication that the motor must be doing something subtle to the microtubule lattice/protofilament end. By contrast, the last two constructs that are analyzed in the manuscript show no activity, and the authors therefore conclude that: "Taken together, these results suggest that both the regulatory rCC region and the C-terminal WD-40 containing domain can contribute to the ability of KIF21B to stay attached to the growing MT plus end and to induce pausing." These two interpretations are at odds with each other and must be reconciled. Second, the authors suggest that "It is possible that at growing MT plus ends the number of binding sites for which the C-terminus of KIF21B would have preference or its affinity for these sites would be affected by protofilament curvature." In Figure 4—figure supplement 1, the authors show clear examples of microtubule bending prior to a KIF21B-induced catastrophe, but KIF21B is not bound to these deformed plus ends. The statement in the Discussion is thus countered by their own data. I would suggest that the authors are more explicit regarding their ideas or remove the sentence from the Discussion.

Reviewer #3:

van Riel et al. have submitted a second revision of their paper on the kinesin-4 KIF21B. The Akhmanova lab has responded forcefully to every technical concern that was raised. My main concern after the 2nd review was that they had not performed experiments at multiple tubulin concentrations. These data are now included in the manuscript and, indeed, the results are an interesting addition to their story. I am supportive of publication. Although many of the results are still puzzling, the puzzle should now belong to the field.

---

## [Author Response]

*Essential revisions:*

All three reviewers raised major concerns that must be addressed in order to consider publication in eLife.

*1) The authors' conclusions for Figure 1 are very puzzling. Although the quality of the kymographs in Figure 1 is not high, individual particles of MD-CC1 can be seen arriving at the microtubule tip occasionally and immediately detaching. The fraction of time that the tip is in contact with one such molecule is very small relative to the total time of microtubule growth, while during this time the microtubule end polymerizes smoothly and continuously at a rate slower than in the control. The authors' conclusion that the slowing of growth velocity is due to these infrequent encounters seems improbable because this would imply a long-distance effect (e.g. the motor walks on the wall and this causes the tip to polymerize slowly) or some kind of memory (e.g. the motor did something to the tip and even after the motor detaches the rate of assembly remains slower than normal). Other explanations are possible. For example, are there any differences in the buffer conditions? The authors did not provide the final concentrations for buffers in which these observations were made, but the methods section suggests that the experiments with MD-CC1 were done in the presence of some additional reagents (e.g. detergent) and TEV protease, while EB3 control experiments lacked these additions. The authors should rule out this trivial explanation by adding GFP protein purified analogously to MD-CC1 to their EB3 control. If it really appears that microtubule growth is perturbed even when no MD-CC1 molecules are seen at the tip, the authors should add a discussion about possible molecular mechanisms for this very striking observation. A similar conclusion was made for the full length motor, but again microtubule polymerization appears to have been slowed when only a few motor molecules were present at the tip.*

To address this concern, we have purified GFP alone from HEK293T cells using TEV protease cleavage (see Figure 1—figure supplement 3 for an image of a Coomassie blue stained SDS PAGE), and performed microtubule dynamics assays with this protein as a control. We observed no effect of GFP alone on microtubule polymerization, while increasing concentrations of the MD-CC1 protein slowed down microtubule growth (new Figure 1). A further argument that the observed effect is real and not an artifact of our in vitro assay is provided by the observation that a similar approximately two-fold decrease of microtubule growth rate was induced by expressing MD-CC1 of KIF21B as well as it homologue KIF21A in cells (Figure 1). New kymographs, including an enlargement that illustrates better the behavior of KIF21B MD-CC1 proteins in vitro are now included in Figure 1 and Figure 1—figure supplement 4.

Additional information on the concentrations of the buffers used in the in vitro assays is now included in the Materials and methods section.

We fully agree with the reviewers that the obtained results are somewhat puzzling as KIF21B motors slow down microtubule growth although they do not accumulate at tips. We think that the most likely explanation is that motors running off the microtubule plus end somehow alter its structure by affecting the conformation of the terminal tubulin dimers with which they interact, and this has an effect on the addition of new tubulin dimers. We note that in our very recent paper on the mechanism of microtubule growth inhibition by the anticancer drug Eribulin we showed that binding of single drug molecule (which can interact with only one tubulin dimer at a time) was sufficient to perturb growth of a microtubule tip by slowing down its polymerization rate (Doodhi et al., 2016). We think that in a similar mechanism, single KIF21B motors running off the polymerizing microtubule plus ends could transiently perturb their growth resulting in an overall reduction of the microtubule polymerization rate. We now discuss this point in the Discussion section of the revised manuscript.

Related to this concern, the microtubule growth rate (~4 μm/ min at 15 μm tubulin) shown in Figure 1 is quite fast compared to other reports in the literature. Please provide a complete plot of microtubule growth rate vs. tubulin concentration, with and without KIF21B as well as microtubule growth rate vs KIF21B concentration.

The experiment described here was performed in the presence of EB3, which we have shown previously to accelerate microtubule polymerization (Komarova et al., 2009). We have now also included a plot of microtubule growth rates at different concentrations of KIF21B with and without EB3 (Figure 1) to show that the reduction in growth rate is stronger at higher KIF21B MD-CC1 concentrations, and that this effect does not depend on the presence of EB3. We do not think that using different tubulin concentrations in these experiments will provide any additional mechanistic insight into the activity of KIF21B MD-CC1 and therefore did not include them.

2) That single molecules of KIF21B may be sufficient to cause microtubules to cease growing and shortening is fascinating. The main evidence is Figure 2, where a kymograph shows an alleged single molecule hitting the end and causing it to pause. However, the authors do not pursue this phenomenon at a mechanistic level, or provide a reasonable discussion that explains the observation. Both should be addressed in a revised manuscript and the following points must be taken into account:

A) What is the positional resolution of that experiment? Has the microtubule really paused, or has growth simply dropped to a level that is below detection?

The resolution of our experiments is certainly limited by the diffraction limit as well as the pixel size of the camera, and therefore, we cannot resolve individual tubulin dimers. Thus, we cannot distinguish genuine pausing from very slow microtubule growth or shortening. However, we include in the assay both Rhodamine-labeled tubulin and mCherry-EB3 to facilitate detection of microtubule growth and detect pauses by observing the loss of enrichment of the red EB3 signal at the microtubule plus end, which indicates that microtubule growth was at least very severely slowed down. Furthermore, we have added new data showing that when a microtubule extends beyond the point of KIF21B attachment to the microtubule plus end, the growth of this microtubule is strongly perturbed: we observed either short growth or shortening excursions, and/or strongly increased flexibility of the extending microtubule, suggesting that the growing tube might be incomplete. These data are reminiscent of our recently published data on the microtubule targeting agent Eribulin, where we found that a single drug molecule, which can block only a single protofilament, can significantly perturb microtubule growth (Doodhi et al., 2014). The main difference between KIF21B and Eribulin is that KIF21B can also stabilize microtubule plus ends, while Eribulin can only perturb microtubule elongation but has no stabilizing effect. We therefore propose that pausing by a single KIF21B molecule is caused by occluding only some protofilaments, while the rest remain dynamic and may grow and shorten for some distances.

B) It is suggested that the affinity of the WD40 domain for GMPCPP tubulin might stabilize motor-tip interactions, but this hypothesis still cannot explain an enzyme activity that probably involves more than one protofilament. A discussion of motor dimensions may help. The WD40 domain, for example, is 4 nanometers in diameter. What is the predicted length of the stalk? Can a single KIF21B motor crosslink multiple neighboring protofilaments?

KIF21B is a long polypeptide which contains potentially unstructured regions that can be quite extended that could allow the folded domains of KIF21B to interact with different sites on the microtubule. A KIF21B dimer contains six potential microtubule-interacting regions – two motors, two WD40 domains and two microtubule-binding regions in the stalk, which could all interact with the same or different protofilaments. As also mentioned above, our recently published experiments with Eribulin (Doodhi et al., 2016) showed that an agent binding to a single tubulin dimer at the microtubule tip is sufficient to strongly perturb microtubule growth. Therefore, one or two KIF21B molecules bearing six microtubule binding domains each can indeed be envisaged to induce transient microtubule pausing (or strong growth inhibition) even if they do not occlude all protofilaments of a microtubule. These aspects are now discussed in the Discussion section of our revised manuscript.

*A minor issue related to this point concerns the purity of the KIF21B preparation (Figure 1—figure supplement 2). What is the stoichiometric band that runs at ~65 kDa? Are the authors sure that this protein is irrelevant for the observed KIF21B effect on microtubule dynamics?*

To address this comment, we performed a complete mass spectrometry analysis of our purified proteins (see new [Supplementary-material SD14-data]). We did not find any microtubule regulators in our purifications; the main contaminants are chaperones, spectrin, vimentin, desmoplakin and 14-3-3 proteins. The ~65 kDa band corresponds to a heat shock protein. We note that the contaminants observed in KIF21B full length (FL), FL-ΔrCC and MD-CC-ΔrCC samples were very similar, while only the full length KIF21B protein efficiently caused pausing, supporting the specificity of the observed effect.

C) The FL motor kymographs in Figure 2 are difficult to see, but almost all of the visible events in panel D are consistent with "stay on MT lattice" behavior, while the authors claim that the majority of events (~45%) are pausing of MT growth. The pausing is very striking and compelling for high motor decoration, but is the same phenotype observed for the dim complexes, which might be single molecules? For example, in panel D, the first 2 events in the upper kymograph have dim signals and are clearly "lattice" events. The third event with a somewhat brighter complex does look like pausing, but strangely the pause appears to start before the motor arrives. In the second row of kymographs the first event is also clearly a wall-bound complex that does not induce pausing. In the leftmost kymograph in Figure 2—figure supplement 1, the microtubule tip appears to pause even in the absence of any complex. In the middle panel in the same figure one pause starts before the motors arrive.

We now show better examples of pausing events induced by single KIF21B motors (see Figure 2); we also show plots of motor intensity overlaid with plots of microtubule growth and X-Y plots of the motor vs microtubule tip position (see Figure 2). We note that we have never observed any pausing in our in vitro assays in the presence of EB3 alone, and all observed pauses could be correlated with the presence of KIF21B at the microtubule tip. However, it is true that we did observe events when microtubule polymerization appeared to slow down before the arrival of KIF21B and subsequent pausing. It is possible that a slower growing microtubule is more easily converted into a pausing state. Furthermore, as discussed above, we now provide a more extensive illustration and analysis of the “stay on the lattice” events (Figure 3—figure supplement 1 and Figure 3—figure supplement 2), and show that such events are often manifested by microtubule growth perturbation, though not a complete pausing, and are consistent with idea that a single KIF21B molecule would block growth of a few protofilaments.

D) The lines for moving motors are segmented, with only 3-6 dots per 2 micron (based on Figure 2—figure supplement 1 images). Thus, single GFP dimers near the tip would be easy to miss (see rightmost image in the supplemental figure where only the longest lines are clear). This raises the possibility that several molecules could cause pausing events. Please provide more images with improved time resolution and better contrast to determine if single molecules can indeed induce pausing. XY plots for motor vs. tip position should also be presented.

As indicated above, we now show the plots of motor intensity vs motor position/microtubule elongation (Figure 2, Figure 3—figure supplement 1). Collecting the data with a better temporal resolution proved difficult because at low motor concentrations, the pausing events are quite rare, so a lot of material needs to be screened in order to find them. At higher KIF21B concentration all seeds are blocked and no microtubule growth/pausing events can be observed.

E) Plotting the duration of pause vs. tip brightness will address the concern that the pauses are brought about by multiple molecules.

Plots of motor intensity vs motor position/microtubule elongation are shown in Figure 2.

F) The quantification of the number of GFP molecules is very important for this study. For example, the evidence in Figure 2 is not very convincing. The population of molecules appears to be very heterogeneous. The authors should show bleaching step analysis with pairwise intensity distributions, as is common in the single molecule field. The impact of the evanescence field depth should also be taken into account and discussed.

Over the past years, we have tested and applied multiple approaches to carefully analyze the number of GFP-tagged proteins involved in microtubule-based events (e.g. Kapitein et al. JCB 2008, Curr Biol. 2008, Doodhi et al., 2014, 2016). We, as many others, have found that the rich photophysical properties of GFP (e.g. blinking and different fluorescent states, Dickson et al. Nature 1997) make a bleaching step analysis not as reliable as it can be for stable organic dyes. While example traces with clearly identifiable bleaching steps can certainly be observed and analyzed (new Figure 1, Figure 2, Figure 5), a more reliable approach is to compare the initial, pre-bleach intensities with known standards measured in the same conditions, as shown in Figure 1, Figure 2 and Figure 5. We now also show that the histogram of intensities of individual GFP fluorophores can be fitted with lognormal distribution (Figure 1, Figure 2, Figure 5). This fact precisely accounts for the effect of evanescence field on otherwise expected normal distribution of intensities.

3) The claim that the full-length motor induces catastrophe is not well justified. Please provide quantitative data to confirm this.

We now show a clearer example of a catastrophe induction event as well as a quantification of the catastrophe frequency in the presence of full length KIF21B (Figure 3).

4) The authors provide velocity measurements for KIF21B constructs (full length and MD-CC1) that are expressed in cells. Do these represent the speed distributions of single molecules or motors that travel in teams? Either should be substantiated by fluorescence intensity distributions of GFP spots that have been tracked.

While the velocity of kinesins in cells can be determined unambiguously from kymographs, obtaining meaningful intensity distributions is much more difficult due to low signal and high background in the images. We note that we do not make any conclusions about the single molecule behavior or clustering of KIF21B in cells.

5) Although GFP intensity measurements for full length KIF21B are provided (Figure 2) – which support the conclusion that KIF21B is a dimer – similar measurements are not provided for other constructs. In some cases, statements are made that are not supported by such data, for example "We conclude that the motor domain of KIF21B in a dimeric configuration is motile and can inhibit but not block microtubule plus-end depolymerization". This should be addressed.

GFP intensity measurements, demonstrating that these proteins are dimers are now shown for all motile KIF21B constructs used in this study (Figure 1, Figure 2, Figure 5).

6) There is some negative-stain EM that is presented as evidence that KIF21B is folded and globular. Please include more than 2 images as evidence, perhaps in a small gallery in the Supplement.

Additional EM images are now included in Figure 4—figure supplement 1.

7) Why do so few KIF21B molecules can escape the seed area (see Figure 2). A higher affinity for the seed might recruit motors there, but why are they stuck?

It is possible that the border between the seed and the GDP microtubule lattice displays some defects to which KIF21B preferentially binds. This possibility is now mentioned in the Discussion section of our revised manuscript.

8) The data on the length-dependence of these results (Figure 2) is underdeveloped relative to the data on other kinesin-4's and especially kinesin-8's.

We have now extended the data demonstrating that the ability of KIF21B to block microtubule seeds depends on the length of the seeds (Figure 2). Unfortunately, we cannot extend these data to studies of the length-dependence of the effect of KIF21B on dynamic microtubules, because in the conditions that we used all landing events are observed on the seeds, and the motors that do move on to GDP lattice are highly processive.

9) The extent of protein overexpression in cells (Figure 1 and Figure 3) is not document or discussed.

The COS-7 cells used in this study do not express endogenous KIF21B (this is now mentioned in the first paragraph of the Results section of our revised manuscript), and therefore we cannot discuss the level of KIF21B overexpression in this system. The experiments in cells are used only to illustrate the ability of the motor to strongly block microtubule growth, as support for this otherwise in vitro study. We do not make any strong conclusions about the cellular activity of the protein.

10) It is interesting that the L-WD40 construct localizes strongly to the GMPCPP seed, but what is the explanation for why the construct does not localize to growing microtubule tips (Figure 4)? If the WD40 domain of KIF21B "senses" the nucleotide state of tubulin within the lattice, then one might expect for it to target tips (similar to TPX2, as recently shown by the Surrey laboratory). That it does not suggests that the interaction is blocked by a feature at microtubule plus ends, perhaps curvature. Related to this point, L-WD40 localizes uniformly along the microtubule lattice in cells (Figure 3), which is presumably representative of GDP-tubulin binding. This requires clarification.

It is still not clear how well microtubule lattices bound to different GTP analogues mimic the tubulin states present at the growing microtubule tip and how these states are distributed at the tip. Therefore, the isolated L-WD40 domain might be difficult to detect at the microtubule tips because the number of actual high-affinity sites is low, or because the conformation of these sites, e.g., the curvature, is different at the growing microtubule tips. Furthermore, using mass spectrometry analysis of KIF21B binding partners, we have detected weak interactions between the WD40 domain and some MAPs (which are not present in our purified KIF21B samples due to high salt washes). Describing these interactions and their impact on KIF21B function goes beyond the scope of this manuscript, but we think that they might contribute to the uniform distribution of L-WD40 domain along cellular microtubules. These MAPs could in fact serve as additional microtubule loading factors for KIF21B in cells, explaining how KIF21B would attach to microtubules in the absence of a GMPCPP seed. Furthermore, tubulin post- translational modifications could also play a role. Thus our in vitro reconstitutions cover only a part of the complexity of KIF21B interactions with microtubules, as now discussed in the Discussion section of our revised manuscript.

11) The authors emphasize repeatedly that L-WD40 domain shows "strong preference" for the GMCPP lattice, speculating about the implications of this property for motor's behavior at the tip. This statement is based on couple images in Figure 4—figure supplement 1, in which GMPCPP seed and GDP microtubule walls are compared. However, these microtubule fragments differ in other respects, such as percent of labeled tubulin, biotin label and attachment to the coverslip (seed is attached while the GDP part is not). Moreover, this experiment was done with cell extracts, rather than with purified proteins, so many other MAPs were present. To be certain that L-WD40 can discriminate different microtubule lattices, microtubule binding affinity should be quantified using purified proteins and similarly prepared and coverslip-attached microtubules.

12) Please provide histograms of run lengths and velocities, not just their averages.

We have now compared side by side the binding of purified L-WD40 domain at different concentration to GMPCPP- and taxol-stabilized seeds, which were prepared and attached to glass in the same way, and found that, in agreement with our previous conclusion, the L-WD40 fragment strongly preferred GMPCPP seeds but displayed no specific affinity for taxol-stabilized seeds (Figure 5, Figure 5—figure supplement 1).

*13) The authors did a great job dissecting the domain structure of KIF21B, and the microtubule binding results are represented clearly with Table in Figure 3. An analogous table for the* in vitro *data should be included.*

A table summarizing the microtubule-binding results for different KIF21B mutants in vitro is shown in Figure 5—figure supplement 1.

14) Please describe all buffer conditions in more detail (see major point 1). Also, is ATP present in the buffers used for the microtubule co-pelleting experiments (Figure 3)?

A more detailed description of buffer conditions is now included in the Materials and methods section. No additional ATP was added in the pelleting assay.

15) A short discussion on how the results obtained in this study compares to the recent paper by Muhia et al., published in Cell Reports should be included. Muhia et al. find that microtubule growth rates are increased by KIF21B: in their hands, microtubules grow faster when the motor is overexpressed, and slower when the motor is depleted. Muhia et al. do show that KIF21B reduces the average time to catastrophe (although the effect is quite modest), which is consistent with results here.

We have now included a short discussion on the comparison of our results with those of Muhia et al. (which appeared after our paper was submitted) in our revised manuscript. In brief, In KIF21B knockout neurons, the length of growth episodes detected with EB3-GFP was increased while the growth rate was decreased, which would be consistent with the idea that the presence of KIF21B induces either catastrophes or pausing, since both types of events would cause a loss of EB3 signal. If the primary impact of KIF21B is through interrupting growth episodes, the effect of its loss on microtubule growth rate could be indirect, by affecting the tubulin pool, which would be decreased if more microtubules keep growing in the absence of KIF21B, and this would result in a slower growth rate.

In Hela cells, Muhia et al. observed that the overexpression of the full length KIF21B, a KIF21B construct lacking the C-terminal part (similar to our MD-CC1) as well as a KIF21B point mutant in the motor domain all reduced the length of microtubule growth episodes, and that the full length protein also increased the growth rate and increased the catastrophe frequency. The increase in catastrophe frequency is consistent with our data, as described above. However, the changes in growth rate are not consistent: we observe a reduction of microtubule growth rate induced by KIF21B-MD-CC1 and the full length KIF21B both in COS-7 cells and in vitro, while Muhia et al. see an opposite effect with the full length KIF21B in HeLa cells. Unfortunately, no images of different overexpressed constructs are shown in the Muhia et al. study, and it is thus unclear whether in the conditions used, the proteins decorate microtubules, accumulate at microtubule tips, etc, making it difficult to directly compare the two datasets. The difference between the results obtained by us COS-7 cells and the results by Muhia et al. obtained in HeLa cells might be due to differences in the cell type used. COS-7 cells express no endogenous KIF21A or KIF21B, while HeLa cells express KIF21A and display very strong regional differences in microtubule plus end dynamics (cortex vs cell interior) dependent on the localization of this protein (van der Vaart et al., Dev Cell 2013). The potential interplay between KIF21A and KIF21B is currently unclear. Therefore, it is difficult to make a conclusive comparison between the KIF21B overexpression results obtained by us in COS-7 cells and by Muhia et al. in HeLa cells, and we prefer not to comment on this specific issue.

[Editors’ note: what now follows is the decision letter after the authors submitted for further consideration.]

Three reviewers, whose evaluation was overseen by a Reviewing Editor, have now reviewed your revised manuscript. While all agreed that this was a stronger manuscript, unfortunately there were remaining major concerns that preclude acceptance for publication in eLife. We hope that these specific comments will help you revise your work for publication elsewhere.

1) The reviewers continued to be unconvinced that single molecules of KIF21B cause pausing, a statement that was toned down in places in the manuscript, but explicitly stated in others. Some comments included:

– The KIF21B preparation appears to be heterogeneous (as evidenced by the broad distribution of brightness in Figure 5). There are many noticeable differences in brightness throughout the kymographs that are shown. This raises the possibility that multiple motors are being analyzed in some cases.

Some heterogeneity is inherent to all single molecule experiments, but, as shown in more detail below, we have additional data demonstrating that we are mostly observing one or two KIF21B molecules walking on microtubules and that a small number of KIF21B molecules is sufficient to induce microtubule pausing.

– The brightness of single GFP molecules is compared to KIF21B. However, single GFP molecules will be present on the glass surface, while KIF21B will be located on the MT. This places GFP and KIF21B in a different region of the evanescent field, which decays exponentially. This could also lead to an underestimate of the number of KIF21B molecules.

As outlined below, we have already published an analysis where we compared the intensity of single GFP molecules on glass to GFP-tagged kinesin-1 dimers (KIF5B deletion mutant 1- 560, which contains the motor domain and a dimeric coiled coil) walking on microtubules (Doodhi et al., 2014). As a result of being slightly further away from the cover slip, dimeric kinesin-1 was not 2-fold brighter than a single GFP molecule, but only by a factor 1.7-1.8. This fits very well with the data in the plot shown in Figure 2 of the previous version of the manuscript (Figure 5 of the new version of the paper), where we compare the intensity of motors inducing microtubule pausing with the intensity of immobilized dimers and find a ratio of 0.75-1.9, consistent with 0.9-2.2 motors (based on multiplication by 2/1.75).

In the newly revised manuscript, we have now also compared the intensities of GFP-tagged kinesin-1 (amino acids 1-560 of KIF5B, denoted KIF5B-560), which are dimers based on our own analysis as well as numerous previous publications, to our KIF21B full length molecules moving on microtubules in separate chambers *on the same coverslip* and thus in identical imaging conditions. We found that these intensities were the same, confirming our conclusion that we are observing events of motility and pause induction by one or two KIF21B molecules.

– Photobleaching of KIF21B that could induce pausing was not taken into account. Prior photobleaching could also lead to an underestimate of the number of molecules present.

Essentially, the concern here is that fluorescent spots that we interpret as one or two KIF21B molecules are in fact larger but partially photobleached oligomers that managed to escape from the seed to the dynamic microtubule lattice and appear as dimers on the latter part of the microtubule. First, we now show that most of the experiments described in the paper were performed at a laser power and frame rate where the average kinesin photobleaching time (~200 s) was significantly longer than the duration of kinesin runs (tens of seconds) (new Figure 5—figure supplement 2). We have also included in the paper a new experiment, in which we show an example of an event observed at a 12 times higher frame rate (and thus with a ~12 times shorter bleaching time) (Figure 5 of the revised paper). In this case, photobleaching does occur, and the intensity of the imaged KIF21B molecule is reduced by half, supporting the notion that it is a dimer. Furthermore, we provide a comparison of intensities for molecules moving on the seeds to molecules moving on dynamic microtubules to illustrate that the movements from the seed to the dynamic microtubule lattice do not occur preferentially for larger oligomers of KIF21B (Figure 5, Figure 5—figure supplement 1). Finally, we directly compare intensities of motile single KIF21B molecules to the intensities of single KIF5B-560 molecules, which are known to be dimers, and show that they are very similar (Figure 5).

2) Perhaps most troubling, in two places in the revision the figures that are shown do not match the data described in the text.

– In Figure 1 at 2 nM MD-CC1 KIF21B moves towards the microtubule plus end, but at 10 nM MD-CC1 KIF21B appears to move in the opposite direction. Does this mean that it was difficult to distinguish plus and minus ends? How often where minus ends confused with plus ends? This is not discussed in the text and is quite troubling.

We very sincerely apologize for the mistake in preparing this figure. While microtubule plus and minus ends were easily distinguished based on the directionality of the motor, we inadvertently showed the wrong end in one of the panels. In the conditions shown in this panel (10 nM KIF21B-MD-CC1), the growth rate of microtubule plus and minus ends was almost the same (compare new Figure 2 with Figure 2—figure supplement 3), while normally microtubule plus ends grow in vitro significantly faster than the minus ends. This observation demonstrates once again that the effect of KIF21B-MD-CC1 is not due to some impurities in the protein preparation, which affect the assay, but is rather due to a specific effect on microtubule plus end growth, as the rate of minus end polymerization is not altered. These data are now illustrated more clearly in the revised Figure 2, and Figure 2—figure supplement 3.

– The text claims that pausing is never observed in the presence of EB3 alone and that all observed pauses correlated with the presence of KIF21B. However, there are numerous examples where this is not the case: 1) Figure 1—figure supplement 4: there is a pause in the presence of 10 nM GFP. 2) Figure 2—figure supplement 1: in the first panel the second pause is not associated with GFP-labeled KIF21B. 3) Figure 3—figure supplement 3 it appears that the microtubule is already paused prior to KIF21B-GFP arrival.

We believe that this comment is incorrect: the examples listed above with the exception of one are *not* in the presence of EB3, and the last example in fact shows no pausing without KIF21B, as explained below and now illustrated better in the revised paper (Figure 3—figure supplement 2 in the current paper version, formerly Figure 3—figure supplement 3). We fully acknowledge that small pauses (or events that can be mistaken for pauses due to irregular microtubule lattice labeling) can be seen when microtubule dynamics are imaged with rhodamine-tubulin alone. This was exactly the reason why for most of the analyses in this paper, we have used the conditions when microtubules were labeled with both rhodamine-tubulin and mCherry-EB3, as this allowed us to unambiguously distinguish between microtubule growth and pausing.

The detailed comments of the reviewers are below.

Reviewer #1:

The revised manuscript by van Riel et al. contains a substantial amount of new data. Most new data relate to reviewers' comments. It is notable, however, that the authors now include new data that address the mechanism of Kif21B autoinhibition (Figure 4—figure supplement 3).

Overall, I found the revised paper to be significantly improved. I still remain mystified by how single Kif21B molecules can affect microtubule plus-end dynamics (even after the plus-end has extended beyond the point of Kif21B-microtubule association), and was not terribly satisfied by its comparison to the Eribulin. However, the work has been carefully conducted and contains the appropriate controls. I now support publication of this manuscript in eLife.

We acknowledge that we do not have a biochemical explanation of the effect of KIF21B motor domains on microtubule growth. Detailed structural studies similar to those performed over the years with kinesin-13s would be needed to understand the mechanism, and such studies obviously go beyond the scope of this paper. Our data on Eribulin published in our recent Current Biology paper (Doodhi et al., 2016) represent the first demonstration that blocking of a single microtubule protofilament is sufficient to very significantly perturb microtubule growth when a microtubule extends beyond the point of Eribulin attachment. These data suggest that protofilaments can elongate asynchronously: when one protofilament is blocked, the remaining ones can grow, but when a microtubule is incomplete, its growth is perturbed.

We agree that more data are needed to make this concept generally accepted. In fact, by showing that full length KIF21B molecules that stay on microtubule lattice can perturb microtubule growth beyond the point of their attachment, we provide independent support for this concept. In particular, we show that if KIF21B stays attached to the microtubule plus end but the microtubule keeps growing, microtubule polymerization becomes irregular, it can alternate between phases of growth and shortening, and microtubule elongation is often accompanied by bending. All these aberrations suggest that the microtubule extending beyond the site where KIF21B is immobilized is incomplete. These data are now illustrated better in the new Figure 4, and additional examples are provided.

Further, we propose that in case of dimeric KIF21B motor domains, the proteins do not stay on the microtubule plus ends, but their transient association with these ends can still be sufficient to transiently perturb the structure of the tip and thus reduce its growth rate. We have now obtained some experimental support for this idea. We reasoned that transient perturbation of the microtubule tip might be reflected in the brightness of the EB3 signal. Therefore, we performed the assay in the presence of 1 nM KIF21B-MD-CC1-GFP (to make the assay less crowded with the proteins running on microtubules) with faster imaging conditions (50-100 ms/frame). As shown in the new Figure 2, in these conditions, we could clearly see KIF2B-MD-CC1-GFP molecules running on microtubules and hitting the microtubule tip. In these conditions, we often see a reduction of EB3 signal, suggesting an alteration at the microtubule plus end in the presence of KIF21B-MD-CC1-GFP.

To quantify this effect, we analyzed fluctuations of EB3 intensity in the presence of 1 nM GFP or 1 nM KIF21B-MD-CC1-GFP in separate chambers on the same coverslip, in the same imaging conditions. In the plots shown in the new Figure 2, we characterized the distributions of EB3 intensities (normalized to the maximum value) during the course of a growth event. If EB3 intensity would be constant, the average value would be close to the maximum value and the distribution would be narrow with a mean in the range of 90-100%. If the signal frequently fluctuates between 0 and 100% of the maximum intensity, the distribution would be more wide and flat, resulting in a smaller mean value and a larger standard deviation (SD). We note that this analysis is not dependent on the absolute growth rate of microtubules, which can affect the absolute EB3 signal, because the analyzed intensities were normalized to the maximum value. In both cases, we excluded from our analysis the EB3 signal during the last phase of growth before catastrophe, since it is known that at this point the comet intensity is reduced (see, for example, Maurer, Fourniol et al., 2012, Cell; Mohan, Katrukha et al., 2013 Proc Natl Acad Sci USA). The data show that in the presence of 1 nM KIF21B-MD-CC1, fluctuations of EB3 signal are more pronounced than in the presence of GFP (Figure 2 of the revised manuscript). These data suggest that the occasional arrivals of KIF21B-MD-CC1-GFP can affect the structure of the growing microtubule tip and this might be the reason behind the slower microtubule growth rate in the presence of this kinesin fragment.

Reviewer #2:

The authors have addressed many critical points, leading to a stronger manuscript. However, the main criticism concerning the authors' claim that a single molecule of KIF21B is capable of causing specific modulation of the microtubule tip behavior has not been addressed fully. Although some statements to this effect have been softened in the main text, this claim is firmly stated in several places, e.g. see impact statement or labels in Figure 5.

*I am concerned about this statement because there are multiple indications in this manuscript that the molecular composition of KIF21B preparation is heterogeneous: the distribution of brightnesses of these molecules is broad (e.g. see Figure 5 and point 2 below), there are visible differences in the brightness of molecules in virtually all kymogpraphs (e.g. see Figure 3) and movies (e.g. Video 2, Figure 3—figure supplement 3), there are visible differences in the size of globules attributed to KIF21B in EM images (some globules are almost 15-20 nm in diameter, exceeding the size of single molecule of Kif21B molecular weight), and the molecular effects at microtubule tip are also heterogeneous (e.g. some molecules induce pausing, while others induce catastrophe). My interpretation of these results is that small oligomers, not just the single molecules of KIF21B, are present in experimental chambers, which is often the case with purified proteins.*

We certainly agree that there is some heterogeneity in our KIF21B preparations. The question is if all effects that we see are caused by the small fraction of multimers that could be present in our preparation. As detailed below, we have now performed a series of additional control experiments that, in our view, justify the interpretation that most motile entities represent one or two KIF21B molecules. Most importantly, we now systematically performed the assays with KIF21B side-by-side with experiments using the robustly dimeric KIF5B motor.

As to the heterogeneity of effects on microtubule growth, we note that this is intrinsic to the dynamic instability of microtubules – some microtubules grow and some switch to shrinking, and this reflects many factors such as “aging” of a microtubule tip. It is possible and even likely that depending on the state of the microtubule tip, a certain type of transition such as pausing or a catastrophe becomes more likely.

To support their claim that only single molecules can elicit certain effects the authors provide few selected examples in which one fluorescent complex appears to lead to some specific effect (e.g. pausing) but the rigorous proof that these specific dots correspond to 1 or 2 Kif21B molecules is lacking. Many or some of these dots, in my opinion, could correspond to complexes of 3-4 molecules, consistent with the heterogeneities listed above. I do not think the slightly larger size of the complexes changes the main message of the paper, but if the authors insist on their quantitative 1-2 molecules statement, I will have to ask for a more convincing evidence. This manuscript is considered for a publication in a very visible and respected journal, so keeping high standards for single molecule aspects in this paper is important.

We agree with the reviewer that even a slightly larger size of a KIF21B cluster will not affect the main conclusions of our paper. We have now formulated our conclusions more carefully.

Specific criticism of the brightness analyses in this paper:

1) The single most important quantification that is lacking in this paper is a quantitative comparison of the brightness of a single GFP molecule with the brightness of the complexes at the MT tip imaged under identical conditions and while taking into account the differences that could not be eliminated. Brightness of single GFP molecule is measured in this paper but it is reported in the manner that prevents any meaningful comparisons. Indeed, brightness of a single GFP molecule is roughly 2,000 au in Figure 1,Figure 5 au in Figure 2; ~1,200 au in Figure 2; in the range of 1,700- 2,500 au in different panels in Figure 5. At the same time, the authors claim that brightness 20 au corresponds to a single motor (2 GFPs) in Figure 2, ~ 40 au and > 60 au in Figure 3—figure supplement 1 A and B. This clearly illustrates a lack of effort to use these quantifications for obtaining a convincing estimate of the number of GFP molecules at the microtubule tip.

The absolute intensity of a single GFP molecule depends on the imaging parameters that could not be kept entirely constant from one experiment to another or were actively altered depending on the aim of the experiment (e.g. exposure time and frame rate). To enable robust quantifications and comparisons, we used for each quantification (Figure 2, Figure 3—figure supplement 1, Figure 5 and Figure 7) an additional control lane of GFP on the same coverslip, with the exactly same TIRF angle, the same focus plane, the same laser power and the same imaging conditions. This is where we put extra effort to provide robust quantifications for obtaining convincing estimates, no matter how different the imaging conditions might be. As shown in the paper, the average GFP intensity value indeed differed 2-5 times among our experiments, but since we always used the same reference GFP lane, we were able to compare the quantifications with each other.

The second point is the order of magnitude difference in the signal intensity for GFP number quantifications (Figure 1, Figure 2 and Figure 5, which are Figure 2, Figure 3—figure supplement 1 and 7A in the new version of the paper) and for kinesin molecules in microtubule dynamics assays (Figure 2, Figure 3, corresponding to new Figure 3). This reflects different laser power levels used for imaging in these conditions. In the first case, we intentionally used high laser power to observe bright single step bleaching events, which happened during 5-10 seconds (Figure 2, Figure 3—figure supplement 1 and 7A of the new version of the paper). For imaging of microtubule dynamics (e.g. Figure 3, Figure 3—figure supplement 1, Figure 3—figure supplement 2, Figure 4, Figure 7 of the new version of the paper), we used 5-10 times lower laser power with 10-12 times lower frame rate to prevent photobleaching of GFP fused to kinesin molecules. This also partly addresses the concerns of the reviewer about photobleaching conditions (see below).

To address this omission, the authors should provide all peak values with errors for their fits and use consistent imaging conditions and data representation, so that all measurements for single GFP reported in this work are consistent and can be used to compare with the brightness of kinesin molecules that elicit certain response at the microtubule tip (e.g. pausing, see previous criticism 2E, which has not been addressed).

All fitted parameters with corresponding errors of fit are provided in [Supplementary-material SD15-data] (please note that within the figures, *median* intensity values are indicated). Further, as argued below, we have also now re-performed experiments with full length KIF21B using a proper dimeric intensity standard within the same experiment (i.e. in a parallel sample chamber). In our opinion, this is the most robust way to address the major concerns of this reviewer.

For accurate comparison, the authors will have to take into account two additional factors that can greatly affect their conclusions:

First, brightness of single GFP molecules was measured in this work at the coverslip surface, while microtubule tips are located at some distance away. I do not believe that measuring this distance for every event can be done (or should be done) accurately, but since the evanescent field decays exponentially, this effect cannot be simply ignored, as in this paper. Approximate estimate of the average depth at which MT tips were imaged should be obtained and used to calculate the brightness at the coverslip surface. Without such an adjustment, the ratio of approximately 1 of Kif21B intensity at the microtubule tip to the GFP intensity on the coverslip (Figure 2) cannot be used to argue that these complexes have the same number of molecules.

Indeed, the exponential decay of the evanescent field can affect single molecule intensity analysis. TIRF angles used for imaging of our assays provided characteristic penetration depths in the range of 80-200 nm, same as in (Grigoriev, Akhmanova Methods in Cell Biology 2010). Under these conditions, if there is a 25 nm difference in z coordinates of two fluorescent molecules (equal to the maximum distance between the coverslip and the top of a microtubule), the very minimal relative intensity of a molecule lying further away from the coverslip should be ~75% of the molecule at the coverslip (Figure 8). For a higher penetration depth, the signal loss will be less.

Author response image 1.**DOI:**
http://dx.doi.org/10.7554/eLife.24746.041

In our recent paper (Doodhi et al., 2014), we have already performed a similar comparison of “kinesin-on-top-of-microtubule” to the intensity of single GFP molecules absorbed on glass, see Supplementary Figure S2B in that paper and Figure 9. At that time, we compared the intensity of KIF5B-560-SxIP-GFP molecules running on the lattice of microtubules to the single GFP intensity of the same molecules attached to the coverslip at the moment just before complete bleaching. Figure 9 shows the two distributions of fluorescent intensities (values of intensities at the microtubule tips are removed from original Figure S2B of Doodhi et al. 2014).

Author response image 2.**DOI:**
http://dx.doi.org/10.7554/eLife.24746.042

KIF5B-560 construct is known to be a dimer. Our analysis provided the ratio of 154/90 = 1.7 between the median intensity values of KIF5B-560-SxIP-GFP and single GFP (estimated from the last bleaching frame), which is very close to a factor of 2. The relative signal can be estimated as 1.7/2 = 83%, which is close to the 75% estimation provided above.

To further validate and improve the intensity analysis in the current paper and address the concerns of the reviewer, we carried out a set of additional experiments. First, we performed comparisons of immobilized single GFP molecules to purified KIF5B-GFP molecules running on microtubules. We compared initial pre-bleach intensity of immobilized GFP to the intensity of KIF5B-GFP (5 nM) molecules moving on dynamic growing microtubules on the same coverslip (Figure 5). The obtained ratio of median intensities KIF5B-GFP/GFP was equal to 1.8, i.e., again very close to the value of 2 (1.8/2=90%). This is in a good agreement with the requirement for KIF5B being a dimer to be able to walk on microtubules and with our previous estimations of the penetration depth. This additional control experiment reassured us that we can get reasonable estimates of GFP numbers in spite of the TIRF evanescence field effect and that we can use KIF5B-GFP moving on microtubules as a fluorescence intensity reference of GFP dimers imaged at the same average height.

We then compared intensities of KIF5B-GFP and KIF21B-FL-GFP measured in two different chambers on the same coverslip in identical conditions. Note that these imaging experiments were performed at a higher frame rate (100 ms vs 1.2 s/frame) and with a more sensitive camera than those shown in the other figures of the paper. This imaging regime allowed for obtaining nicer kymographs; however, it is less suitable for collecting statistics on different events because of higher photobleaching and much smaller number of events observed in each movie.

First, we compared the intensities of KIF5B-GFP with that of KIF21B-FL-GFP running on the microtubule seed, the growth of which was blocked by KIF21B-FL-GFP. The intensity distributions were very similar, suggesting that KIF21B-FL represents mostly dimers (Figure 5). We then proceeded with a comparison of KIF5B-GFP to KIF21B-FL-GFP intensities on dynamic microtubules. We observed multiple KIF5B-GFP runs, while KIF21B-FL-GFP single molecule runs on growing microtubules were rare, because we had to use a very low KIF21B-FL-GFP concentration due to the very potent microtubule growth inhibition by KIF21B.

The observed intensity distributions of moving KIF5B-GFP and those rare events of KIF21B- FL-GFP running on both the seed and newly grown microtubule again were almost identical (Figure 5). In Figure 5, KIF21B-FL-GFP motility along the seed and microtubule lattice, arrival at the microtubule tip, induction of microtubule pausing and switching to depolymerization can be clearly distinguished. Further, we also compared the intensities of KIF21B-GFP molecules running on the seed and on the dynamic microtubule lattices observed on the same microtubule. In all the analyzed examples, the median intensities were found to be very similar (Figure 5, Figure 5—figure supplement 1). These measurements argue against the idea that only larger KIF21B aggregates can “escape” from the seed to the freshly polymerized microtubule lattice.

Together, our new data indicate that the main conclusions of our intensity analysis remain valid when taking into account additional aspects of illumination and acquisition methods.

Second, the authors should provide evidence that they are justified to ignore bleaching. The initial intensity of GFP molecules was measured, while KIF21B dots were visualized for extended time. Few examples of Kif21B intensity kinetics on microtubule shown in this paper are too noisy to make reliable conclusions about the extent of bleaching during the observation time, so this issue should be addressed by direct measurements. This concern is valid especially for KIF21B because, as reported in this paper, this motor first binds at the GMPCPP seeds and only few molecules escape later and run along the GDP lattice. The exact time these "escaped" motors spent on the seed is not known. It is possible that only the relatively large clusters escape and that at this time some of their GFPs have already bleached.

Importantly, both factors, the z distance and bleaching, can lead to underestimation of the size of Kif21B complexes, so the dots that seem to contain 1-2 molecules may in fact be larger. Moreover, It has not been ruled out by the authors that processivity of Kif21B molecules, or their ability to walk on GDP lattice, or the strength of the effects they cause at the MT tip increase with increasing number of clustered molecules. Any one of these plausible possibilities could help to "select" larger complexes from heterogeneous population even if they were present in minor quantities.

As discussed above, KIF21B-FL-GFP motors escaping from the seed to the dynamic lattice have intensities corresponding to single dimers. Furthermore, we analyzed the average photobleaching time in the conditions used for the estimation of the number of molecules (Figure 3, Figure 4, Figure 5 and Figure 7, Figure 1 frame per 1.2 s with 100 ms exposure time) and found that it was ~200 s (Figure 5—figure supplement 2), while the duration of kinesin runs was significantly shorter, on the order of tens of seconds.

To further illustrate that the effect of photobleaching is not significant and that end-binding of one molecule is sufficient to induce a pause followed by a catastrophe, we analyzed the trajectory of a moving KIF21B-FL-GFP molecule from the example shown in Figure 5, where we were illuminating our sample 12 times more frequently (100 ms, stream acquisition) at a similar laser power. In this case, we measured the intensities of moving KIF5B-GFP molecules in a parallel chamber on the same coverslip. The characteristic values of median intensities of KIF5B-GFP are indicated in the plot in Figure 5: as can be seen, the initial intensity of the analyzed individual KIF21B-FL-GFP molecule when it starts its movement on the seed is in the range of 1x median of KIF5B-GFP molecules. According to our previous reference measurements, this means that it is in the range of two GFP molecules, i.e., the molecule is a dimer. While it moves onto the freshly polymerized microtubule lattice, its intensity is reduced by half, which we attribute to the bleaching of one of the GFP molecules. After arriving to the microtubule tip and inducing a pause, the molecule bleaches or desorbs as the microtubule switches to catastrophe. These data show that the laser power used for illumination was gentle enough for 20 seconds of imaging of a single molecule at 10 frames per second. This fits well with the average photobleaching time in these conditions, which was found to be ~16 s at 100 ms/stream acquisition (Figure 5—figure supplement 2). As indicated above, in our regular experiments, where the data are acquired at a 12 times slower frame rate, the average bleaching time is ~12 times higher, and thus much longer than the characteristic duration of kinesin runs. These data indicate that photobleaching does not significantly affect our conclusions.

Other errors in the manuscript:

2) In the rebuttal letter and manuscript text the authors state that the lognormal distribution (Figure 1, Figure 2, Figure 5) of the intensities of individual GFP fluorophores "precisely accounts for the effect of evanescence field on otherwise expected normal distribution of intensities". No reference was provided. I am not familiar with this effect and believe that this statement is wrong. Methods section explains that these intensities were collected from molecules attached to the coverslip surface, so all molecules were positioned in the same plane. The decay in intensity at distances away from the coverslip is therefore irrelevant for interpreting the shape of these experimental distributions. Moreover, even if the molecules were located at variable distances from the coverslips, this would have increased the proportion of molecules with the lower than peak intensity. The lognormal shape of the distribution, characterized by the presence of complexes with higher than peak intensities, can be caused by other experimental factors.

The lognormal distribution of single molecule intensities imaged using TIRF setup is a phenomenon observed before, see for example: Mutch SA, Fujimoto BS,et al. *Deconvolving single-molecule intensity distributions for quantitative microscopy measurements*. Biophys J. 2007 Apr 15;92(8):2926-43; Mehta SB, McQuilken M, et al. *Dissection of molecular assembly dynamics by tracking orientation and position of single molecules in live cells.* Proc Natl Acad Sci U S A. 2016 Sep 27. pii: 201607674; and references therein. However, we agree with the reviewer that we do not have enough experimental data to claim the precise physical mechanism underlying this effect, so we have removed this explanation and used lognormal fitting only as a descriptive tool.

First, by a relatively high density of molecules on the coverslip, so that 2 or 3 single molecules could not be resolved. This explanation is indeed possible in the current work because the authors use 2-9 times higher density than recommended by others for this type of analysis (Jain et al., 2012). Second, the lognormal shape is expected for heterogeneous preparations, so it may simply reflect presence of a significant proportion of complexes that are larger than single molecules. This second possibility supports my concern that the actual size of Kif21B complexes acting at the microtubule tip is larger than stated by the authors.

We assume that the reviewer refers to the paper by Jain, Liu, Xiang and Ha (*Single-molecule pull-down for studying protein interactions.* Nat Protoc. 2012 Feb 9;7(2):445-52), and, more specifically, to Figure 4 in this paper.

We went back to our raw data and performed precise density measurements for our quantifications and found that they were in the recommended range of 0.1-0.4 fluorophores per µm^2^ in all our „counting‟ experiments. We apologize for the confusion; the single molecule density reported in the original version of the paper was given as a very rough estimation (in the text “400-800 molecules per 30x30 µm area”). We have now provided fluorophore density values for each measurement/plot/condition in respective figure legends. As an example, we show typical density in Figure 10 in comparison to Figure 4 of Jain et al., 2012:

Author response image 3.Example image of KIF21B-FL-GFP density used for quantification.It can be seen that the density of single molecules is in the optimal range**DOI:**
http://dx.doi.org/10.7554/eLife.24746.043

</Figure 10 title/legend>

3) New Figure 1 is remarkable because it shows that at 2 nM the MD-CC1 motor runs toward the tip of MT, but at 10 nM it switches polarity and runs toward the seed. The authors will have to explain this result, but I suspect that for 10 nM panel they show imaging of the minus MT end, not the plus. This is very disturbing because it implies that these experiments were done under conditions when discriminating between different MT ends was difficult, the directionality of motor's walking was not taken into account, and the data from different types of ends were combined to show retardation of MT polymerization rate. In Discussion section the authors write: "The dimeric version of the motor is sufficient to inhibit microtubule growth.. It is possible that KIF21B motors arriving at the microtubule tip somehow affect the conformation of terminal tubulin dimers, thereby reducing the tubulin on-rate. " If this is the model, how do the authors explain that the velocity of growth of minus end was also reduced in Kif21B presence? Or is it possible that the reported results are affected by the relative sampling of the plus and minus end kinetics and the authors failed to discriminate these effects? This situation seems unacceptable to me because it raises serious concern about the quality of of data collection and analysis.

We thank the reviewer for spotting this mistake in the preparation of our figure, which we sincerely regret: the shown kymograph indeed represented a microtubule minus end. In fact, microtubule plus and minus ends were very easy to distinguish in this experiment by the slope of kymographs in the kinesin channel. Interestingly, at 10 nM KIF21B-MD-CC1-GFP, the growth rate of microtubule plus and minus ends became very similar, while normally microtubule plus ends grow in vitro approximately twice as fast as the minus ends (see new Figure 2 and Figure 2—figure supplement 3). This observation demonstrates once again that the effect of KIF21B-MD-CC1 is not due to some impurities in the protein, which affect the assay but rather due to a specific effect on microtubule plus end growth. To rule out that potentially incorrect plus-end assignment has introduced an error, we have completely re- analyzed our data but detected no significant differences with the previous analyses (new Figure 2). Furthermore, we included the analysis of microtubule minus end growth rate and showed that it was not affected by 10 nM KIF21B-MD-CC1-GFP (new Figure 2—figure supplement 3)

*4) The authors state that they "..have never observed any pausing in our* in vitro *assays in the presence of EB3 alone, and all observed pauses could be correlated with the presence of KIF21B at the microtubule tip. However, it is true that we did observe events when microtubule polymerization appeared to slow down before the arrival of KIF21B and subsequent pausing." The distinction between "pausing" and "slowing down" is unclear to me. Multiple examples in this paper show pausing/slowing in the absence of KIF21B. See Figure 1—figure supplement 4 (Tub 15 μm with GFP middle of the first track). In Figure 2—figure supplement 1 panel A only first pause is associated with GFP. In Figure 3—figure supplement 3 polymerization pauses at 10-25 s, ie before the arrival of GFP-labeled molecule. Figure 3 panel F lower example clearly shows that MT growth is stalled in the absence of GFP signal. It could either be caused by the bleached GFP-kinesins or by a low quality of tubulin preparation. These issues have not been addressed or even mentioned. Since pausing at the MT end is the primary focus of this paper, I am very concerned about the discrepancy between author's statement and actual images.*

Actually, the statement as written in the paper is correct, and there is no discrepancy between this statement and the actual images. We state that we did not observe pausing *in the presence of EB3 alone*. Current Figure 2—figure supplement 2 (former Figure 1—figure supplement 4) and current Figure 3—figure supplement 1 (former Figure 2—figure supplement 1) show microtubule dynamics *in the absence* of EB3, and we fully acknowledge that some pausing might be present in these conditions (or that pausing and growing microtubules are not easy to distinguish because of the strongly speckled tubulin signal). This is exactly the reason why mCherry-EB3 was included in all our quantitative analyses. The slow growth shown in the current Figure 3—figure supplement 2 (formerly Figure 3—figure supplement 3) can be taken as a pause because of the stretched time scale and low signal of fluorescent tubulin. We now show an enlarged version of the same plot to illustrate the slow growth events better. Furthermore, as can be seen from the kymograph, there is a clear EB3 signal at the tip, meaning that the microtubule is in the growth phase. Finally in the former Figure 3 lower part of panel F (which became panel 3C in the revised paper), there is no stalled microtubule growth in the absence of GFP signal – this panel starts with a microtubule depolymerization event.

Reviewer #3:

van Riel et al. have resubmitted their paper on the kinesin-4 KIF21B. Overall, I feel they have done a fairly solid job of responding to our concerns. The data quality has improved in many of the figure panels. The mass-spec of their contaminants and the analysis of particle brightness are particularly welcome.

It is still very puzzling that a single transient encounter of KIF21B with the microtubule end can slow down the growth rate by half. Their rebuttal just says: "this is how it is"--and they make a point about Eribulin. It was indeed a striking point about their Curr Biol paper that a single eribulin could slow down growth (and produce a secondary EB comet!). But of course it was also the case that microtubule growth resumed after the Eribulin unbinds (e.g., see Figure 4, bottom half).

As explained above in response to the comment of reviewer 1, we propose that transient short perturbations of the growing microtubule tip caused by the arrival of KIF21B-MD-CC1- GFP molecules result in an effective reduction of the microtubule growth rate. The key point of our Eribulin paper (Doodhi et al., 2016) was that perturbing just one protofilament is enough to affect growth of the whole microtubule. We think that this concept is highly pertinent to understanding how single KIF21B motors arriving at the microtubule plus end can slow down its growth. As outlined above in response to the comments of reviewer 1, we now provide further support for this idea by showing that the EB3 signal at growing microtubule plus end is affected (becomes more variable) even at 1 nM KIF21B- MD-CC1.

I was disappointed that they did not perform their experiments at multiple tubulin concentrations. They say that "further mechanistic insights" won't come out of such experiments, but that's clearly not true. Microtubule end structure is thought to change as a function of tubulin concentration, so the motor may struggle/succeed in suppressing growth differently at different tubulin concentrations. I'm on the fence about whether to insist that they perform these experiments. At the end of the day, this paper isn't the kind that deals with biophysical precision of that type, and while we could force them to perform the experiment, it's unclear that their data/interpretation would be definitive.

Since we do not know how exactly KIF21B motor exerts its action on microtubule plus ends (structural work would be needed to determine this), and since, to our best knowledge, it is not clear how exactly microtubule plus end structure changes as a function of different tubulin concentrations, we felt that the combination of these two unknowns will not necessarily lead to a mechanistic insight. Therefore, in the short time allowed to us for the initial revision, we preferred to focus on improving other aspects of the study. However, we agree that such measurements might be useful, and we have now performed them and included them in the revised version of the manuscript (new Figure 2, Figure 2—figure supplement 3). We found that KIF21B-MD-CC1-GFP could almost completely block microtubule growth at 7.5 μM tubulin, while control microtubules could still grow at this tubulin concentration. Furthermore, the reduction of microtubule growth rate in the presence of KIF21B-MD-CC1- GFP was observed at all other tubulin concentrations tested, from 10 to 30 μM.

We also included the data for microtubule growth with 30 μM tubulin in the presence of 0.5 nM full-length KIF21B (Figure 3 and Figure 4—figure supplement 1). In this case, we did observe a significant difference with the samples containing 15 μM: fewer microtubule seeds were blocked and microtubule elongation was observed more readily, suggesting that KIF21B inhibits tubulin addition to microtubule plus end, but this inhibition can be overcome when tubulin concentration is increased. Interestingly, also in this case we observed a reduction of microtubule growth rate, in spite of the fact that KIF21B molecules were not continuously present at elongating microtubule plus ends. This can be explained by the observation that KIF21B molecules that stably attached to microtubule tips often perturbed microtubule growth beyond the point of attachment, likely because only some of the protofilaments could elongate in such situations. In this way, “action at a distance” is indeed possible, because microtubules consist of multiple protofilaments that are stabilized not only by longitudinal but also lateral interactions between tubulin dimers. Therefore, when some protofilaments are prevented from growing, other protofilaments experience hindrance even although they are not blocked (as the number of lateral contacts they can form is reduced), and as a result, a microtubule grows more slowly and undergoes catastrophe more readily, just like we previously observed with Eribulin (Doodhi et al., 2016).

Taken together, all our data are consistent with the view that KIF21B can pause protofilament elongation without causing their disassembly. This can result in an apparent pausing of the whole microtubule, but, in case when some protofilaments are blocked and others continue growing, the overall microtubule growth can become slow and irregular.

[Editors’ note: what now follows is the decision letter after the authors submitted for further consideration.]

Reviewer #2:

In this second revision by van Riel et al., the authors address major criticisms related to whether single molecules of KIF21B are indeed capable of causing microtubule plus ends to pause. There is an abundance of new content, including more intensive data analysis and experiments using kinesin-1 as a dimer control. As a result, the paper is now quite dense (I recognize that this is the outcome of the review process), and I am more convinced that low numbers (likely single molecules) of KIF21B motors are capable of stunting microtubule assembly. The authors are now much more explicit regarding models regarding how KIF21B might act; I found these text additions quite useful.

I would encourage the authors to also update their model figure to include the action of KIF21B on single protofilaments.

We have updated the model in the Figure 7. We have depicted two options of how a single KIF21B molecule can affect microtubule growth without blocking all the protofilaments.

Overall, my disposition is that this manuscript is appropriate for eLife, but there are two issues that should be addressed. The first concerns the relationship between the dwell time of the motor on a protofilament, and the ability of the motor to induce a catastrophe. The MD-CC1 truncation does a good job of slowing microtubule growth, without dwelling appreciably anywhere on the lattice. This is quite a remarkable activity, and an indication that the motor must be doing something subtle to the microtubule lattice/protofilament end. By contrast, the last two constructs that are analyzed in the manuscript show no activity, and the authors therefore conclude that: "Taken together, these results suggest that both the regulatory rCC region and the C-terminal WD-40 containing domain can contribute to the ability of KIF21B to stay attached to the growing MT plus end and to induce pausing." These two interpretations are at odds with each other and must be reconciled.

We apologize for the confusion. The statement included above is indeed correct, and there are no discrepancies. We did not state anywhere that the two mutants described in the last part of the paper, FL-ΔrCC and MD-CC ΔrCC, show no activity at all. We just stated that they do not affect microtubule depolymerization and are much less potent at inducing pausing than the full length molecule. Importantly, these mutants can also induce some catastrophes and some pausing (Figure 7). Moreover, both mutants reduce microtubule growth rate, similar to the full length KIF21B and MD-CC1 mutant. These data are now included in the new Figure 7.

Second, the authors suggest that "It is possible that at growing MT plus ends the number of binding sites for which the C-terminus of KIF21B would have preference or its affinity for these sites would be affected by protofilament curvature." In Figure 4—figure supplement 1, the authors show clear examples of microtubule bending prior to a KIF21B-induced catastrophe, but KIF21B is not bound to these deformed plus ends. The statement in the Discussion is thus countered by their own data. I would suggest that the authors are more explicit regarding their ideas or remove the sentence from the Discussion.

We again apologize for the confusion here. We do not propose that the curling of the microtubules before catastrophe is caused by the kinesin present at the outmost microtubule tip. We propose that microtubule bending occurs because the kinesin blocks some of the protofilaments, while the remaining ones can elongate for some time and, since the microtubule is incomplete, it would be more flexible and more prone to catastrophe. This possibility is now depicted in Figure 7.

In the sentence cited above, we refer to a different kind of curvature – the very mild outward protofilament bending that is generally believed to be found at the outmost microtubule tips and which was suggested to contribute to the preference some proteins, such as doublecortin, have for the microtubule plus ends.